# DriftLite: Lightweight Drift Control for Inference-Time Scaling of Diffusion Models

**Yinuo Ren[1], Wenhao Gao[2], Lexing Ying[3,1], Grant M. Rotskoff[2,1], Jiequn Han[4]***

[1] Institute for Computational and Mathematical Engineering, Stanford University

[2] Department of Chemistry, Stanford University

[3] Department of Mathematics, Stanford University

[4] Center for Computational Mathematics, Flatiron Institute

{yinuoren,whgao,lexing,rotskoff}@stanford.edu
*Correspondence to: Jiequn Han <jhan@flatironinstitute.org>

## Abstract

We study inference-time scaling for diffusion models, where the goal is to adapt a pre-trained model to new target distributions without retraining. Existing guidance-based methods are simple but introduce bias, while particle-based corrections suffer from weight degeneracy and high computational cost. We introduce *DriftLite*, a lightweight, training-free particle-based approach that steers the inference dynamics on the fly with provably optimal stability control. DriftLite exploits a fundamental degree of freedom in the Fokker-Planck equation between the drift and particle potential, and yields two practical instantiations: *Variance- and Energy-Controlling Guidance (VCG/ECG)* for approximating the optimal drift with modest and scalable overhead. Across Gaussian mixture models, particle systems, and large-scale protein-ligand co-folding problems, DriftLite consistently reduces variance and improves sample quality over pure guidance and sequential Monte Carlo baselines. These results highlight a principled, efficient route toward scalable inference-time adaptation of diffusion models. Our source code is publicly available at https://github.com/yinuoren/DriftLite.

## 1 Introduction

Diffusion- (Sohl-Dickstein et al., 2015; Ho et al., 2020; Song & Ermon, 2019; Song et al., 2020) and flow-based (Zhang et al., 2018; Lipman et al., 2022; Albergo & Vanden-Eijnden, 2022; Liu et al., 2022; Ren et al., 2025b) models have revolutionized generative modeling, achieving state-of-the-art performance in domains ranging from creative media synthesis (Rombach et al., 2022; Le et al., 2023; Ho et al., 2022; Austin et al., 2021) to fundamental scientific discovery (Xu et al., 2022; Watson et al., 2023; Duan et al., 2023; Gao et al., 2024; Zhu et al., 2024; Zeni et al., 2025; Duan et al., 2025). They typically rely on a neural network to approximate a time-dependent vector field, which guides a stochastic process from noises to a complex target. However, training is resource-intensive, making it impractical to retrain from scratch for every new setting. This renders a lightweight adaptation of pre-trained models to target distributions that is both compelling and essential.

To this end, a spectrum of adaptation methods has emerged. At one end are guidance-based techniques, the most popular and straightforward inference-time techniques, which inject new information into the drift term, such as classifier (Dhariwal & Nichol, 2021) or classifier-free guidance (Ho & Salimans, 2022) and its many variants (Chung et al., 2022; Trippe et al., 2022; Bansal et al., 2023; Song et al., 2023a;b; He et al., 2023; Guo et al., 2024; Zheng et al., 2024; Rojas et al., 2025). While simple and effective for many tasks, these methods are often heuristic and introduce uncontrolled bias (Chidambaram et al., 2024; Wu et al., 2024a), a significant drawback for scientific applications where sampling accuracy is paramount. On the opposite are methods that resort to extra training, such as fine-tuning (Fan & Lee, 2023; Fan et al., 2023; Black et al., 2023; Clark et al., 2023; Wallace et al., 2024) as in the LLM context (Ouyang et al., 2022; Rafailov et al., 2023), learning within a stochastic control framework (Domingo-Enrich et al., 2024a; Uehara et al., 2024; Thornton et al., 2025), similar to learning-based samplers (Zhang & Chen, 2021; Vargas et al., 2023a;b;

Domingo-Enrich et al., 2024b; Richter & Berner, 2023; Zhu et al., 2025), or adding additional training objectives (Venkatraman et al., 2024; Thornton et al., 2025), but this shifts the computational burden back to retraining, forfeiting the efficiency of inference-time approaches.

Between these ends lies a middle ground of training-free but more sophisticated inference-time approaches. A promising direction formulates the problem in Bayesian and Monte Carlo sampling frameworks (Du et al., 2023; Xu & Chi, 2024; Wu et al., 2024b; Coeurdoux et al., 2024; Bruna & Han, 2024; Zheng et al., 2025). In particular, Sequential Monte Carlo (SMC) methods (Del Moral et al., 2006; Doucet et al., 2000) have been recently introduced to correct for the bias of guidance by simulating the target dynamics with weighted particles (Wu et al., 2023; Cardoso et al., 2023; Skreta et al., 2025; Chen et al., 2025; Singhal et al., 2025; Lee et al., 2025). Despite their strong theoretical grounding and asymptotic guarantees, these particle-based methods face a critical practical bottleneck: *weight degeneracy*. As the simulation progresses, the weights of a few particles grow exponentially while the rest decay, causing the effective sample size to collapse. To mitigate this, one may increase the number of particles, raising computational cost, or use fewer particles, resulting in instability and degraded sample quality.

Our work introduces *DriftLite*, a lightweight, training-free inference-time scaling approach that resolves the inherent instability of particle-based methods without sacrificing mathematical rigor. By exploiting a fundamental degree of freedom in the Fokker-Planck equation, we actively control particle drift on the fly. This proactive steering mechanism absorbs sources of weight variation, preventing the weight collapse common in passive reweighting schemes and dramatically improving stability. The method's modest and scalable computational overhead requiring only the solution of a small linear system per step, makes it fundamentally lightweight, especailly compared with methods that require retraining or extra training of additional control networks. Unlike learning-based control (Vargas et al., 2023b; Richter & Berner, 2023) or heuristic control frameworks (He et al., 2025), DriftLite is a **training-free** solution derived directly from the principle of variance reduction. Comparing with computationally intensive PDE solvers (Albergo & Vanden-Eijnden, 2024) and trajectory balance-based GFlowNet methods (Bengio et al., 2023; Sendera et al., 2024; Venkatraman et al., 2024), our method is applied **on the fly** without any parameter updating or backpropagating through the diffusion model. It is designed to scalably match an entire target distribution in high-dimensional, continuous systems, more rigorous than targeting sample-focused metrics (Ma et al., 2025) or solving problems in simpler discrete settings (Chertkov et al., 2025).

**Our Contributions.** Building on this insight, our work makes the following contributions:

- We formulate and exploit a fundamental degree of freedom in the Feynman-Kac-type Fokker-Planck equation (2.5), establishing a principled trade-off between the particle drift and reweighting potential, and exploit it directly to actively minimize particle weight variance.
- We introduce *DriftLite*, a lightweight and training-free framework that computes a control drift on-the-fly to stabilize the sampling dynamics. We derive two practical instantiations, Variance-Controlling Guidance (VCG) and Energy-Controlling Guidance (ECG), which are computationally efficient and require solving only a small additional linear system at each time step.
- We conduct extensive experiments on challenging benchmarks, including high-dimensional Gaussian mixture models, molecular particle systems, and large-scale protein-ligand co-folding. Our results demonstrate that DriftLite substantially reduces weight variance, stabilizes the Effective Sample Size (ESS), and improves final sample quality over current baselines.

## 2 PRELIMINARIES

In this section, we establish the problem setting, including the fundamentals of diffusion models and the inference-time scaling tasks central to our study.

### 2.1 DIFFUSION MODELS

We begin with a *pre-trained* diffusion or flow-matching model, to which we refer as the base model. This model defines both a forward process $(\boldsymbol{x}_s)_{s \in [0,T]}$ governed by the following stochastic differential equation (SDE) and Fokker-Planck (FP) equation:

$$\mathrm{d}\boldsymbol{x}_s = \boldsymbol{u}_s(\boldsymbol{x}_s)\mathrm{d}s + U_s\mathrm{d}\boldsymbol{w}_s \text{ (SDE)}, \quad \partial_s p_s(\boldsymbol{x}) = -\nabla \cdot [p_s(\boldsymbol{x})\boldsymbol{u}_s(\boldsymbol{x})] + \frac{U_s^2}{2}\Delta p_s(\boldsymbol{x}) \text{ (FP)}, \quad (2.1)$$

where $\boldsymbol{u}_s$ is the forward drift, $p_s$ is the marginal distribution at time $s$, and $(\boldsymbol{w}_s)_{s\geq 0}$ is a Wiener process. $p_0$ represents the data distribution, and $p_T$ is a simple prior, typically a standard Gaussian.

Generative modeling is performed using the backward process. Letting $t = T - s$ be the reverse time and denote $\overleftarrow{*}_t = *_{T-t}$, the backward process $(\overleftarrow{\boldsymbol{x}}_t)_{t\in[0,T]}$ is then described by:

$$\mathrm{d}\overleftarrow{\boldsymbol{x}}_t = \boldsymbol{v}_t(\overleftarrow{\boldsymbol{x}}_t)\mathrm{d}t + V_t\mathrm{d}\boldsymbol{w}_t \text{ (SDE)}, \quad \partial_t\overleftarrow{p}_t(\boldsymbol{x}) = -\nabla\cdot[\overleftarrow{p}_t(\boldsymbol{x})\boldsymbol{v}_t(\boldsymbol{x})] + \frac{V_t^2}{2}\Delta\overleftarrow{p}_t(\boldsymbol{x}) \text{ (FP)},$$

where $\boldsymbol{v}_t$ is the backward drift. The process starts from the noise distribution $\overleftarrow{p}_0 \approx p_T$ and recovers the data distribution $\overleftarrow{p}_T = p_0$. In traditional diffusion models, the backward drift $\boldsymbol{v}_t$ is related to the forward drift $\boldsymbol{u}_s(\boldsymbol{x}_s) = -F_s\boldsymbol{x}_s$ via the score function $\nabla\log\overleftarrow{p}_t$:

$$\boldsymbol{v}_t(\boldsymbol{x}) = -\overleftarrow{\boldsymbol{u}}_t(\boldsymbol{x}) + \frac{\overleftarrow{U}_t^2 + V_t^2}{2}\nabla\log\overleftarrow{p}_t(\boldsymbol{x}). \tag{2.2}$$

The word "pre-trained" signifies that we have access to the forward drift $\boldsymbol{u}_s$ and a reliable NN approximation of the score $\nabla\log\overleftarrow{p}_t$, which in turn defines the backward drift $\boldsymbol{v}_t$.

## 2.2 INFERENCE-TIME SCALING

Our goal is to adapt the generative process of a pre-trained model to new, related tasks at inference time. This approach avoids the significant computational cost and data requirements of retraining from scratch, making it desirable to leverage existing models. We focus on two primary scenarios:

- *Annealing*: Given a factor $\gamma$, the goal is to sample from $q_T \propto p_0^\gamma$. This is common in physics for generating low-temperature samples concentrated around primary modes of a distribution (Karczewski et al., 2024), using a model trained on easier-to-obtain high-temperature data.
- *Reward-Tilting*: Given a reward function $r(\boldsymbol{x})$, the goal is to sample from $q_T \propto p_0\exp(r)$. This can be interpreted as posterior sampling with $p_0$ being the prior and the reward $r$ being the posterior likelihood. It is widely used in applications, such as inverse design (Chung et al., 2022), where the reward function encodes the desired properties of the generated samples.

In this work, we mainly focus on the settings where the reward $r$ is twice-differentiable. In scientific applications, it is typically a physics-based energy with analytic gradients (*e.g.*, Passaro et al. (2025)), which guarantees cheap access to derivatives at inference time. For black-box rewards, one could in principle approximate derivatives using stochastic estimators.

**Distribution Path Selection.** We can unify both scenarios by defining the target compactly as

$$q_T(\boldsymbol{x}) \propto \overleftarrow{p}_T(\boldsymbol{x})^\gamma\exp(r(\boldsymbol{x})) = p_0(\boldsymbol{x})^\gamma\exp(r(\boldsymbol{x})).$$

To sample from $q_T$, we define a modified backward process that evolves along a path of distributions $(q_t)_{t\in[0,T]}$ that smoothly connects from initial noise to our target $q_T$. Following recent works (Skreta et al., 2025; Chen et al., 2025), we adopt a both conceptually and computationally simple path:

$$q_t(\boldsymbol{x}) \propto \overleftarrow{p}_t(\boldsymbol{x})^\gamma\exp\left(r_t(\boldsymbol{x})\right),$$

where the reward $r_t$ interpolates from an initial state $r_0$ chosen such that $q_0$ is easy to sample from, to the final reward $r_T = r$. While more complex paths can be learned via optimal control (Liu et al., 2025), we focus on such pre-defined paths to maintain a training-free framework.

**Guidance-Based Dynamics.** A common and intuitive approach, to which we refer as *pure guidance* (Nichol et al., 2021; Ho & Salimans, 2022), is to inject the new information directly into the drift term by replacing the original score $\nabla\log\overleftarrow{p}_t$ with a heuristic score $\nabla\log q_t$ corresponding to the marginal $q_t$, leading to the following Fokker-Planck equation:

$$\partial_t q_t(\boldsymbol{x}) = -\nabla\cdot[\widetilde{\boldsymbol{v}}_t(\boldsymbol{x})q_t(\boldsymbol{x})] + \frac{V_t^2}{2}\Delta q_t(\boldsymbol{x}), \tag{2.3}$$

where the modified drift $\widetilde{\boldsymbol{v}}_t$ is defined below (*cf.*, Eqn. (2.2)):

$$\widetilde{\boldsymbol{v}}_t(\boldsymbol{x}) = -\overleftarrow{\boldsymbol{u}}_t(\boldsymbol{x}) + \frac{\overleftarrow{U}_t^2 + V_t^2}{2}(\gamma\nabla\log\overleftarrow{p}_t(\boldsymbol{x}) + \nabla r_t(\boldsymbol{x})). \tag{2.4}$$

However, this method is known to be intrinsically biased because it fails to account for the changing normalization constant of $q_t$ over time (Chidambaram et al., 2024). To correct this bias, the true dynamics must include a self-normalizing reweighting term, as formalized below.

**Proposition 2.1** (Guidance-Based Dynamics). *The exact time evolution of the density* $(q_t)_{t \in [0,T]}$ *follows the following Feynman-Kac-type Fokker-Planck equation:*

$$\partial_t q_t(\boldsymbol{x}) = -\nabla \cdot [\widetilde{\boldsymbol{v}}_t(\boldsymbol{x}) q_t(\boldsymbol{x})] + \frac{V_t^2}{2} \Delta q_t(\boldsymbol{x}) + q_t(\boldsymbol{x}) g_t(\boldsymbol{x}), \tag{2.5}$$

*where* $\widetilde{\boldsymbol{v}}_t$ *is the same drift as in pure guidance* (2.4), *and the reweighting potential* $g_t(\boldsymbol{x}) = G_t(\boldsymbol{x}) - \mathbb{E}_{q_t}[G_t(\cdot)]$ *is given by:*

$$G_t = \dot{r}_t - (1-\gamma)\nabla \cdot \overleftarrow{\boldsymbol{u}}_t + \frac{\overleftarrow{U}_t^2}{2}\left(\Delta r_t - \gamma(1-\gamma)\|\nabla \log \overleftarrow{p}_t\|^2\right) + \nabla r_t^\top \left(-\overleftarrow{\boldsymbol{u}}_t + \gamma \overleftarrow{U}_t^2 \nabla \log \overleftarrow{p}_t + \frac{\overleftarrow{U}_t^2}{2}\nabla r_t\right).$$

We refer readers to App. A.1 for the proof. The PDE describes dynamics that diffuse with the guidance drift $\widetilde{\boldsymbol{v}}_t$, while densities continuously reweight according to the centered potential $g_t$.

**Weighted Particle Method.** The corrected PDE (2.5) can be simulated using Sequential Monte Carlo (SMC) (Doucet et al., 2000; Del Moral et al., 2006), where the density $q_t$ is approximated by an empirical distribution formed by an ensemble of $N$ weighted particles $\{\boldsymbol{x}_t^{(i)}, w_t^{(i)}\}_{i \in [N]}$:

$$\begin{cases} \mathrm{d}\boldsymbol{x}_t^{(i)} = \widetilde{\boldsymbol{v}}_t(\boldsymbol{x}_t^{(i)})\mathrm{d}t + V_t \mathrm{d}\boldsymbol{w}_t^{(i)}, & i \in [N], \\ \mathrm{d}\log w_t^{(i)} = \widehat{g}_t(\boldsymbol{x}_t^{(i)}) := G_t(\boldsymbol{x}_t^{(i)}) - \sum_{j=1}^N w_t^{(j)} G_t(\boldsymbol{x}_t^{(j)}), & i \in [N]. \end{cases} \tag{2.6}$$

We refer to this baseline as Guidance-SMC (G-SMC) (Skreta et al., 2025; Chen et al., 2025). This method is provably convergent, with the KL divergence to the target scaling as $\mathcal{O}(N^{-1})$ in the diffusion context (Andrieu et al., 2018; Huggins & Roy, 2019; Domingo-Enrich et al., 2020; Cardoso et al., 2023; Chen et al., 2025). A brief justification of this method is given in App. A.2.

## 3 METHOD: LIGHTWEIGHT DRIFT CONTROL

While the principled dynamics outlined in Prop. 2.1 offer a path to unbiased sampling, their reliance on weighted particles introduces the critical vulnerability of weight degeneracy. As the simulation progresses, the exponential dependency of the weights $w$ on the potential $g_t$ leads to rapid weight degeneracy and collapse of the effective sample size. This instability makes the standard Guidance-SMC approach computationally inefficient, especially with a limited number of particles.

This section introduces our solution: *DriftLite*, a lightweight, training-free framework that actively controls the drift to stabilize the weights. We develop in three steps: (1) we formulate a fundamental degree of freedom in the governing Fokker-Planck equation (2.5), (2) we exploit this freedom to formulate an objective for minimizing the variance of the reweighting potential $g_t$, and (3) we derive two practical, computationally efficient algorithms (VCG and ECG) for achieving this control.

### 3.1 DEGREE OF FREEDOM IN THE FOKKER-PLANCK EQUATION

Our key insight is that we can dynamically modify the particle SDE to counteract the sources of weight variance. Instead of passively reweighting particles, we can proactively steer them by "offloading" the problematic parts of the potential $g_t$ into a new, corrective drift term. This is enabled by a degree of freedom within the Fokker-Planck equation, which we formalize below.

**Proposition 3.1** (Degree of Freedom). *For any control drift* $\boldsymbol{b}_t(\boldsymbol{x})$, *the Feynman-Kac-type Fokker-Planck equation* (2.5) *is equivalent to:*

$$\partial_t q_t(\boldsymbol{x}) = -\nabla \cdot \left[ \left( \widetilde{\boldsymbol{v}}_t(\boldsymbol{x}) + \boldsymbol{b}_t(\boldsymbol{x}) \right) q_t(\boldsymbol{x}) \right] + \frac{V_t^2}{2}\Delta q_t(\boldsymbol{x}) + q_t(\boldsymbol{x})\phi_t(\boldsymbol{x}), \tag{3.1}$$

*where the residual potential is* $\phi_t(\boldsymbol{x}) = g_t(\boldsymbol{x}) + h_t(\boldsymbol{x}; \boldsymbol{b}_t(\boldsymbol{x}))$ *with control potential* $h_t$ *being:*

$$h_t(\boldsymbol{x}; \boldsymbol{b}_t) = (\gamma \nabla \log \overleftarrow{p}_t(\boldsymbol{x}) + \nabla r_t(\boldsymbol{x})) \cdot \boldsymbol{b}_t(\boldsymbol{x}) + \nabla \cdot \boldsymbol{b}_t(\boldsymbol{x}).$$

*Proof Sketch.* The core of the proof is detailed in App. A.3. Briefly, we have $-\nabla \cdot (\boldsymbol{b}_t(\boldsymbol{x})q_t(\boldsymbol{x})) + q_t(\boldsymbol{x})h_t(\boldsymbol{x}; \boldsymbol{b}_t) = 0$, since $h_t(\boldsymbol{x}; \boldsymbol{b}_t)$ is constructed using $\nabla \log q_t = \gamma \nabla \log \overleftarrow{p}_t + \nabla r_t$. An important property is that the correction term has zero expectation under $q_t$, *i.e.*, $\mathbb{E}_{q_t}[h_t(\cdot; \boldsymbol{b}_t)] = 0$. $\square$

This proposition provides a powerful tool: we can introduce any control drift $\boldsymbol{b}_t$ to alter the dynamics, as long as it is compensated by an extra control potential $h_t(\cdot; \boldsymbol{b}_t)$. Since a large variance in

the potential $g_t$ is the direct cause of weight degeneracy, our goal is *to choose $\boldsymbol{b}_t$ strategically to minimize the variance of the new residual potential $\phi_t$.* An ideal control would make $\phi_t$ constant, completely stabilizing the particle weights. In fact, a perfect, variance-eliminating control always exists for any base potential $g_t$, as shown in the following proposition:

---

**Proposition 3.2** (Optimal Control, Informal Version). *There exists a unique curl-free control $\boldsymbol{b}_t^*(\boldsymbol{x}) = \nabla A_t^*(\boldsymbol{x})$ such that $\phi_t^*(\boldsymbol{x}) = g_t(\boldsymbol{x}) + h_t(\boldsymbol{x}; \boldsymbol{b}_t^*) = 0$ for all $\boldsymbol{x}$, where the optimal scalar potential $A_t^*(\boldsymbol{x})$ is the solution to the following Poisson equation:*

$$\nabla \cdot (q_t(\boldsymbol{x}) \nabla A_t^*(\boldsymbol{x})) = -q_t(\boldsymbol{x}) g_t(\boldsymbol{x}). \tag{3.2}$$

---

The proof and further discussion are provided in App. A.4. Intuitively, Eqn. (3.2) follows by noticing that the control potential satisfies $q_t(\boldsymbol{x}) h_t(\boldsymbol{x}; \boldsymbol{b}_t) = \nabla \cdot (q_t(\boldsymbol{x}) \boldsymbol{b}_t(\boldsymbol{x}))$ from its definition.

This mechanism is closely related to twisted proposals in SMC (Briers et al., 2010; Whiteley & Lee, 2014; Heng et al., 2020) and its recent applications (Lawson et al., 2022; Zhao et al., 2024; Lu & Wang, 2024) and Prop. 3.1 can be viewed as a diffusion formulation of this degree of freedom.

## 3.2 IN SEARCH OF OPTIMAL CONTROL

While Prop. 3.2 guarantees a perfect solution, solving the high-dimensional PDE in (3.2) at every time step is computationally intractable. In contrast to learning-based methods that uses neural networks to approximate the optimal control via backpropagation (Albergo & Vanden-Eijnden, 2024; Vargas et al., 2023b) , we propose two training-free, practical methods that approximate this optimal control by balancing effectiveness with efficiency. Both methods share a core strategy: restricting the search for the control drift $\boldsymbol{b}_t$ to a finite-dimensional subspace. This simplification is key, as it transforms the complex problem of minimizing the residual potential $\phi_t$ into solving a small linear system. This reduction from an intractable PDE to a tractable linear solve makes the control truly lightweight, hence the name *DriftLite*.

**Variance-Controlling Guidance (VCG).** The most direct approach is to find a control $\boldsymbol{b}_t$ that explicitly minimizes the variance of the residual potential:

$$\min_{\boldsymbol{b}_t} \mathrm{Var}_{\boldsymbol{x} \sim q_t} [\phi_t(\boldsymbol{x})] = \mathrm{Var}_{\boldsymbol{x} \sim q_t} [g_t(\boldsymbol{x}) + h_t(\boldsymbol{x}; \boldsymbol{b}_t)]. \tag{3.3}$$

Instead of parameterizing $\boldsymbol{b}_t$ with a neural network (Albergo & Vanden-Eijnden, 2024), we seek a lightweight solution by approximating it as a linear combination of basis functions.

---

**Ansatz 3.3** (Linear Control Drift). *The optimal control drift $\boldsymbol{b}_t^*(\boldsymbol{x})$ is approximated as $\boldsymbol{b}_t(\boldsymbol{x}) = \sum_{i=1}^n \theta_t^i \boldsymbol{s}_i(\boldsymbol{x})$, where $\{\boldsymbol{s}_i(\boldsymbol{x})\}_{i \in [n]}$ are pre-defined vector bases and $\boldsymbol{\theta}_t = (\theta_t^1, \cdots, \theta_t^n)^\top$ are the coefficients to be found.*

---

Under this ansatz, the residual potential becomes $\phi_t(\boldsymbol{x}) = g_t(\boldsymbol{x}) + \sum_{i=1}^n \theta_t^i h_t^i(\boldsymbol{x})$, where $h_t^i(\boldsymbol{x}) = h_t(\boldsymbol{x}; \boldsymbol{s}_i)$. The objective (3.3) corresponds to a standard least-square problem, whose solution is obtained by solving an $n \times n$ linear system $\boldsymbol{A}_t \boldsymbol{\theta}_t = \boldsymbol{c}_t$, where $\boldsymbol{A}_{ij} = \mathbb{E}_{q_t}[h_t^i h_t^j]$ and $\boldsymbol{c}_i = -\mathbb{E}_{q_t}[g_t h_t^i]$.

**Energy-Controlling Guidance (ECG).** An alternative approach directly targets the curl-free optimal control $\boldsymbol{b}_t^*$ in Prop. 3.2 by variationally solving the Poisson equation (3.2). As shown by Yu & E (2018), this equation is the Euler-Lagrange equation for the following energy functional:

$$\min_{A_t} \mathcal{E}_t[A_t] = \int \left( \frac{1}{2} q_t(\boldsymbol{x}) \|\nabla A_t(\boldsymbol{x})\|^2 - q_t(\boldsymbol{x}) g_t(\boldsymbol{x}) A_t(\boldsymbol{x}) \right) \mathrm{d}\boldsymbol{x}. \tag{3.4}$$

We can efficiently find an approximate minimizer using the Ritz method for the scalar potential $A_t$.

---

**Ansatz 3.4** (Linear Control Potential). *The optimal scalar potential $A_t^*(\boldsymbol{x})$ is approximated as $A_t(\boldsymbol{x}) = \sum_{i=1}^n \theta_t^i s_t^i(\boldsymbol{x})$, where $\{s_t^i(\boldsymbol{x})\}_{i \in [n]}$ are scalar bases. The control drift is then given by $\boldsymbol{b}_t(\boldsymbol{x}) = \nabla A_t(\boldsymbol{x}) = \sum_{i=1}^n \theta_t^i \nabla s_t^i(\boldsymbol{x})$.*

---

Substituting into the energy functional (3.4) again yields a linear system of equations $\boldsymbol{A}_t \boldsymbol{\theta}_t = \boldsymbol{c}_t$, where $\boldsymbol{A}_{ij} = \mathbb{E}_{q_t}[\nabla s_t^{i\top} \nabla s_t^j]$ and $\boldsymbol{c}_i = \mathbb{E}_{q_t}[g_t s_t^i]$.

### 3.3 PRACTICAL IMPLEMENTATION

**Choice of Bases.** The effectiveness of VCG and ECG depends on the choice of suitable basis functions. While the formal solution for the optimal control $\boldsymbol{b}_t^*$ is intractable (*cf.*, App. A.4), its structure reveals that the ideal control is a function of temporally locally available quantities like the score $\nabla \log \overleftarrow{p}_t$, the reward gradient $\nabla r_t$, and the potential $g_t$ (containing the forward drift $\overleftarrow{\boldsymbol{u}}_t$ and higher-order terms). This suggests that a natural low-rank approximation is obtained by projecting the optimal control onto the span of locally available vector fields, as defined in the following.

- *Variance-Controlling Guidance (VCG):* We use the following vector basis functions:
$$\boldsymbol{s}_1(\boldsymbol{x}) = \nabla r_t(\boldsymbol{x}), \quad \boldsymbol{s}_2(\boldsymbol{x}) = \nabla \log \overleftarrow{p}_t(\boldsymbol{x}), \quad \boldsymbol{s}_3(\boldsymbol{x}) = \overleftarrow{\boldsymbol{u}}_t(\boldsymbol{x}).$$
  Note that using $\boldsymbol{s}_2$ requires computing the Laplacian $\Delta \log \overleftarrow{p}_t(\boldsymbol{x})$, which can be approximated efficiently with Hutchinson's trace estimator (Hutchinson, 1989) in high dimensions.
- *Energy-Controlling Guidance (ECG):* We use the corresponding scalar potentials:
$$s_1(\boldsymbol{x}) = r_t(\boldsymbol{x}), \quad s_2(\boldsymbol{x}) = \log \overleftarrow{p}_t(\boldsymbol{x}), \quad s_3(\boldsymbol{x}) = \overleftarrow{U}_t(\boldsymbol{x}),$$
  where $\overleftarrow{U}_t$ is a potential such that $\nabla \overleftarrow{U}_t = \overleftarrow{\boldsymbol{u}}_t$. This method is especially convenient when the log-likelihood $\log \overleftarrow{p}_t$ is readily available from upstream training tasks (Akhound-Sadegh et al., 2025; Guth et al., 2025; Thornton et al., 2025). If not, approximations or alternative bases may be used, such as the score norm $\|\nabla \log \overleftarrow{p}_t\|^2$ or random projections of the score $\nabla \log \overleftarrow{p}_t \cdot \boldsymbol{\xi}$ for random $\boldsymbol{\xi}$.

For annealing tasks, reward-based bases ($\boldsymbol{s}_1$ and $s_1$) are automatically dismissed.

**Weighted Particle Simulation.** As discussed in Sec. 2.2, we simulate the Feynman-Kac-type Fokker-Planck equation (3.1) using the SMC/weighted particle method detailed in Alg. 1. The key difference from G-SMC Eqn. (2.6) is the use of the controlled drift $\widetilde{\boldsymbol{v}}_t + \boldsymbol{b}_t$ and the residual potential $\phi_t = g_t + h_t(\cdot; \boldsymbol{b}_t)$. To prevent weight collapse, particles are resampled when the Effective Sample Size (ESS) drops below a threshold $\tau$. These principled versions with resampling are denoted VCG-SMC/ECG-SMC. For high-dimensional problems where resampling introduces additional stochastic instability, we also consider simpler variants, denoted VCG/ECG, which use the low-variance residual potential $\phi_t$, retain continuous path-level weights, but omit resampling steps.

Our method adds modest computational overhead. The linear solve itself is negligible, while the dominant additional cost comes from evaluating basis functions and building a small $n \times n$ linear system at each time step, where $n$ is the number of bases, typically $n \leq 3$ in our experiments. The components of this system ($\boldsymbol{A}_t$ and $\boldsymbol{c}_t$) are computed as expectations over the current weighted particles, reusing terms like the score $\nabla \log \overleftarrow{p}_t$ and the reward gradient $\nabla r_t$ that are already computed for the base guidance drift. While accurate evaluation of the score Laplacian $\Delta \log \overleftarrow{p}_t$ can improve control quality, efficiency is preserved with stochastic approximations, and thus the per-step overhead remains constant in dimension and fully parallelizable across particles, resulting in moderate runtime increase compared to the pure guidance baseline (*cf.*, empirical results in Tabs. 5 and 8).

## 4 EXPERIMENTS

In this section, we empirically test the performance of DriftLite by designing a series of challenging annealing and reward-tilting tasks, comparing our DriftLite methods (VCG and ECG with and without SMC) against two key baselines: Pure Guidance (PG) (2.3) (Ho & Salimans, 2022), Guidance-SMC (G-SMC) (2.5) (Skreta et al., 2025; Chen et al., 2025). Our implementation uses JAX (Bradbury et al., 2018) to ensure efficient, parallelized computation on GPUs. Our source code is publicly available at `https://github.com/yinuoren/DriftLite`.

### 4.1 GAUSSIAN MIXTURE MODEL

We begin with a 30-dimensional Gaussian Mixture Model (GMM) (*cf.*, App. B.1 for detailed settings), a controlled environment where the exact score $\nabla \log p_t$ and the potential $\log p_t$ are known analytically, allowing us to isolate and evaluate the performance of the sampling algorithms themselves, free from any confounding errors of a learned score network. We evaluate the methods with multiple metrics, including the Negative Log-Likelihood difference ($\Delta$NLL), Maximum Mean Discrepancy (MMD), and Sliced Wasserstein Distance (SWD) (*cf.*, App. B.5).

**Annealing.** We first test the ability to sharpen the GMM's modes by annealing, which tests each method's ability to maintain the correct relative mode weights. As shown in Fig. 1, the pure guid-

---

**Algorithm 1:** DriftLite-VCG/ECG-SMC Implementation

---

**Input:** Guidance drift path $\widetilde{\boldsymbol{v}}_t$ from (2.4), uncentered potential path $G_t$ from Prop. 2.1, time steps $\{t_k\}_{k=0}^M$, reward $r(\boldsymbol{x})$, schedule $\beta_t$, basis functions, number of particles $N$, ESS threshold $\tau$.

1 Initialize particles $\boldsymbol{x}_0^{(i)} \sim \overleftarrow{p}_0$, log-weights $\ell_0^{(i)} \leftarrow 0$, and weights $w_0^{(i)} \leftarrow \frac{1}{N}$ for $i = 1, \ldots, N$;

2 **for** $k \leftarrow 0$ *to* $M - 1$ **do**

3     Set $\Delta t_k \leftarrow t_{k+1} - t_k$ and form estimates of $\boldsymbol{A}_{t_k}$ and $\boldsymbol{c}_{t_k}$ using $\{(\boldsymbol{x}_{t_k}^{(i)}, w_{t_k}^{(i)})\}_{i \in [N]}$;

4     Solve $\boldsymbol{A}_{t_k} \boldsymbol{\theta}_{t_k} = \boldsymbol{c}_{t_k}$ to obtain the control drift $\boldsymbol{b}_{t_k}(\cdot)$;

5     $\boldsymbol{v}_{t_k}^{\text{eff}}(\cdot) \leftarrow \widetilde{\boldsymbol{v}}_{t_k}(\cdot) + \boldsymbol{b}_{t_k}(\cdot), \quad H_{t_k}^{\text{eff}}(\cdot) \leftarrow G_{t_k}(\cdot) + h_{t_k}(\cdot; \boldsymbol{b}_{t_k});$

6     $\widehat{g}_{t_k}^{\text{eff}}(\boldsymbol{x}_{t_k}^{(i)}) \leftarrow H_{t_k}^{\text{eff}}(\boldsymbol{x}_{t_k}^{(i)}) - \sum_{j=1}^N w_{t_k}^{(j)} H_{t_k}^{\text{eff}}(\boldsymbol{x}_{t_k}^{(j)});$

7     $\ell_{t_{k+1}}^{(i)} \leftarrow \ell_{t_k}^{(i)} + \widehat{g}_{t_k}^{\text{eff}}(\boldsymbol{x}_{t_k}^{(i)}) \Delta t_k, \quad \boldsymbol{w}_{t_{k+1}} \leftarrow \text{softmax}(\boldsymbol{\ell}_{t_{k+1}});$

8     $\boldsymbol{x}_{t_{k+1}}^{(i)} \leftarrow \boldsymbol{x}_{t_k}^{(i)} + \boldsymbol{v}_{t_k}^{\text{eff}}(\boldsymbol{x}_{t_k}^{(i)}) \Delta t_k + V_{t_k} \sqrt{\Delta t_k}\, \boldsymbol{z}^{(i)}$, where $\boldsymbol{z}^{(i)} \sim \mathcal{N}(0, \boldsymbol{I})$;

9     **if** *ESS*$(\boldsymbol{w}_{t_{k+1}}) < \tau$ *or periodically* **then**

10         Resample $\{\boldsymbol{x}_{t_{k+1}}^{(i)}\}_{i \in [N]}$ according to $\{w_{t_{k+1}}^{(i)}\}_{i \in [N]}$ and reset $\ell_{t_{k+1}}^{(i)} \leftarrow 0$,
        $w_{t_{k+1}}^{(i)} \leftarrow \frac{1}{N}$ for all $i$;

**Output:** Final samples $\{\boldsymbol{x}_T^{(i)}, w_T^{(i)}\}_{i \in [N]}$ from the last completed pass.

---

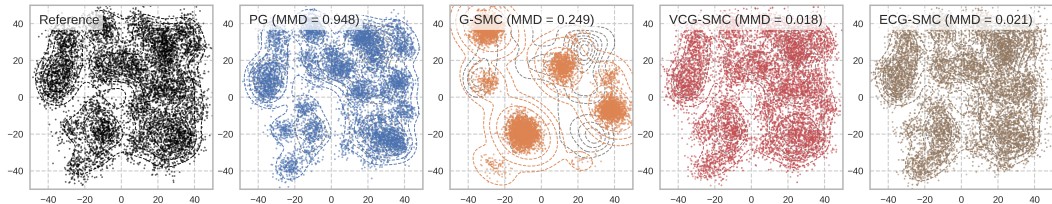

Figure 1: Qualitative comparison of sampling methods on the GMM annealing task ($\gamma = 2.5$).

ance (PG) method produces visibly biased samples, while G-SMC suffers from mode collapse, a direct consequence of the weight degeneracy that our work aims to solve. In contrast, our methods (VCG and ECG) accurately sample from the correct modes, also corroborated with quantitative comparisons in Tab. 3. A closer look at the ESS and potential variance evolution during the inference dynamics Fig. 2 reveals why DriftLite succeeds. Our control mechanism reduces the variance of the reweighting potential by several orders of magnitude compared to G-SMC. This directly prevents weight degeneracy, leading to a stable Effective Sample Size (ESS) throughout the simulation and superior final sample quality. Notably, ECG, while not directly minimizing variance, achieves a similar stabilizing effect, validating the energy-based control perspective. Fig. 3 shows the performance of all methods as the number of particles varies. It indicates that our methods not only outperform the baselines but also converge more efficiently, achieving better results with fewer particles.

**Reward-Tilting.** The results of the reward-tilting task where the distribution is shifted towards a region defined by a quadratic reward (Figs. 4, 9 and 10 and Tab. 4) confirm our findings from the annealing task. We refer to App. C.1 for further experimental results.

**Learning-Based Baseline.** To contrast DriftLite's training-free inference-time control with amortized drift-learning, we additionally implement a training-based baseline, *Neural Controlling Guidance (NCG)*, which parameterizes the control drift by a neural network and optimizes the variance objective (3.3) via backpropagation. See App. C.1 for full setup details and quantitative results.

**Iterative Refinement.** Furthermore, we introduce an iterative refinement procedure, where the learned control drift $\widetilde{\boldsymbol{v}}_t + \boldsymbol{b}_t$ and potential $\phi_t = g_t + h_t(\cdot; \boldsymbol{b}_t)$ from one full pass are used as the base dynamics for the next. As further discussed in App. C.4, this process progressively reduces variance and stabilizes ESS over multiple rounds (*cf.*, Figs. 17 and 18), further enhancing sample quality (*cf.*,

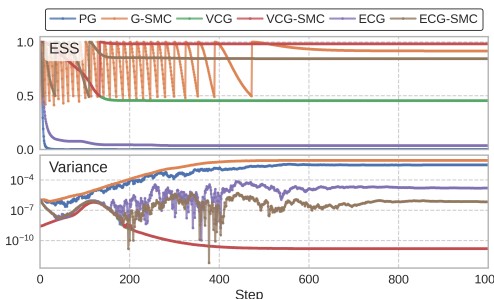
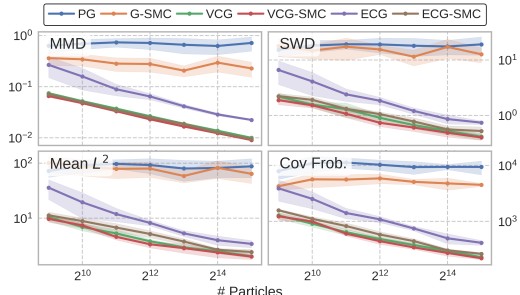

Figure 2: Evolution of ESS and potential variance during inference on the GMM annealing task ($\gamma = 2.2$). Our methods (VCG/ECG) substantially reduce variance and stabilize ESS.

Figure 3: Performance metrics versus number of particles for the GMM annealing task ($\gamma = 2.0$). Our methods consistently outperform baselines and show strong scaling.

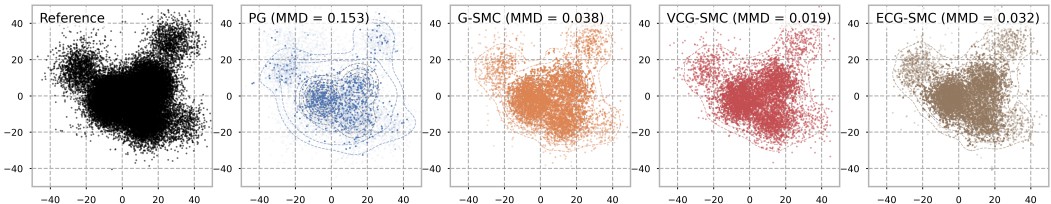

Figure 4: Qualitative comparison of sampling methods on the GMM reward-tilting task ($\sigma = 200.0$).

Tabs. 15 and 16). The monotonic decrease in variance across rounds also acts as a proxy for the reduction of approximation error in the linear control ansatz (Thms. 3.3 and 3.4).

## 4.2 PARTICLE SYSTEMS

Next, we move to more realistic scientific benchmarks where the score is approximated by an NN trained on finite data. We evaluate on two standard systems with complex, multimodal energy landscapes: a 2D 4-particle Double-Well (DW-4) and a 3D 13-particle Leonard-Jones system (LJ-13), both widely used as benchmarks (Klein et al., 2023; Akhound-Sadegh et al., 2024; 2025; Liu et al., 2025; Skreta et al., 2025; Zhang et al., 2025).

The score is obtained by training an $E(n)$-Equivariant Graph Neural Network (EGNN) (Satorras et al., 2021b) (*cf.*, App. B.2). For all particle systems, ground-truth reference is obtained from underdamped Langevin dynamics simulations with BAOAB splitting scheme (Leimkuhler & Matthews, 2013) (*cf.*, App. B.3). The EDM framework (Karras et al., 2022) is adopted for both training and inference (*cf.*, App. B.4). We measure performance using additional metrics that capture physical correctness, including the Radial Distribution Function (RDF) for structure and the energy distribution for thermodynamics (*cf.*, App. B.5). Based on the GMM results showing VCG's superior performance over ECG and the lack of pre-trained log-likelihood, we proceed with only the VCG variants of DriftLite in the following experiments.

**Double-Well-4 (DW-4).** We first consider the DW-4 system (*cf.*, App. B.1). This system features two energy minima separated by a barrier. Annealing requires the sampler to correctly populate both modes, even when sharpened at low temperatures. As shown in Fig. 13, VCG-SMC achieves a nearly perfect match with the ground-truth RDF and energy distribution. This demonstrates that by reducing variance to maintain an ensemble of high-quality particles, DriftLite effectively leverages global information to navigate challenging energy landscapes where baselines fail to do so.

Motivated by Schebek et al. (2024), we consider applying an additional harmonic potential as a reward, and the reward-tilted distribution corresponds to another DW-4 system with a different configuration. The quantitative results in Tab. 1 confirm that our methods consistently outperform baselines by a large margin across all metrics. The ESS/potential variance plot in Fig. 11 confirms the stabilizing effect of our method on ESS. An ablation study in Fig. 12 demonstrates that our method converges as the number of particles increases across metrics.

Table 1: Performance comparison on Particle Systems (DW-4 and LJ-13). Results are mean$_{\pm\text{std}}$ over 5 runs. Best results per column are in bold.

| Method | DW-4, Annealing ($T = 2.0, \gamma = 2.0$) | | | | | DW-4, Reward-Tilting ($T = 2.0, \lambda' = 0.5$) | | | | |
|---|---|---|---|---|---|---|---|---|---|---|
| | $\Delta$NLL | MMD | SWD | $W_1^{\text{RDF}}$ | $W_1^{\mathcal{E}}$ | $\Delta$NLL | MMD | SWD | $W_1^{\text{RDF}}$ | $W_1^{\mathcal{E}}$ |
| PG | $0.159_{\pm 1.232}$ | $0.400_{\pm 0.168}$ | $1.088_{\pm 0.384}$ | $0.208_{\pm 0.008}$ | $0.551_{\pm 0.009}$ | $0.867_{\pm 1.437}$ | $0.771_{\pm 0.085}$ | $1.714_{\pm 0.232}$ | $0.627_{\pm 0.003}$ | $1.837_{\pm 0.013}$ |
| G-SMC | $0.038_{\pm 0.338}$ | $0.365_{\pm 0.058}$ | $1.012_{\pm 0.253}$ | $0.208_{\pm 0.146}$ | $0.190_{\pm 0.080}$ | $0.329_{\pm 0.016}$ | $0.087_{\pm 0.039}$ | $0.194_{\pm 0.082}$ | $0.118_{\pm 0.004}$ | $0.330_{\pm 0.016}$ |
| VCG | $-0.043_{\pm 0.022}$ | $0.014_{\pm 0.001}$ | $0.037_{\pm 0.008}$ | $\mathbf{0.043_{\pm 0.002}}$ | $0.663_{\pm 0.015}$ | $0.699_{\pm 1.905}$ | $0.614_{\pm 0.139}$ | $1.692_{\pm 0.438}$ | $0.161_{\pm 0.033}$ | $0.461_{\pm 0.094}$ |
| VCG-SMC | $\mathbf{-0.032_{\pm 0.009}}$ | $\mathbf{0.014_{\pm 0.001}}$ | $\mathbf{0.035_{\pm 0.002}}$ | $0.060_{\pm 0.006}$ | $\mathbf{0.031_{\pm 0.007}}$ | $\mathbf{0.296_{\pm 0.016}}$ | $\mathbf{0.021_{\pm 0.001}}$ | $\mathbf{0.048_{\pm 0.002}}$ | $\mathbf{0.107_{\pm 0.005}}$ | $\mathbf{0.296_{\pm 0.016}}$ |

| Method | LJ-13, Annealing ($T = 2.0, \gamma = 2.5$) | | | | | LJ-13, Reward-Tilting ($T = 2.0, \lambda' = 0.8$) | | | | |
|---|---|---|---|---|---|---|---|---|---|---|
| | $\Delta$NLL | MMD | SWD | $W_1^{\text{RDF}}$ | $W_1^{\mathcal{E}}$ | $\Delta$NLL | MMD | SWD | $W_1^{\text{RDF}}$ | $W_1^{\mathcal{E}}$ |
| PG | $13.58_{\pm 16.73}$ | $0.603_{\pm 0.095}$ | $0.598_{\pm 0.087}$ | $0.037_{\pm 0.000}$ | $11.40_{\pm 0.129}$ | $4.975_{\pm 5.159}$ | $0.719_{\pm 0.021}$ | $0.797_{\pm 0.058}$ | $0.070_{\pm 0.001}$ | $5.430_{\pm 0.074}$ |
| G-SMC | $13.64_{\pm 9.949}$ | $0.616_{\pm 0.012}$ | $0.523_{\pm 0.060}$ | $0.040_{\pm 0.030}$ | $13.48_{\pm 9.626}$ | $1.783_{\pm 0.202}$ | $0.081_{\pm 0.019}$ | $0.084_{\pm 0.032}$ | $0.023_{\pm 0.002}$ | $1.784_{\pm 0.202}$ |
| VCG | $-1.084_{\pm 0.931}$ | $0.136_{\pm 0.032}$ | $0.144_{\pm 0.044}$ | $0.069_{\pm 0.001}$ | $22.86_{\pm 0.177}$ | $7.629_{\pm 12.37}$ | $0.695_{\pm 0.127}$ | $0.702_{\pm 0.174}$ | $0.036_{\pm 0.020}$ | $2.790_{\pm 1.531}$ |
| VCG-SMC | $\mathbf{-0.699_{\pm 0.189}}$ | $\mathbf{0.102_{\pm 0.050}}$ | $\mathbf{0.098_{\pm 0.044}}$ | $\mathbf{0.002_{\pm 0.000}}$ | $\mathbf{0.286_{\pm 0.100}}$ | $\mathbf{1.734_{\pm 0.106}}$ | $\mathbf{0.015_{\pm 0.001}}$ | $\mathbf{0.015_{\pm 0.001}}$ | $\mathbf{0.022_{\pm 0.001}}$ | $\mathbf{1.735_{\pm 0.106}}$ |

(a) Radial Distribution Function.  (b) Energy Distribution.

Figure 5: Comparison of generated distributions for the LJ-13 annealing task ($\gamma = 2.5$). VCG-SMC is the only method that successfully recovers all three peaks in the (a) RDF and closely matches the (b) Energy Distribution. Insets provide a zoomed-in view.

**Lennard-Jones-13 (LJ-13).** We conclude with a highly challenging annealing task on the LJ-13 system (*cf.*, App. B.1), a complex benchmark known for its rugged energy landscape and singular behaviors at short distances. Fig. 5 presents the result of a demanding inference-time annealing task from $T = 1.0$ to $0.4$ with $\gamma = 2.5$. The target distribution exhibits a third peak in its RDF corresponding to a structural feature almost absent at the initial temperature (Fig. 8). In a powerful demonstration of its capabilities, VCG-SMC is the only method that successfully discovers and samples from all modes, matching both the RDF and energy distribution with high precision. Metrics in Tab. 1 further confirm a significant performance gap over the baselines in this complex setting.

We refer readers to App. C.2 for additional results and visualization on DW-4 and LJ-13, with varying base temperatures $T$, annealing factor $\gamma$, constraint strength $\lambda'$, and number of particles $N$.

## 4.3 PROTEIN-LIGAND CO-FOLDING

Lastly, we apply DriftLite to the protein-ligand co-folding problem (Abramson et al., 2024; Bryant et al., 2024), a central task of structural biology and drug discovery. The goal is to generate 3D protein structures and their binding partners (ligands, particularly small molecules) simultaneously and in a mutually dependent manner, given the protein sequence and the ligand identity. This problem extends the classical protein folding problem (Jumper et al., 2021; Baek et al., 2021) and is crucial for elucidating protein-ligand interactions. Despite the recent progress achieved by diffusion models, notably AlphaFold3 (Abramson et al., 2024), Protenix (Team et al., 2025), and Boltz-2 (Passaro et al., 2025), a persistent challenge is that purely data-driven generative approaches tend to overemphasize global structural similarity while often producing conformations that violate basic physical constraints (Buttenschoen et al., 2024; Masters et al., 2024). Recent studies demonstrated that incorporating physics-based steering potentials can help mitigate this limitation (Passaro et al., 2025).

We adopt the experimental setup of Boltz-2 (Passaro et al., 2025), an open-weight diffusion model, as base, and apply VCG-SMC to steer the generation of protein-ligand structures toward physically

Table 2: Performance comparison on steering the physical validity of protein-ligand co-folding. Results are mean$_{\pm\text{std}}$ over 3 runs. Best results per column are in bold.

| Method | Valid Fraction ↑ | Clash Free Fraction ↑ | Bond Length ↓ | Bond Angle ↓ | Internal Clash ↓ | Chiral Atom ↓ | Chain Clashes ↓ |
|---|---|---|---|---|---|---|---|
| Base | $0.374_{\pm0.003}$ | $0.490_{\pm0.007}$ | $55.00_{\pm3.61}$ | $133.00_{\pm7.00}$ | $138.67_{\pm4.04}$ | $118.33_{\pm12.74}$ | $398.67_{\pm4.16}$ |
| FKS | $0.379_{\pm0.014}$ | $0.490_{\pm0.007}$ | $52.67_{\pm2.89}$ | $127.33_{\pm5.69}$ | $140.33_{\pm2.08}$ | $126.33_{\pm5.51}$ | $377.00_{\pm20.30}$ |
| G-SMC | $0.838_{\pm0.008}$ | $0.945_{\pm0.005}$ | $42.33_{\pm13.05}$ | $98.00_{\pm23.07}$ | $\mathbf{31.33}_{\pm4.93}$ | $2.33_{\pm0.58}$ | $31.67_{\pm1.53}$ |
| **VCG-SMC** | $\mathbf{0.856}_{\pm0.008}$ | $\mathbf{0.950}_{\pm0.003}$ | $\mathbf{24.33}_{\pm9.29}$ | $\mathbf{61.00}_{\pm19.08}$ | $32.33_{\pm4.16}$ | $\mathbf{1.00}_{\pm1.00}$ | $\mathbf{30.00}_{\pm1.00}$ |

| Reference | Base | G-SMC | VCG-SMC |
|---|---|---|---|

Figure 6: The reference and predicted complex structure of Hst2 bound to 2'-O-benzoyl ADP ribose. The reference corresponds to the experimentally determined crystal structure (PDB ID: 7F51). The unsteered base prediction inverted a chiral center in the ligand (highlighted with a red box). G-SMC failed to correct this issue and even broke the bonding, whereas VCG-SMC successfully guided the generation toward the correct chirality and preserved a chemically meaningful structure.

valid conformations using a physics-based potential as reward. We compare our method with two additional baselines: the unsteered model (Base) and Feynman-Kac Steering (FKS) (Singhal et al., 2025). We assess physical validity using the widely adopted PoseBuster V2 benchmark (Buttenschoen et al., 2024). Results are summarized in Tab. 2. In this task, evaluating the reward and its gradient adds relatively little overhead. VCG-SMC exhibits the strongest performance with fewer or without rule violations, improving the quality of partially valid structures, and increasing the proportion of fully valid ones. This underscores its effectiveness in a complex real-world setting. An example highlighting these improvements is shown in Fig. 6. Details are provided in App. B. Additional experimental results on the ESS evolution and runtime comparisons are provided in App. C.3.

## 5 CONCLUSION

We introduce DriftLite, a lightweight, training-free framework that resolves a critical trade-off in the inference-time scaling of pre-trained diffusion models. By identifying and exploiting a fundamental degree of freedom in the Fokker-Planck equation, DriftLite actively controls the sampling drift, thereby mitigating the weight degeneracy that plagues previous particle-based methods. Our practical instantiations, VCG and ECG, impose modest and scalable overhead while dramatically improving the stability and accuracy of inference-time scaling. Experiments further confirm their effectiveness and strong scaling with the number of particles, and we observe that the VCG variant is generally more robust, while the ECG holds promise in several specific scenarios. Across particle and protein systems, our approach consistently produces higher-quality samples and handles complex distributions more robustly compared to existing inference-time scaling baselines.

While DriftLite proves effective, its reliance on a fixed set of linear basis functions presents a potential limitation. Future work could explore more expressive yet still efficient representations for the control drift, such as compact neural networks or adaptive basis sets, including those involving the posterior mean (Chung et al., 2022). Our current instantiations also assume twice-differentiability of the reward $r$ and extending DriftLite to less smooth rewards is an open direction. We anticipate that future work will extend DriftLite to more complex physical systems, including LJ-55, ALDP, by pairing our inference-time scaling framework with continued advances in generative modeling. Furthermore, we have focused on annealing and reward-tilting tasks with non-heuristic targets and accuracy demands, but the DriftLite framework is broadly applicable beyond these tasks. Extending it to other generative problems, such as product-of-experts models or conditional generation, is a promising direction for future research. It is also interesting to study the inference-time scaling problem in the discrete settings under both the uniform (Campbell et al., 2022; Ren et al., 2024; 2025a) and the absorbing (Lou et al., 2023; Shi et al., 2024) frameworks.

ACKNOWLEDGMENTS

This work was conducted during Yinuo Ren's internship and visit at Center for Computational Mathematics, Flatiron Institute, an internal research division of the Simons Foundation. We thank the Scientific Computing Core, Flatiron Institute, for the computing facilities and support. This material is based upon work supported by the National Science Foundation under Award No. CHE-2441297 to Grant M. Rotskoff. Lexing Ying acknowledges support by the National Science Foundation under Award No. DMS-2208163.

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

## A  PROOFS

In this section, we present the omitted proofs of several propositions and additional discussions of the theoretical claims in the main content.

### A.1  PROOF OF PROPOSITION 2.1

To aid reading, we reiterate Prop. 2.1 below.

**Proposition A.1** (Guidance-Based Dynamics). *The exact time evolution of the density* $(q_t)_{t \in [0,T]}$ *follows the following Fokker-Planck equation:*

$$\partial_t q_t(\boldsymbol{x}) = -\nabla \cdot [\widetilde{\boldsymbol{v}}_t(\boldsymbol{x}) q_t(\boldsymbol{x})] + \frac{V_t^2}{2} \Delta q_t(\boldsymbol{x}) + q_t(\boldsymbol{x}) g_t(\boldsymbol{x}),$$

*where* $\widetilde{\boldsymbol{v}}_t$ *is given by*

$$\widetilde{\boldsymbol{v}}_t(\boldsymbol{x}) = -\overleftarrow{\boldsymbol{u}}_t(\boldsymbol{x}) + \frac{\overleftarrow{U}_t^2 + V_t^2}{2}(\gamma \nabla \log \overleftarrow{p}_t(\boldsymbol{x}) + \nabla r_t(\boldsymbol{x}))$$

$$= -(1-\gamma)\overleftarrow{\boldsymbol{u}}_t(\boldsymbol{x}) + \gamma \boldsymbol{v}_t(\boldsymbol{x}) + \frac{\overleftarrow{U}_t^2 + V_t^2}{2}\nabla r_t(\boldsymbol{x}),$$

*and the reweighting potential* $g_t(\boldsymbol{x}) = G_t(\boldsymbol{x}) - \mathbb{E}_{q_t}[G_t(\cdot)]$ *is given by:*

$$G_t = \dot{r}_t - (1-\gamma)\nabla \cdot \overleftarrow{\boldsymbol{u}}_t + \frac{\overleftarrow{U}_t^2}{2}\left(\Delta r_t - \gamma(1-\gamma)\|\nabla \log \overleftarrow{p}_t\|^2\right)$$

$$+ \nabla r_t^\top\left(-\overleftarrow{\boldsymbol{u}}_t + \gamma \overleftarrow{U}_t^2 \nabla \log \overleftarrow{p}_t + \frac{\overleftarrow{U}_t^2}{2}\nabla r_t\right).$$

*Proof.* We begin with the definition of the target density $q_t(\boldsymbol{x})$:

$$\log q_t(\boldsymbol{x}) = \gamma \log \overleftarrow{p}_t(\boldsymbol{x}) + r_t(\boldsymbol{x}) - Z_t,$$

where we define the log-partition function

$$Z_t = \log \int \overleftarrow{p}_t(\boldsymbol{y})^\gamma \exp(r_t(\boldsymbol{y})) d\boldsymbol{y},$$

and taking the time derivative gives:

$$\partial_t \log q_t(\boldsymbol{x}) = \gamma \partial_t \log \overleftarrow{p}_t(\boldsymbol{x}) + \dot{r}_t(\boldsymbol{x}) - \partial_t Z_t. \tag{A.1}$$

Recall that the Fokker-Planck equation for the backward process marginals $\overleftarrow{p}_t$ is as follows:

$$\partial_t \overleftarrow{p}_t(\boldsymbol{x}) = -\nabla \cdot (\boldsymbol{v}_t(\boldsymbol{x}) \overleftarrow{p}_t(\boldsymbol{x})) + \frac{V_t^2}{2}\Delta \overleftarrow{p}_t(\boldsymbol{x}),$$

which can be expressed in terms of the log-density as:

$$\partial_t \log \overleftarrow{p}_t(\boldsymbol{x}) = -\overleftarrow{p}_t^{-1}(\boldsymbol{x})\nabla \cdot (\boldsymbol{v}_t(\boldsymbol{x}) \overleftarrow{p}_t(\boldsymbol{x})) + \frac{V_t^2}{2}\overleftarrow{p}_t^{-1}(\boldsymbol{x})\Delta \overleftarrow{p}_t(\boldsymbol{x})$$

$$= -\nabla \cdot \boldsymbol{v}_t(\boldsymbol{x}) - \boldsymbol{v}_t(\boldsymbol{x}) \cdot \nabla \log \overleftarrow{p}_t(\boldsymbol{x}) + \frac{V_t^2}{2}\left(\Delta \log \overleftarrow{p}_t(\boldsymbol{x}) + \|\nabla \log \overleftarrow{p}_t(\boldsymbol{x})\|^2\right). \tag{A.2}$$

We posit that the time evolution of $q_t$ is governed by the Fokker-Planck equation with an additional reweighting term:

$$\partial_t q_t(\boldsymbol{x}) = -\nabla \cdot [\widetilde{\boldsymbol{v}}_t(\boldsymbol{x}) q_t(\boldsymbol{x})] + \frac{V_t^2}{2}\Delta q_t(\boldsymbol{x}) + q_t(\boldsymbol{x}) g_t(\boldsymbol{x}),$$

and dividing both sides by $q_t(\boldsymbol{x})$, we can solve for the potential $g_t(\boldsymbol{x})$:

$$g_t(\boldsymbol{x}) = \partial_t \log q_t(\boldsymbol{x}) + q_t^{-1}(\boldsymbol{x})\nabla \cdot [\widetilde{\boldsymbol{v}}_t(\boldsymbol{x}) q_t(\boldsymbol{x})] - \frac{V_t^2}{2}q_t^{-1}(\boldsymbol{x})\Delta q_t(\boldsymbol{x}).$$

Since we have

$$\int \partial_t q_t(\boldsymbol{x}) \mathrm{d}\boldsymbol{x} = \frac{\mathrm{d}}{\mathrm{d}t} \int q_t(\boldsymbol{x}) \mathrm{d}\boldsymbol{x} = 0,$$

and

$$\int \left( -\nabla \cdot [\widetilde{\boldsymbol{v}}_t(\boldsymbol{x}) q_t(\boldsymbol{x})] + \frac{V_t^2}{2} \Delta q_t(\boldsymbol{x}) \right) \mathrm{d}\boldsymbol{x} = 0$$

by applying the divergence theorem and assuming suitable boundary conditions, the reweighting term must have zero expectation, *i.e.*,

$$\mathbb{E}_{\boldsymbol{x} \sim q_t}[g_t(\boldsymbol{x})] = \int q_t(\boldsymbol{x}) g_t(\boldsymbol{x}) = 0.$$

Thus, we can write $g_t(\boldsymbol{x}) = G_t(\boldsymbol{x}) - \mathbb{E}_{q_t}[G_t(\cdot)]$, where we define $G_t(\boldsymbol{x})$ by absorbing the spatially independent term $\partial_t Z_t$:

$$G_t(\boldsymbol{x}) = g_t(\boldsymbol{x}) + \partial_t Z_t$$

$$= \partial_t \log q_t(\boldsymbol{x}) + \partial_t Z_t + q_t^{-1}(\boldsymbol{x}) \nabla \cdot [\widetilde{\boldsymbol{v}}_t(\boldsymbol{x}) q_t(\boldsymbol{x})] - \frac{V_t^2}{2} q_t^{-1}(\boldsymbol{x}) \Delta q_t(\boldsymbol{x})$$

$$= \dot{r}_t(\boldsymbol{x}) + \gamma \partial_t \log \overleftarrow{p}_t(\boldsymbol{x}) + q_t^{-1}(\boldsymbol{x}) \nabla \cdot [\widetilde{\boldsymbol{v}}_t(\boldsymbol{x}) q_t(\boldsymbol{x})] - \frac{V_t^2}{2} q_t^{-1}(\boldsymbol{x}) \Delta q_t(\boldsymbol{x})$$

$$= \dot{r}_t(\boldsymbol{x}) + \gamma \left( -\nabla \cdot \boldsymbol{v}_t(\boldsymbol{x}) - \boldsymbol{v}_t(\boldsymbol{x})^\top \nabla \log \overleftarrow{p}_t(\boldsymbol{x}) + \frac{V_t^2}{2} \left( \Delta \log \overleftarrow{p}_t(\boldsymbol{x}) + \|\nabla \log \overleftarrow{p}_t(\boldsymbol{x})\|^2 \right) \right)$$

$$+ \nabla \cdot \widetilde{\boldsymbol{v}}_t(\boldsymbol{x}) + \widetilde{\boldsymbol{v}}_t(\boldsymbol{x})^\top \nabla \log q_t(\boldsymbol{x}) - \frac{V_t^2}{2} \left( \Delta \log q_t(\boldsymbol{x}) + \|\nabla \log q_t(\boldsymbol{x})\|^2 \right)$$

$$= \dot{r}_t(\boldsymbol{x}) \underbrace{-\gamma \nabla \cdot \boldsymbol{v}_t(\boldsymbol{x}) + \nabla \cdot \widetilde{\boldsymbol{v}}_t(\boldsymbol{x})}_{\text{(I)}} \underbrace{-\gamma \boldsymbol{v}_t(\boldsymbol{x})^\top \nabla \log \overleftarrow{p}_t(\boldsymbol{x}) + \widetilde{\boldsymbol{v}}_t(\boldsymbol{x})^\top \nabla \log q_t(\boldsymbol{x})}_{\text{(II)}}$$

$$+ \underbrace{\frac{\gamma V_t^2}{2} \left( \Delta \log \overleftarrow{p}_t(\boldsymbol{x}) + \|\nabla \log \overleftarrow{p}_t(\boldsymbol{x})\|^2 \right) - \frac{V_t^2}{2} \left( \Delta \log q_t(\boldsymbol{x}) + \|\nabla \log q_t(\boldsymbol{x})\|^2 \right)}_{\text{(III)}},$$

where the second equality is due to Eqn. (A.1), and the second-to-last is due to Eqn. (A.1).

For compactness, write $s_t(\boldsymbol{x}) = \nabla \log \overleftarrow{p}_t(\boldsymbol{x})$ and $C_t = (\overleftarrow{U}_t^2 + V_t^2)/2$. We now substitute the expressions for $\partial_t \log \overleftarrow{p}_t$ and $\log q_t$ and simplify term by term.

- *(I) Divergence Terms:*

$$\text{(I)} = -\gamma \nabla \cdot \boldsymbol{v}_t(\boldsymbol{x}) + \nabla \cdot \widetilde{\boldsymbol{v}}_t(\boldsymbol{x}) = \nabla \cdot (\widetilde{\boldsymbol{v}}_t(\boldsymbol{x}) - \gamma \boldsymbol{v}_t(\boldsymbol{x}))$$
$$= \nabla \cdot (-(1 - \gamma) \overleftarrow{\boldsymbol{u}}_t(\boldsymbol{x}) + C_t \nabla r_t(\boldsymbol{x})) = -(1 - \gamma) \nabla \cdot \overleftarrow{\boldsymbol{u}}_t(\boldsymbol{x}) + C_t \Delta r_t(\boldsymbol{x}).$$

- *(II) Inner Product Terms:*

$$\text{(II)} = -\gamma \boldsymbol{v}_t(\boldsymbol{x})^\top s_t(\boldsymbol{x}) + \widetilde{\boldsymbol{v}}_t(\boldsymbol{x})^\top \nabla \log q_t(\boldsymbol{x})$$
$$= -\gamma \boldsymbol{v}_t(\boldsymbol{x})^\top s_t(\boldsymbol{x}) + \widetilde{\boldsymbol{v}}_t(\boldsymbol{x})^\top (\gamma s_t(\boldsymbol{x}) + \nabla r_t(\boldsymbol{x}))$$
$$= \gamma (\widetilde{\boldsymbol{v}}_t(\boldsymbol{x}) - \boldsymbol{v}_t(\boldsymbol{x}))^\top s_t(\boldsymbol{x}) + \widetilde{\boldsymbol{v}}_t(\boldsymbol{x})^\top \nabla r_t(\boldsymbol{x})$$
$$= \gamma C_t ((\gamma - 1) s_t(\boldsymbol{x}) + \nabla r_t(\boldsymbol{x}))^\top s_t(\boldsymbol{x}) + (-\overleftarrow{\boldsymbol{u}}_t(\boldsymbol{x}) + C_t(\gamma s_t(\boldsymbol{x}) + \nabla r_t(\boldsymbol{x})))^\top \nabla r_t(\boldsymbol{x})$$
$$= \nabla r_t(\boldsymbol{x})^\top (-\overleftarrow{\boldsymbol{u}}_t(\boldsymbol{x}) + 2\gamma C_t s_t(\boldsymbol{x}) + C_t \nabla r_t(\boldsymbol{x})) - C_t \gamma (1 - \gamma) \|s_t(\boldsymbol{x})\|^2.$$

- *(III) Laplacian Terms:*

$$\text{(III)} = \frac{\gamma V_t^2}{2} \left( \Delta \log \overleftarrow{p}_t(\boldsymbol{x}) + \|s_t(\boldsymbol{x})\|^2 \right) - \frac{V_t^2}{2} \left( \Delta \log q_t(\boldsymbol{x}) + \|\nabla \log q_t(\boldsymbol{x})\|^2 \right)$$
$$= \frac{V_t^2}{2} \left( \gamma \Delta \log \overleftarrow{p}_t(\boldsymbol{x}) - \Delta(\gamma \log \overleftarrow{p}_t(\boldsymbol{x}) + r_t(\boldsymbol{x})) + \gamma \|s_t(\boldsymbol{x})\|^2 - \|\gamma s_t(\boldsymbol{x}) + \nabla r_t(\boldsymbol{x})\|^2 \right)$$
$$= -\frac{V_t^2}{2} \Delta r_t(\boldsymbol{x}) - \frac{V_t^2}{2} \left( \gamma(\gamma - 1) \|s_t(\boldsymbol{x})\|^2 + 2\gamma s_t(\boldsymbol{x}) \cdot \nabla r_t(\boldsymbol{x}) + \|\nabla r_t(\boldsymbol{x})\|^2 \right).$$

Combining (I)–(III) gives the stated expression: the $\Delta r_t$, $\|s_t\|^2$, $s_t \cdot \nabla r_t$, and $\|\nabla r_t\|^2$ coefficients reduce respectively to $\overleftarrow{U}_t^2/2$, $-\overleftarrow{U}_t^2\gamma(1-\gamma)/2$, $\gamma\overleftarrow{U}_t^2$, and $\overleftarrow{U}_t^2/2$. Hence

$$G_t = \dot{r}_t - (1-\gamma)\nabla \cdot \overleftarrow{\boldsymbol{u}}_t + \frac{\overleftarrow{U}_t^2}{2}\left(\Delta r_t - \gamma(1-\gamma)\|s_t\|^2\right)$$

$$+ \nabla r_t^\top \left(-\overleftarrow{\boldsymbol{u}}_t + \gamma\overleftarrow{U}_t^2 s_t + \frac{\overleftarrow{U}_t^2}{2}\nabla r_t\right).$$

This completes the proof. $\square$

**Remark A.2.** *This proof is similar to the proof in related works (Skreta et al., 2025; Chen et al., 2025), where Fokker-Planck equations are derived for each task-specific setting. While being more general, our approach also omits the computation of the time derivative of the log-partition function $\partial_t Z_t$, since we notice that it cancels out in the final expression for $G_t$. This simplification makes the proof more straightforward and concise.*

## A.2 JUSTIFICATION OF WEIGHTED PARTICLE SYSTEM (2.6)

**Proposition A.3** (Weighted Particle Simulation). *Let $q_t : \mathbb{R}^d \to \mathbb{R}_{\geq 0}$ solve*

$$\partial_t q_t(\boldsymbol{x}) = -\nabla \cdot \left[\widetilde{\boldsymbol{v}}_t(\boldsymbol{x})q_t(\boldsymbol{x})\right] + \frac{V_t^2}{2}\Delta q_t(\boldsymbol{x}) + q_t(\boldsymbol{x})g_t(\boldsymbol{x}).$$

*Then this PDE can be simulated in the mean-field limit $N \to \infty$ by the weighted particle system*

$$\begin{cases} \mathrm{d}\boldsymbol{x}_t^{(i)} = \widetilde{\boldsymbol{v}}_t(\boldsymbol{x}_t^{(i)})\mathrm{d}t + V_t\mathrm{d}\boldsymbol{w}_t^{(i)}, & i \in [N], \\ \mathrm{d}\log w_t^{(i)} = \widehat{g}_t(\boldsymbol{x}_t^{(i)})\mathrm{d}t, & i \in [N], \end{cases}$$

*where the weights satisfy the normalization $\sum_{i=1}^N w_t^{(i)} = 1$, and the empirical centered version of $g_t$:*

$$\widehat{g}_t(\boldsymbol{x}) = G_t(\boldsymbol{x}) - \sum_{j=1}^N w_t^{(j)}G_t(\boldsymbol{x}_t^{(j)})$$

*ensures mass preservation.*

The proof of this argument is standard (Moral, 2004; Doucet et al., 2000; Andrieu et al., 2010) under typical assumptions, including local Lipschitz continuity and linear growth of the drift $\widetilde{\boldsymbol{v}}_t$, boundedness of the diffusion coefficient $V_t$, moderate growth of $\widehat{g}_t$, sufficient regularity of $q_t$ to justify integration by parts, and either fast decay at infinity or no-flux boundary conditions to eliminate boundary terms. We provide a proof sketch below for the reader's convenience. We also point out that similar arguments also apply to the weighted particle simulation for the controlled dynamics in Prop. 3.1, which we omit for simplicity.

*Proof Sketch.* The main steps are as follows:

- *Step 1 (Empirical measure).* Define the weighted empirical distribution

$$\mu_t^N = \sum_{i=1}^N w_t^{(i)}\delta_{\boldsymbol{x}_t^{(i)}}.$$

- *Step 2 (Test function evolution).* For $\varphi \in C_c^\infty(\mathbb{R}^d)$, consider

$$\langle \varphi, \mu_t^N \rangle = \sum_{i=1}^N w_t^{(i)}\varphi(\boldsymbol{x}_t^{(i)}).$$

Apply Itô's lemma to $\varphi(\boldsymbol{x}_t^{(i)})$ under the SDE and combine with the weight dynamics via the product rule. This yields

$$\mathrm{d}\langle\varphi,\mu_t^N\rangle = \left\langle \widetilde{\boldsymbol{v}}_t^\top\nabla\varphi + \frac{V_t^2}{2}\Delta\varphi + \widehat{g}_t\varphi, \mu_t^N \right\rangle\mathrm{d}t + \mathrm{d}M_t^N,$$

where $M_t^N$ is a martingale term.

- *Step 3 (Limit $N\to\infty$).* By law of large numbers and propagation of chaos (Sznitman, 1991; Lacker, 2018), we have the weak convergence:

$$\mu_t^N \Rightarrow q_t(\boldsymbol{x})\mathrm{d}\boldsymbol{x}, \quad \text{as } N\to\infty,$$

while the martingale term vanishes. Passing to the limit gives the weak form of the PDE:

$$\frac{\mathrm{d}}{\mathrm{d}t}\int\varphi(\boldsymbol{x})q_t(\boldsymbol{x})\mathrm{d}\boldsymbol{x} = \int\left[\widetilde{\boldsymbol{v}}_t(\boldsymbol{x})^\top\nabla\varphi(\boldsymbol{x}) + \frac{V_t^2}{2}\Delta\varphi(\boldsymbol{x}) + \widehat{g}_t(\boldsymbol{x})\varphi(\boldsymbol{x})\right]q_t(\boldsymbol{x})\mathrm{d}\boldsymbol{x}.$$

- *Step 4 (Integration by parts).* Using divergence theorem identities, we have

$$\frac{\mathrm{d}}{\mathrm{d}t}\int\varphi(\boldsymbol{x})q_t(\boldsymbol{x})\mathrm{d}\boldsymbol{x} = \int\varphi(\boldsymbol{x})\left[-\nabla\cdot\left(\widetilde{\boldsymbol{v}}_t(\boldsymbol{x})q_t(\boldsymbol{x})\right) + \frac{V_t^2}{2}\Delta q_t(\boldsymbol{x}) + \widehat{g}_t(\boldsymbol{x})q_t(\boldsymbol{x})\right]\mathrm{d}\boldsymbol{x},$$

for all test functions $\varphi$.

Hence, we have

$$\partial_t q_t(\boldsymbol{x}) = -\nabla\cdot\left(\widetilde{\boldsymbol{v}}_t(\boldsymbol{x})q_t(\boldsymbol{x})\right) + \frac{V_t^2}{2}\Delta q_t(\boldsymbol{x}) + \widehat{g}_t(\boldsymbol{x})q_t(\boldsymbol{x}),$$

and the proof is complete. $\qquad\square$

### A.3 PROOF OF PROPOSITION 3.1

The following proposition is the same as Prop. 3.1, but with a more detailed proof.

---

**Proposition A.4** (Degree of Freedom). *For any control drift $\boldsymbol{b}_t(\boldsymbol{x})$, the Fokker-Planck equation*

$$\partial_t q_t(\boldsymbol{x}) = -\nabla\cdot[\widetilde{\boldsymbol{v}}_t(\boldsymbol{x})q_t(\boldsymbol{x})] + \frac{V_t^2}{2}\Delta q_t(\boldsymbol{x}) + q_t(\boldsymbol{x})g_t(\boldsymbol{x}),$$

*is equivalent to the following one with an additional control drift term $\boldsymbol{b}_t(\boldsymbol{x})$:*

$$\partial_t q_t(\boldsymbol{x}) = -\nabla\cdot\left[\left(\widetilde{\boldsymbol{v}}_t(\boldsymbol{x}) + \boldsymbol{b}_t(\boldsymbol{x})\right)q_t(\boldsymbol{x})\right] + \frac{V_t^2}{2}\Delta q_t(\boldsymbol{x}) + q_t(\boldsymbol{x})\phi_t(\boldsymbol{x}),$$

*where the residual potential is $\phi_t(\boldsymbol{x}) = g_t(\boldsymbol{x}) + h_t(\boldsymbol{x};\boldsymbol{b}_t)$ with the control potential $h_t(\cdot;\boldsymbol{b}_t)$ defined as:*

$$h_t(\boldsymbol{x};\boldsymbol{b}_t) = (\gamma\nabla\log\overleftarrow{p}_t(\boldsymbol{x}) + \nabla r_t(\boldsymbol{x}))\cdot\boldsymbol{b}_t(\boldsymbol{x}) + \nabla\cdot\boldsymbol{b}_t(\boldsymbol{x}).$$

---

*Proof.* The terms added to the right-hand side of Eqn. (2.5) to obtain Eqn. (3.1) are:

$$-\nabla\cdot(\boldsymbol{b}_t(\boldsymbol{x})q_t(\boldsymbol{x})) + q_t(\boldsymbol{x})(h_t(\boldsymbol{x};\boldsymbol{b}_t) - \mathbb{E}_{q_t}[h_t(\boldsymbol{x};\boldsymbol{b}_t)]).$$

First, we prove that $\mathbb{E}_{q_t}[h_t(\boldsymbol{x};\boldsymbol{b}_t)] = 0$:

$$\mathbb{E}_{q_t}[h_t(\boldsymbol{x};\boldsymbol{b}_t)] = \int q_t(\boldsymbol{x})\left[(\nabla r_t(\boldsymbol{x}) + \gamma\nabla\log\overleftarrow{p}_t(\boldsymbol{x}))\cdot\boldsymbol{b}_t(\boldsymbol{x}) + \nabla\cdot\boldsymbol{b}_t(\boldsymbol{x})\right]\mathrm{d}\boldsymbol{x}$$

$$= \int\nabla\cdot(q_t(\boldsymbol{x})\boldsymbol{b}_t(\boldsymbol{x}))\mathrm{d}\boldsymbol{x} = 0,$$

where the last equality follows from the divergence theorem, assuming appropriate boundary conditions (*e.g.*, $q_t\boldsymbol{b}_t$ vanishes at infinity).

Then, we show that the remaining added terms cancel each other out:

$$- \nabla \cdot (\boldsymbol{b}_t(\boldsymbol{x})q_t(\boldsymbol{x})) + q_t(\boldsymbol{x})h_t(\boldsymbol{x}; \boldsymbol{b}_t)$$
$$= - \nabla \cdot (\boldsymbol{b}_t(\boldsymbol{x})q_t(\boldsymbol{x})) + q_t(\boldsymbol{x})[\nabla \log q_t(\boldsymbol{x}) \cdot \boldsymbol{b}_t(\boldsymbol{x}) + \nabla \cdot \boldsymbol{b}_t(\boldsymbol{x})]$$
$$= - \nabla \cdot (\boldsymbol{b}_t(\boldsymbol{x})q_t(\boldsymbol{x})) + \nabla \cdot (q_t(\boldsymbol{x})\boldsymbol{b}_t(\boldsymbol{x})) = 0.$$

Since the added terms sum to zero (given $\mathbb{E}_{q_t}[h_t(\boldsymbol{x}; \boldsymbol{b}_t)] = 0$), the two PDEs are equivalent. $\qquad\square$

## A.4 FORMAL SOLUTION FOR THE OPTIMAL CONTROL DRIFT

In Prop. 3.2, we claim that there exists a uniform optimal control drift as we rewrite its mathematical rigorous version in the following proposition.

---

**Proposition A.5** (Optimal Control). *Let $\Omega \subset \mathbb{R}^d$ be a bounded Lipschitz domain. Assume that $q_t : \Omega \to \mathbb{R}$ is measurable and uniformly elliptic: there exist constants $0 < \lambda \le \Lambda < \infty$ such that*

$$\lambda \le q_t(x) \le \Lambda, \qquad \text{for a.e. } x \in \Omega,$$

*and suppose $g_t \in L^2(\Omega)$. Then there exists a unique potential $A_t^* \in H_0^1(\Omega)$ solving*

$$-\nabla \cdot (q_t(x)\nabla A_t^*(x)) = q_t(x)g_t(x) \quad \text{in } H^{-1}(\Omega).$$

*Defining the control $\boldsymbol{b}_t^*(x) := \nabla A_t^*(x) \in L^2(\Omega; \mathbb{R}^d)$, one has*

$$\phi_t^*(x) = g_t(x) + h_t(x; \boldsymbol{b}_t^*) = g_t(x) + \frac{1}{q_t(x)}\nabla \cdot (q_t(x)\boldsymbol{b}_t^*(x)) = 0 \quad \text{in } H^{-1}(\Omega).$$

*In particular, $\boldsymbol{b}_t^*$ is the unique curl-free control that drives $\phi_t$ to zero.*

---

*Proof.* This follows directly from the Lax–Milgram theorem. The bilinear form

$$a(u, v) := \int_\Omega q_t \nabla u \cdot \nabla v \, dx, \qquad u, v \in H_0^1(\Omega),$$

is bounded and coercive, while the linear functional

$$L(v) := \int_\Omega q_t g_t v \, dx$$

is continuous on $H_0^1(\Omega)$. By the Lax–Milgram theorem, there exists a unique $A_t^* \in H_0^1(\Omega)$ satisfying (3.2). The existence and uniqueness of weak solutions to such elliptic PDEs are standard results; see, for example, Evans (2022, Chapter 6). $\qquad\square$

While the existence and the uniqueness of the solution to the Poisson equation (3.2) are shown above, we present a formal solution for the control drift $\boldsymbol{b}_t^*$ in the following, leading to our choice of basis functions in Sec. 3.3.

Let $\boldsymbol{f}_t(\boldsymbol{x}) = q_t(\boldsymbol{x})\boldsymbol{b}_t^*(\boldsymbol{x})$. By the Helmholtz-Hodge theorem, any sufficiently smooth vector field $\boldsymbol{f}_t$ can be decomposed into a curl-free component (the gradient of a scalar potential $A$) and a divergence-free component $\boldsymbol{u}_\perp$, so that

$$\boldsymbol{f}_t(\boldsymbol{x}) = \nabla A(\boldsymbol{x}) + \boldsymbol{u}_\perp(\boldsymbol{x}),$$

where $\nabla \cdot \boldsymbol{u}_\perp \equiv 0$.

Substituting this decomposition into the equation gives:

$$\nabla \cdot (\nabla A(\boldsymbol{x}) + \boldsymbol{u}_\perp(\boldsymbol{x})) = \Delta A(\boldsymbol{x}) = -q_t(\boldsymbol{x})g_t(\boldsymbol{x}).$$

This is a standard Poisson equation for the scalar potential $A$. The solution for $A$ can be expressed formally using the Green's function for the Laplacian in $d$ dimensions, $G_d(\boldsymbol{x} - \boldsymbol{y})$:

$$A(\boldsymbol{x}) = -\int G_d(\boldsymbol{x} - \boldsymbol{y})q_t(\boldsymbol{y})g_t(\boldsymbol{y})\mathrm{d}\boldsymbol{y}.$$

The desired control is then given by $\boldsymbol{b}_t(\boldsymbol{x}) = q_t(\boldsymbol{x})^{-1}\nabla A(\boldsymbol{x})$. Taking the gradient of $A(\boldsymbol{x})$ with respect to $\boldsymbol{x}$ and using integration by parts with the property $\nabla_{\boldsymbol{x}} G_d(\boldsymbol{x} - \boldsymbol{y}) = -\nabla_{\boldsymbol{y}} G_d(\boldsymbol{x} - \boldsymbol{y})$, we get the following formal solution:

$$\nabla A(\boldsymbol{x}) = -\int \nabla_{\boldsymbol{x}} G_d(\boldsymbol{x} - \boldsymbol{y}) q_t(\boldsymbol{y}) g_t(\boldsymbol{y}) \mathrm{d}\boldsymbol{y}$$

$$= \int G_d(\boldsymbol{x} - \boldsymbol{y}) \nabla_{\boldsymbol{y}} \left( q_t(\boldsymbol{y}) g_t(\boldsymbol{y}) \right) \mathrm{d}\boldsymbol{y}$$

$$= \int G_d(\boldsymbol{x} - \boldsymbol{y}) \left( g_t(\boldsymbol{y}) \nabla_{\boldsymbol{y}} \log q_t(\boldsymbol{y}) + \nabla_{\boldsymbol{y}} g_t(\boldsymbol{y}) \right) q_t(\boldsymbol{y}) \mathrm{d}\boldsymbol{y},$$

where the term $\nabla g_t$ involves derivatives of the corrected uncentered potential $G_t$ from Prop. 2.1 (centering only changes $g_t$ by a spatial constant). In particular, for reward-tilting it contains local quantities such as $\nabla r_t$, $\nabla^2 r_t$, $\nabla \Delta r_t$, the score $\nabla \log \overleftarrow{p}_t$, score Jacobian terms, and derivatives of the forward drift $\overleftarrow{\boldsymbol{u}}_t$. This formal solution is computationally infeasible as it requires integrating over all space $\boldsymbol{y}$, weighted by the unknown density $q_t(\boldsymbol{y})$ that we are trying to simulate. However, its structure motivates our choice of local basis functions: the reward gradient $\nabla r_t(\boldsymbol{x})$, the score $\nabla \log \overleftarrow{p}_t(\boldsymbol{x})$, and the forward drift $\overleftarrow{\boldsymbol{u}}_t(\boldsymbol{x})$, while discarding higher-order derivative terms.

## B  ADDITIONAL IMPLEMENTATION DETAILS

In this section, we provide additional implementation details for the experiments, including the problem setup with parameters, a detailed description of the network architecture, training, inference, and evaluation procedures.

### B.1  PROBLEM SETTINGS

**Gaussian Mixture Model (GMM).**   In the Gaussian Mixture Model (GMM) task, the data distribution is set as

$$p_0(\boldsymbol{x}) = \frac{1}{40} \sum_{i=1}^{40} \mathcal{N}\left(\boldsymbol{x}; \boldsymbol{\mu}_i, 50\boldsymbol{I}\right),$$

where each component mean $\boldsymbol{\mu}_i$ is sampled from $\mathrm{Unif}([-40, 40])$.

For the annealing task where the target distribution is $q_T(\boldsymbol{x}) \propto p_0(\boldsymbol{x})^\gamma$, the reference samples are obtained by rejection sampling with the following proposal distribution:

$$q_0^{\mathrm{proposal}}(\boldsymbol{x}) = \frac{1}{40} \sum_{i=1}^{40} \mathcal{N}\left(\boldsymbol{x}; \boldsymbol{\mu}_i, \frac{50}{\gamma}\boldsymbol{I}\right).$$

For the reward-tilting task, the quadratic reward function is given by

$$r(\boldsymbol{x}) = -\frac{1}{2}(\boldsymbol{x} - \boldsymbol{\mu})^\top \Sigma^{-1}(\boldsymbol{x} - \boldsymbol{\mu}),$$

where the target mean $\boldsymbol{\mu}$ is sampled from $\mathcal{N}(\boldsymbol{\mu}; \boldsymbol{0}, 100\boldsymbol{I})$ and the covariance matrix $\Sigma = \sigma\boldsymbol{I}$ with $\sigma$ being a parameter controlling the spread of the reward. The posterior distribution corresponds to another Gaussian mixture model, in which each component acquires an updated mean and weight after incorporating the quadratic reward:

$$q_0(\boldsymbol{x}) = \sum_{i=1}^{40} \widetilde{w}_i \mathcal{N}\left(\boldsymbol{x}; \widetilde{\boldsymbol{\mu}}_i, \widetilde{\Sigma}\right),$$

with posterior covariance

$$\widetilde{\Sigma} = \left(\Sigma^{-1} + (50\boldsymbol{I})^{-1}\right)^{-1},$$

posterior means

$$\widetilde{\boldsymbol{\mu}}_i = \widetilde{\Sigma}\left((50\boldsymbol{I})^{-1}\boldsymbol{\mu}_i + \Sigma^{-1}\boldsymbol{\mu}\right),$$

and mixture weights reweighted according to the evidence,

$$\widetilde{w}_i \ \propto \ w_i \exp\left(-\tfrac{1}{2}(\boldsymbol{\mu}_i - \boldsymbol{\mu})^\top (\Sigma + 50\boldsymbol{I})^{-1}(\boldsymbol{\mu}_i - \boldsymbol{\mu})\right).$$

For all GMM experiments presented in Sec. 4.1, we set the number of particles $N = 2^{13}$ and perform resampling whenever ESS drops below 0.9. All plots in the GMM experiments are plotted by projecting onto the first two dimensions.

**Double-Well-4 (DW-4).**  For both DW-4 and LJ-13 systems, the target is a Boltzmann distribution of the following form:

$$p_0(\boldsymbol{x}) \propto \exp(-\mathcal{E}(\boldsymbol{x})) = \exp\left(-\frac{1}{T}\left(H(\boldsymbol{x}) + \frac{\lambda}{2}\sum_{i=1}^{n}\|\boldsymbol{r}_i - \overline{\boldsymbol{r}}\|^2\right)\right), \tag{B.1}$$

with the potential $H(\boldsymbol{x})$ system-specific, and a harmonic potential of strength $\lambda$ imposed as a physical constraint.

In the reward-tilting task, we consider the quadratic reward:

$$r(\boldsymbol{x}) = -\frac{\lambda'}{2}\sum_{i=1}^{n}\|\boldsymbol{r}_i - \overline{\boldsymbol{r}}\|^2,$$

and thus the reward-tilted distribution is

$$p_0(\boldsymbol{x}) \propto \exp\left(-\frac{1}{T}\left(H(\boldsymbol{x}) + \frac{\lambda}{2}\sum_{i=1}^{n}\|\boldsymbol{r}_i - \overline{\boldsymbol{r}}\|^2\right) - \frac{\lambda'}{2}\sum_{i=1}^{n}\|\boldsymbol{r}_i - \overline{\boldsymbol{r}}\|^2\right)$$

$$= \exp\left(-\frac{1}{T}\left(H(\boldsymbol{x}) + \frac{\lambda + \lambda'T}{2}\sum_{i=1}^{n}\|\boldsymbol{r}_i - \overline{\boldsymbol{r}}\|^2\right)\right),$$

*i.e.*, another Boltzmann distribution of the same temperature but with a different constraint strength $\lambda + \lambda'T$. This reward thus corresponds physically to strengthening the harmonic confinement around the center of mass, *i.e.*, sampling the same particle system at the same temperature but with an increased trap stiffness.

First introduced by (Köhler et al., 2020), the double-well potential is defined on a system of 4 particles in the two-dimensional space ($\boldsymbol{x} \in \mathbb{R}^{4\times2}$). The potential energy function is given as:

$$H_{\text{DW}}(\boldsymbol{x}) = \frac{1}{2}\sum_{i<j}\left[a(d_{ij} - d_0) + b(d_{ij} - d_0)^2 + c(d_{ij} - d_0)^4\right],$$

where $\boldsymbol{r}_i$ is the coordinate of particle $i$ and $d_{ij} = \|\boldsymbol{r}_i - \boldsymbol{r}_j\|$ denotes the distance between particles $i$ and $j$. We use standard parameters: $a = 0.0$, $b = -4.0$, $c = 0.9$, and $d_0 = 4.0$. This would yield a double-well potential with two minima at $d_{ij} = 4 \pm \frac{2}{3}\sqrt{5}$. For the harmonic potential constraint in the Boltzmann distribution Eqn. (B.1), we set the constraint strength as $\lambda = 0.05$. For all DW-4 experiments presented in Sec. 4.2, we set the number of particles $N = 2^{15}$ and perform resampling every $K = 100$ steps.

**Lennard-Jones-13 (LJ-13).**  The Lennard-Jones potential is a widely used model for simulating interatomic interactions with both repulsive and attractive components. In our case, it is defined on a system of 13 particles in three-dimensional space ($\boldsymbol{x} \in \mathbb{R}^{13\times3}$). The potential energy function is given as:

$$H_{\text{LJ}}(\boldsymbol{x}) = \frac{\epsilon}{2T}\sum_{i<j}\left[\left(\frac{\sigma}{d_{ij}}\right)^{12} - 2\left(\frac{\sigma}{d_{ij}}\right)^{6}\right].$$

We use the following parameters: $\sigma = 1.0$ and $\epsilon = 2.0$. For the harmonic potential constraint in the Boltzmann distribution Eqn. (B.1), we set the constraint strength as $\lambda = 1.0$. For all LJ-13 experiments presented in Sec. 4.2, we set the number of particles $N = 2^{13}$ and perform resampling every $K = 50$ steps.

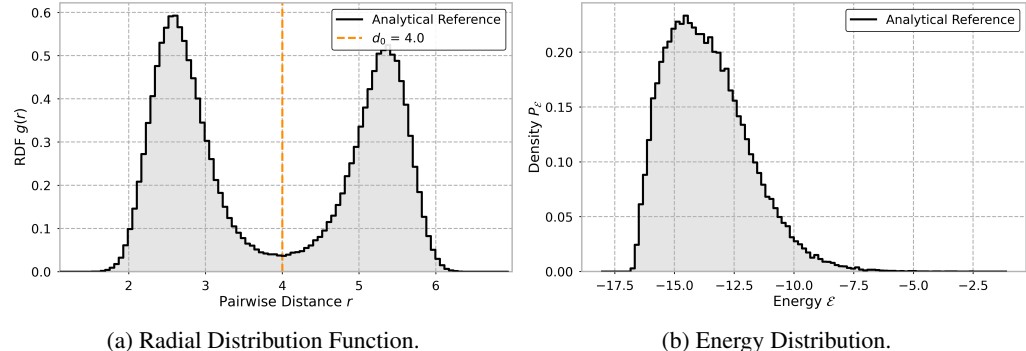

(a) Radial Distribution Function.

(b) Energy Distribution.

Figure 7: Reference distributions for the DW-4 system at the base temperature $T = 1.5$: (a) Radial Distribution Function (RDF) and (b) Energy Distribution.

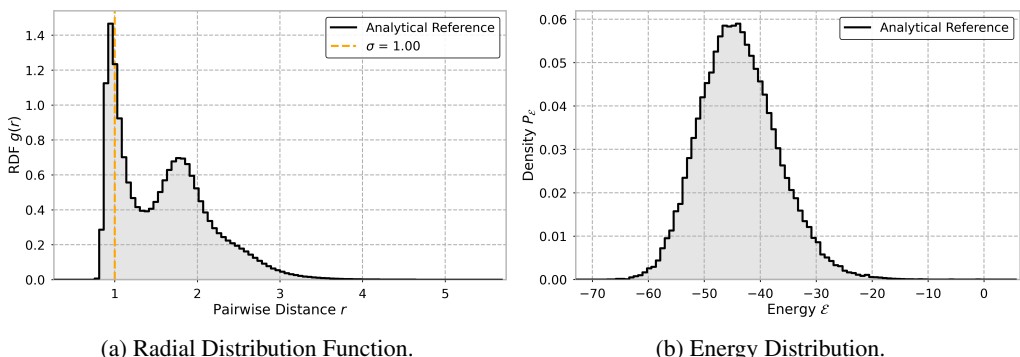

(a) Radial Distribution Function.

(b) Energy Distribution.

Figure 8: Reference distributions for the LJ-13 system at the base temperature $T = 1.0$: (a) Radial Distribution Function (RDF) and (b) Energy Distribution.

**Protein-ligand Co-folding.** The protein-ligand co-folding problem is an extension of the classical protein folding problem: rather than predicting only the structure of the protein itself, the task is to simultaneously predict the structure of its interacting counterpart, a ligand that is typically a small molecule. This problem lies at the core of structural biology and is essential for understanding how proteins and ligands interact, which in turn underpins the elucidation of biological pathways and the design of new drug molecules to modulate biological activity. In this work, we focus on addressing the issue of physical validity in protein-ligand co-folding. Diffusion-based models often overemphasize global structural similarity while occasionally generating unphysical conformations. To mitigate this issue, we incorporate physics-based potentials to steer the generative process, effectively modifying the data distribution $p_\theta(\boldsymbol{x})$ with a physics-based potential function:

$$p_{\text{target}}(\boldsymbol{x}) \ \propto \ p_\theta(\boldsymbol{x}) \exp\big(r(\boldsymbol{x})\big).$$

Following Passaro et al. (2025), we use a physics reward that penalizes local constraint violations defined in the PoseBuster V2 benchmark. Let $\boldsymbol{x}$ denote all atomic coordinates, and let $\mathcal{E}_{\text{phys}}(\boldsymbol{x})$ be a weighted sum of potentials:

$$\mathcal{E}_{\text{phys}}(\boldsymbol{x}) \ = \ \alpha_{\text{bond}}\, U_{\text{bond}}(\boldsymbol{x}) + \alpha_{\text{angle}}\, U_{\text{angle}}(\boldsymbol{x}) + \alpha_{\text{chir}}\, U_{\text{chir}}(\boldsymbol{x}) + \alpha_{\text{int}}\, U_{\text{int-clash}}(\boldsymbol{x}) + \alpha_{\text{chain}}\, U_{\text{chain-clash}}(\boldsymbol{x}),$$

with nonnegative weights $\alpha_\bullet$. The steering reward is then defined as

$$r(\boldsymbol{x}) = -\lambda \mathcal{E}_{\text{phys}}(\boldsymbol{x}).$$

For each covalent bond, bond angle, and chiral center, we apply a flat-bottom potential function that penalizes deviations from the corresponding physical rules while permitting small fluctuations within an acceptable tolerance. This ensures that generated structures remain physically plausible without being overly constrained. Further implementation details of these potentials are provided in

Passaro et al. (2025). Since we also adopt the pre-trained model weights from Passaro et al. (2025), we do not describe the model architecture or training process in details here for which we refer readers to the original work. In this experiment, we use the reward gradient $\nabla r(\boldsymbol{x})$ as a single basis, and its Laplacian $\Delta r(\boldsymbol{x})$ is obtained through auto-differentiation.

## B.2 Network Architectures

The score function $\nabla \log \overleftarrow{p}_t$ in both particle experiments (DW-4 and LJ-13) is approximated by an $E(n)$ Equivariant Graph Neural Network (EGNN) (Satorras et al., 2021b;a; Köhler et al., 2020; Klein et al., 2023; Midgley et al., 2023). The network maps a time scalar $t$ and particle coordinates $\boldsymbol{x} = \{\boldsymbol{r}_1, \ldots, \boldsymbol{r}_n\}$ to an updated set of coordinates representing the score. All MLPs use Swish activations.

**Initialization.** Input coordinates are first centered. The scalar time $t$ is encoded using sinusoidal positional embeddings $\phi_t$. The resulting vector is then passed through an MLP to create the initial node features $\boldsymbol{h}_i^{(0)}$, which are identical for all nodes.

$$\boldsymbol{r}_i^{(0)} = \boldsymbol{r}_i - \frac{1}{n} \sum_{j=1}^{n} \boldsymbol{r}_j, \quad \boldsymbol{h}_i^{(0)} = \text{MLP}\left(\phi_t(t)\right).$$

**Equivariant Layers.** The network consists of $L$ sequential Equivariant Graph Convolutional Layers (EGCL). For each layer $l \in \{0, \ldots, L-1\}$, an initial message $\boldsymbol{m}_{ij}^{(l)}$ is computed for each edge $(i, j)$ using an MLP $\phi_e^{(l)}$:

$$\boldsymbol{m}_{ij}^{(l)} = \phi_e^{(l)}\left(\left[\boldsymbol{h}_i^{(l)}, \boldsymbol{h}_j^{(l)}, \|\boldsymbol{r}_i^{(l)} - \boldsymbol{r}_j^{(l)}\|^2\right]\right)$$

where $[\cdots]$ denotes concatenation. An attention mechanism then refines each message by multiplying it with a learned gating coefficient:

$$\boldsymbol{m}_{ij}^{(l)} \leftarrow \boldsymbol{m}_{ij}^{(l)} \cdot \sigma\left(\phi_{\text{att}}^{(l)}(\boldsymbol{m}_{ij}^{(l)})\right)$$

where $\phi_{\text{att}}^{(l)}$ is a single-layer MLP and $\sigma$ is the sigmoid function. These refined messages are used to produce equivariant updates for the coordinates and invariant updates for the node features via MLPs $\phi_x^{(l)}$ and $\phi_h^{(l)}$. Both updates employ residual connections.

$$\begin{cases} \boldsymbol{r}_i^{(l+1)} &= \boldsymbol{r}_i^{(l)} + \sum_{j \neq i} \dfrac{\boldsymbol{r}_i^{(l)} - \boldsymbol{r}_j^{(l)}}{\|\boldsymbol{r}_i^{(l)} - \boldsymbol{r}_j^{(l)}\| + C} \phi_x^{(l)}(\boldsymbol{m}_{ij}^{(l)}), \\ \boldsymbol{h}_i^{(l+1)} &= \boldsymbol{h}_i^{(l)} + \phi_h^{(l)}\left(\left[\boldsymbol{h}_i^{(l)}, \sum_{j \neq i} \boldsymbol{m}_{ij}^{(l)}\right]\right), \end{cases}$$

where $C$ is a normalization constant for the coordinate update.

**Final Output.** The final output of the network is the set of coordinates from the last layer, $\boldsymbol{r}^{(L)}$, which is re-centered to guarantee overall translation invariance. For each particle, the output is given by centering the coordinates:

$$\boldsymbol{r}_i^{\text{out}} = \boldsymbol{r}_i^{(L)} - \frac{1}{n} \sum_{j=1}^{n} \boldsymbol{r}_j^{(L)}.$$

For the DW-4 experiment, we use $L = 5$ layers with a hidden dimension of 64, and for the LJ-13 experiment, we use $L = 5$ layers with a hidden dimension of 128. The normalization constant $C$ is set to 1.0 for both experiments.

## B.3 Sampling Details

To generate configuration samples, we simulate the underdamped Langevin dynamics for a system of $N$ particles in $D$ dimensions. The dynamics target a canonical Boltzmann distribution:

$$\pi(\boldsymbol{x}, \boldsymbol{v}) \propto \exp\left(-\frac{1}{T}\left(U(\boldsymbol{x}) + \frac{\|\boldsymbol{v}\|^2}{2}\right)\right),$$

where $U(\boldsymbol{x})$ is the target potential energy ($U(\boldsymbol{x}) = H(\boldsymbol{x}) + \frac{\lambda}{2} \sum_{i=1}^{n} \|\boldsymbol{r}_i - \overline{\boldsymbol{r}}\|^2$ in our case), $\boldsymbol{v}$ are the particle velocities, $T$ is the temperature, and we assume unit mass. The corresponding stochastic differential equations (SDEs) are:

$$\begin{cases} \mathrm{d}\boldsymbol{x}_t &= \boldsymbol{v}_t \mathrm{d}t \\ \mathrm{d}\boldsymbol{v}_t &= \boldsymbol{F}(\boldsymbol{x}_t)\mathrm{d}t - \gamma\boldsymbol{v}_t\mathrm{d}t + \sqrt{2\gamma T}\mathrm{d}\boldsymbol{w}_t, \end{cases}$$

where $\boldsymbol{F}(\boldsymbol{x}) = -\nabla U(\boldsymbol{x})$ is the force, $\gamma$ is the friction coefficient, and $(\boldsymbol{w}_t)_{t \geq 0}$ represents a standard Wiener process. For both experiments, we use $\gamma = 0.5$.

**Initialization.** Particle positions $\boldsymbol{x}_0$ are initialized on a perturbed lattice separated by the characteristic length of each system, and the center of mass is removed. Initial velocities $\boldsymbol{v}_0$ are drawn from the equilibrium Maxwell-Boltzmann distribution, with each component sampled independently from $\mathcal{N}(0, T)$.

**Numerical Integration.** The Langevin SDEs are numerically integrated using the BAOAB splitting scheme (Leimkuhler & Matthews, 2013), which is a highly accurate and stable method for thermostatted systems. For a discrete time step $\Delta t$, the update from state $(\boldsymbol{x}_n, \boldsymbol{v}_n)$ to $(\boldsymbol{x}_{n+1}, \boldsymbol{v}_{n+1})$ is performed in five sequential steps:

$$\text{B: } \boldsymbol{v}_{n+1/2} = \boldsymbol{v}_n + \frac{\Delta t}{2}\boldsymbol{F}(\boldsymbol{x}_n),$$

$$\text{A: } \boldsymbol{x}_{n+1/2} = \boldsymbol{x}_n + \frac{\Delta t}{2}\boldsymbol{v}_{n+1/2},$$

$$\text{O: } \widehat{\boldsymbol{v}}_{n+1/2} = e^{-\gamma\Delta t}\boldsymbol{v}_{n+1/2} + \sqrt{T(1 - e^{-2\gamma\Delta t})}\boldsymbol{\xi}_n,$$

$$\text{A: } \boldsymbol{x}_{n+1} = \boldsymbol{x}_{n+1/2} + \frac{\Delta t}{2}\widehat{\boldsymbol{v}}_{n+1/2},$$

$$\text{B: } \boldsymbol{v}_{n+1} = \widehat{\boldsymbol{v}}_{n+1/2} + \frac{\Delta t}{2}\boldsymbol{F}(\boldsymbol{x}_{n+1}),$$

where $\boldsymbol{\xi}_n$ is a vector of independent standard normal random variables, i.e., $\boldsymbol{\xi}_n \sim \mathcal{N}(0, \mathbf{I})$.

**Sampling Protocol.** The simulation begins with a burn-in phase, during which the system is evolved for $N_{\text{burn}}$ steps to allow it to equilibrate with the stationary distribution. Following this, the sampling phase begins. To ensure that the collected samples are approximately independent, the system evolves for $N_{\text{interval}}$ steps between each saved configuration. This process is repeated until the desired number of samples is obtained.

### B.4 TRAINING AND INFERENCE DETAILS

We train our score-network using the Elucidating the Design Space of Diffusion Models (EDM) framework (Karras et al., 2022). This approach frames the learning problem as a denoising task, where a single neural network is trained to remove noise from corrupted data across a continuous range of noise levels.

**Network Preconditioning.** The core component is a neural network $\boldsymbol{F}(\cdot; \sigma)$ that is wrapped with a set of preconditioning functions dependent on the noise level $\sigma$. This design ensures numerical stability and improves performance across all noise scales. The final denoiser, $\boldsymbol{D}(\boldsymbol{x}_{\text{noisy}}; \sigma)$, which predicts a clean sample from a noisy one, is defined as:

$$\boldsymbol{D}(\boldsymbol{x}_{\text{noisy}}; \sigma) = c_{\text{skip}}(\sigma)\boldsymbol{x}_{\text{noisy}} + c_{\text{out}}(\sigma)\boldsymbol{F}\left(c_{\text{in}}(\sigma)\boldsymbol{x}_{\text{noisy}}, c_{\text{noise}}(\sigma)\right). \tag{B.2}$$

The functions $c_{\text{skip}}$, $c_{\text{in}}$, and $c_{\text{out}}$ provide scaling and a skip connection, while $c_{\text{noise}}(\sigma)$ creates a time-like embedding from the noise level.

We adopt the preconditioning functions as follows:

$$c_{\text{skip}}(\sigma) = \frac{\sigma_{\text{data}}^2}{\sigma^2 + \sigma_{\text{data}}^2}, \quad c_{\text{in}}(\sigma) = \frac{1}{\sqrt{\sigma^2 + \sigma_{\text{data}}^2}}, \quad c_{\text{out}}(\sigma) = \frac{\sigma\sigma_{\text{data}}}{\sqrt{\sigma^2 + \sigma_{\text{data}}^2}}, \quad c_{\text{noise}}(\sigma) = \frac{1}{4}\log\sigma,$$

where $\sigma_{\text{data}}$ is a hyperparameter varying with the task. We set $\sigma_{\text{data}} = 1.8$ for the DW-4 experiment and $\sigma_{\text{data}} = 0.68$ for the LJ-13 experiment.

**Training Objective.** The network is trained by corrupting clean data samples $\boldsymbol{x}_{\text{clean}}$ with Gaussian noise of a given standard deviation $\sigma$, creating noisy samples $\boldsymbol{x}_{\text{noisy}} = \boldsymbol{x}_{\text{clean}} + \sigma\boldsymbol{\epsilon}$, where $\boldsymbol{\epsilon} \sim \mathcal{N}(0, \boldsymbol{I})$. The training objective is to minimize the weighted mean squared error between the denoiser's prediction and the original clean data. The loss function is given by:

$$\mathcal{L}_{\text{EDM}} = \mathbb{E}_{\boldsymbol{x}_{\text{clean}} \sim p_0, \sigma, \boldsymbol{\epsilon} \sim \mathcal{N}(0, \boldsymbol{I})} \left[ w(\sigma) \left\| \boldsymbol{D}(\boldsymbol{x}_{\text{noisy}}; \sigma) - \boldsymbol{x}_{\text{clean}} \right\|^2 \right],$$

where the distribution of $\sigma$ and the weighting function $w(\sigma)$ will be specified later.

Following the EDM framework, the noise level $\sigma$ is sampled from a log-normal distribution:

$$\log \sigma \sim \mathcal{N}(P_{\text{mean}}, P_{\text{std}}^2),$$

with $P_{\text{mean}} = -1.2$ and $P_{\text{std}} = 1.2$. To avoid numerical instability, we clip the noise level to be within the range $[\sigma_{\text{min}}, \sigma_{\text{max}}]$ with $\sigma_{\text{min}} = 0.002$ and $\sigma_{\text{max}} = 80$. The weighting function is set as

$$w(\sigma) = \frac{\sigma^2 + \sigma_{\text{data}}^2}{\sigma^2 \sigma_{\text{data}}^2}.$$

**Inference Process.** While the model is trained as a denoiser, sampling requires the score function, $\nabla_{\boldsymbol{x}} \log p(\boldsymbol{x}_{\text{noisy}}; \sigma)$. The trained denoiser $\boldsymbol{D}$ is converted to the score during inference using the following exact relation:

$$\nabla_{\boldsymbol{x}} \log p(\boldsymbol{x}_{\text{noisy}}; \sigma) = \frac{\boldsymbol{D}(\boldsymbol{x}_{\text{noisy}}; \sigma) - \boldsymbol{x}_{\text{noisy}}}{\sigma^2}.$$

The noise level $\sigma$ coincides with the forward time $s$ in our setting in Eqn. (2.1) and we have $T = \sigma'_{\text{max}}$. The time discretization is chosen as

$$\{t_k\}_{k=0}^M = \left\{ \sigma'_{\text{max}} - \left( \sigma'^{1/\rho}_{\text{max}} + \frac{k}{M} \left( \sigma'^{1/\rho}_{\text{min}} - \sigma'^{1/\rho}_{\text{max}} \right) \right)^\rho \right\}_{k=0}^M,$$

where we choose a smaller range of noise levels for the parameters $\sigma'_{\text{min}} = 0.005$ and $\sigma'_{\text{max}} = 50$ to avoid the boundary part of the noise level, which may be less accurate due to the lack of training and singularity. We use $\rho = 7$ to control the spacing of the discretization. We use $M = 500$ for GMM experiments and $M = 1000$ time steps for particle system experiments.

**Training Settings.** For the neural network $\boldsymbol{F}(\cdot; \sigma)$ in the denoiser (B.2), we adopt the EGNN architecture described in App. B.2. For the training process, we use the Adam optimizer (Kingma & Ba, 2014) with a learning rate of $5 \times 10^{-4}$ and a batch size of 1024. We train the network for $10^6$ steps, where we sample a fresh dataset of $10^6$ data points with the same protocol as described in App. B.3 for every $2 \times 10^5$ steps. The warm-up period is only performed once before the first batch. All training is performed on a single NVIDIA A100 GPU. The training code is based on both the JAX library (Bradbury et al., 2018) and the Optax gradient processing and optimization library (DeepMind et al., 2020).

## B.5 EVALUATION METRICS

To quantitatively assess the quality of the samples generated by each method against a ground-truth reference distribution, we employ a suite of five complementary metrics. Let the generated samples be a weighted set $\{\boldsymbol{x}^{(i)}, w^{(i)}\}_{i \in [N]}$ and the reference samples be $\{\boldsymbol{x}_{\text{ref}}^{(j)}, w_{\text{ref}}^{(j)}\}_{j \in [N_{\text{ref}}]}$.

**Negative Log-Likelihood Difference ($\Delta$NLL).** The Negative Log-Likelihood (NLL) measures how well a set of samples fits the target probability distribution $p(\boldsymbol{x})$. For a weighted set of samples, it is estimated as the weighted average of the negative log-probabilities:

$$\text{NLL} = -\sum_{i=1}^N w^{(i)} \log p(\boldsymbol{x}^{(i)}).$$

We report $\Delta$NLL, which is the difference between the NLL of the samples generated by a method and the NLL of the reference samples: $\Delta\text{NLL} = \text{NLL}_{\text{method}} - \text{NLL}_{\text{ref}}$. A lower absolute value indicates a better fit to the target distribution.

**Maximum Mean Discrepancy (MMD).** Maximum Mean Discrepancy (MMD) is an integral probability metric defined in a Reproducing Kernel Hilbert Space (RKHS) $\mathcal{H}$ with a characteristic kernel $k(\cdot, \cdot)$. Any probability distribution $P$ admits a unique *mean embedding* $\mu_P \in \mathcal{H}$. The squared MMD between two distributions $P$ and $Q$ is the squared RKHS distance between their embeddings:

$$\mathrm{MMD}^2(P, Q) = \|\mu_P - \mu_Q\|_{\mathcal{H}}^2 = \|\mathbb{E}_{\boldsymbol{x} \sim P}[k(\boldsymbol{x}, \cdot)] - \mathbb{E}_{\boldsymbol{y} \sim Q}[k(\boldsymbol{y}, \cdot)]\|_{\mathcal{H}}^2.$$

Using the kernel trick, this definition can be expressed entirely in terms of kernel evaluations, avoiding explicit computation of the embeddings. For two weighted sample sets, the unbiased empirical estimator is

$$\mathrm{MMD}^2 = \|\mathbb{E}_{\boldsymbol{x} \sim P}[k(\boldsymbol{x}, \cdot)] - \mathbb{E}_{\boldsymbol{y} \sim Q}[k(\boldsymbol{y}, \cdot)]\|_{\mathcal{H}}^2$$
$$= \mathbb{E}_{\boldsymbol{x}, \boldsymbol{x}' \sim P}[k(\boldsymbol{x}, \boldsymbol{x}')] + \mathbb{E}_{\boldsymbol{y}, \boldsymbol{y}' \sim Q}[k(\boldsymbol{y}, \boldsymbol{y}')] - 2\mathbb{E}_{\boldsymbol{x} \sim P, \boldsymbol{y} \sim Q}[k(\boldsymbol{x}, \boldsymbol{y})]$$
$$\approx \sum_{i,j=1}^{N} w^{(i)} w^{(j)} k(\boldsymbol{x}^{(i)}, \boldsymbol{x}^{(j)}) + \sum_{i,j=1}^{N_{\mathrm{ref}}} w_{\mathrm{ref}}^{(i)} w_{\mathrm{ref}}^{(j)} k(\boldsymbol{x}_{\mathrm{ref}}^{(i)}, \boldsymbol{x}_{\mathrm{ref}}^{(j)}) - 2 \sum_{i=1}^{N} \sum_{j=1}^{N_{\mathrm{ref}}} w^{(i)} w_{\mathrm{ref}}^{(j)} k(\boldsymbol{x}^{(i)}, \boldsymbol{x}_{\mathrm{ref}}^{(j)}).$$

We use the RBF kernel

$$k(\boldsymbol{x}, \boldsymbol{y}) = \exp\left(-\frac{\|\boldsymbol{x} - \boldsymbol{y}\|^2}{2\sigma_k^2}\right),$$

with $\sigma_k = 20$ for the GMM experiment and $\sigma_k = 5$ for the DW-4 and LJ-13 experiments.

Direct evaluation of MMD requires $\mathcal{O}(N^2)$ kernel evaluations, which becomes expensive for large sample sets. To scale up computation, we use *Random Fourier Features (RFF)* to approximate the RBF kernel with an explicit low-dimensional feature map $z(\boldsymbol{x}) \in \mathbb{R}^f$, reducing the complexity to $\mathcal{O}(NN_f)$.

Bochner's theorem states that a shift-invariant kernel can be expressed as the expectation of a product of complex Fourier features with respect to its spectral density $p(\boldsymbol{\omega})$:

$$k(\boldsymbol{x}, \boldsymbol{y}) = \mathbb{E}_{\boldsymbol{\omega} \sim p(\boldsymbol{\omega})}\left[e^{i\boldsymbol{\omega}^\top \boldsymbol{x}} e^{-i\boldsymbol{\omega}^\top \boldsymbol{y}}\right].$$

For the RBF kernel, $p(\boldsymbol{\omega})$ is Gaussian, $\mathcal{N}(0, \sigma_k^{-2}\mathbf{I})$. The RFF approximation replaces the expectation with a Monte Carlo average over $f/2$ sampled frequencies $\boldsymbol{\omega}_k \sim \mathcal{N}(0, \sigma_k^{-2}\mathbf{I})$ and random phases $b_k \sim \mathrm{Unif}[0, 2\pi]$, yielding the explicit feature map

$$\boldsymbol{z}(\boldsymbol{x}) = \sqrt{\frac{2}{f}}\left(\cos(\boldsymbol{\omega}_1^\top \boldsymbol{x} + b_1), \ldots, \cos(\boldsymbol{\omega}_{f/2}^\top \boldsymbol{x} + b_{f/2}), \sin(\boldsymbol{\omega}_1^\top \boldsymbol{x} + b_1), \ldots, \sin(\boldsymbol{\omega}_{f/2}^\top \boldsymbol{x} + b_{f/2})\right)^\top.$$

This ensures that $\boldsymbol{z}(\boldsymbol{x})^\top \boldsymbol{z}(\boldsymbol{y}) \approx k(\boldsymbol{x}, \boldsymbol{y})$. The MMD can then be efficiently approximated as the squared Euclidean distance between the mean feature vectors:

$$\mathrm{MMD}^2 \approx \|\mathbb{E}_{\boldsymbol{x} \sim P}[\boldsymbol{z}(\boldsymbol{x})] - \mathbb{E}_{\boldsymbol{y} \sim Q}[\boldsymbol{z}(\boldsymbol{y})]\|_2^2 \approx \left\|\sum_{i=1}^{N} w^{(i)} \boldsymbol{z}(\boldsymbol{x}^{(i)}) - \sum_{j=1}^{N_{\mathrm{ref}}} w_{\mathrm{ref}}^{(j)} \boldsymbol{z}(\boldsymbol{x}_{\mathrm{ref}}^{(j)})\right\|_2^2.$$

In our experiments, we use 2048 random features ($f = 2048$) for all experiments.

**Sliced-Wasserstein Distance (SWD).** The Wasserstein distance measures the minimum cost to transport mass from one distribution to another. As computing exact 2-Wasserstein distance between high-dimensional empirical distributions with up to $2^{15}$ particles in our experiments is computationally prohibitive, we report Sliced-Wasserstein Distance (SWD) as a computationally efficient approximation that involves projecting the high-dimensional distributions onto a series of random 1D lines, calculating the 1D Wasserstein-2 distance for each projection, and averaging the results:

$$\mathrm{SWD}_2(P, Q)^2 = \int_{\mathbb{S}^{D-1}} W_2^2(\pi_\theta(P), \pi_\theta(Q)) \mathrm{d}\theta \approx \frac{1}{p} \sum_{i=1}^{p} W_2^2(\pi_{\theta_i}(P), \pi_{\theta_i}(Q)),$$

where $\pi_\theta$ denotes the projection onto a line with direction $\theta \in \mathbb{S}^{D-1}$, $p$ is the number of projections, and $\theta_i \sim \mathrm{Unif}(\mathbb{S}^{D-1})$. In our experiments, we use $p = 10$ for all experiments.

**Radial Distribution Function Wasserstein-1 Distance ($W_1^{\mathbf{RDF}}$).** The Radial Distribution Function (RDF), $g(r)$, is a fundamental descriptor of the spatial arrangement of particles in a system, measuring the probability of finding a particle at a distance $r$ from another. For simplicity and following the literature (Akhound-Sadegh et al., 2024; Skreta et al., 2025), we define it as the ensemble-averaged interatomic distance distribution:

$$g(r) = \frac{2}{N(N-1)} \sum_{i<j} \langle \delta(r - \|\boldsymbol{r}_i - \boldsymbol{r}_j\|) \rangle,$$

where $\langle \cdot \rangle$ denotes the ensemble average over all configurations. With slight abuse of notation, we still refer to $g(r)$ as the RDF without the shell factor and density normalization.

To assess the structural accuracy of our samples, we compute the RDF for both the generated and reference configurations, yielding two 1D distributions, $g_{\mathrm{method}}(r)$ and $g_{\mathrm{ref}}(r)$. The $W_1^{\mathrm{RDF}}$ metric is the 1-Wasserstein distance between these two distributions, $W_1(g_{\mathrm{method}}, g_{\mathrm{ref}})$, which quantifies the difference in the learned physical structure.

**Energy Wasserstein-1 Distance ($W_1^{\mathcal{E}}$).** The energy of a configuration $\boldsymbol{x}$ is its negative log-probability, $\mathcal{E}(\boldsymbol{x}) = -\log q(\boldsymbol{x})$. Due to the unknown normalization constant for the reward-tilted and annealed distribution $q(\boldsymbol{x}) \propto p^{\gamma}(\boldsymbol{x}) e^{r(\boldsymbol{x})}$, we report the energy up to a constant, *i.e.*,

$$\mathcal{E}(\boldsymbol{x}) = -\gamma \log p(\boldsymbol{x}) - r(\boldsymbol{x}).$$

By evaluating the energy for every sample, we obtain two 1D distributions of energies, $P_{\mathcal{E}}^{\mathrm{method}}$ and $P_{\mathcal{E}}^{\mathrm{ref}}$. The $W_1^{\mathcal{E}}$ metric is the 1-Wasserstein distance, $W_1(P_{\mathcal{E}}^{\mathrm{method}}, P_{\mathcal{E}}^{\mathrm{ref}})$, between these energy distributions. This metric evaluates how well a method captures the correct energy landscape and the relative probabilities of different configurations.

**Physical Validity.** To evaluate the physical plausibility of generated protein-ligand structures, we adopt the validity metrics from PoseBusters (Buttenschoen et al., 2024), which assess adherence to fundamental physical and chemical rules.

- *Valid Fraction:* The fraction of generated structures that satisfy all validity checks simultaneously, serving as an overall measure of correctness.
- *Clash-Free Fraction:* The proportion of structures without severe steric clashes, i.e., unphysical overlaps between atoms.
- *Bond Length and Bond Angle Violations:* Counts of deviations from standard covalent geometry. Bond length violations occur when bonds are too short or too long, while bond angle violations correspond to unrealistic angular geometries.
- *Internal Clashes:* The number of steric overlaps within the same ligand molecule, reflecting poor internal consistency.
- *Chiral Atom Errors:* The number of stereocenters incorrectly assigned, such as inversions of chirality.
- *Chain Clashes:* The number of steric overlaps between ligand atoms and protein atoms, indicating violations of intermolecular packing constraints.

## C  ADDITIONAL EXPERIMENTAL RESULTS

This appendix provides supplementary results that further explore the performance of our methods, DriftLite-VCG(-SMC) and DriftLite-ECG(-SMC), under various conditions for both the Gaussian Mixture Model (GMM) and the particle-based systems (DW-4 and LJ-13).

### C.1  ADDITIONAL EXPERIMENTAL RESULTS OF GMM

We first present a more extensive ablation study on the GMM. Tab. 3 details the performance of all methods across a wider range of annealing factors, $\gamma \in \{1.5, 2.0, 2.5, 3.0\}$. The higher the annealing factor $\gamma$ becomes, the more challenging the inference-time scaling task is, as it accentuates

the modes of the distribution and increases the energy barriers between them. The results reinforce the conclusions from the main text: as $\gamma$ grows, the performance gap between our DriftLite methods and the baselines widens considerably. VCG-SMC, in particular, consistently achieves the best or near-best performance across all metrics, showcasing its robustness.

Similarly, Tab. 4 extends the reward-tilting experiments to different reward strengths ($\sigma \in \{100, 200, 300, 400\}$). For the definition of the parameter $\sigma$, we refer to the problem settings in App. B.1. Again, our methods demonstrate superior stability and accuracy compared to Pure Guidance (PG) and G-SMC, which degrade significantly as the reward becomes more peaked (smaller reward covariance scale $\sigma$).

Table 3: Performance ablation for the GMM annealing task with varying annealing factor $\gamma$. Results are mean$_{\pm\text{std}}$ over 5 configurations. Best results per column (within each $\gamma$ block) are in bold.

| Method | Annealing ($\gamma = 1.5$) | | | | | Annealing ($\gamma = 2.0$) | | | | |
|---|---|---|---|---|---|---|---|---|---|---|
| | $\Delta$NLL | MMD$_{\times 10^{-1}}$ | SWD | Mean $L_2$ | Cov F$_{\times 10^3}$ | $\Delta$NLL | MMD$_{\times 10^{-1}}$ | SWD | Mean $L_2$ | Cov F$_{\times 10^3}$ |
| PG | $-1.196_{\pm 0.621}$ | $4.471_{\pm 2.452}$ | $11.90_{\pm 6.391}$ | $58.98_{\pm 33.92}$ | $5.274_{\pm 0.660}$ | $-3.674_{\pm 0.774}$ | $6.576_{\pm 2.205}$ | $18.06_{\pm 6.494}$ | $80.10_{\pm 33.03}$ | $9.347_{\pm 2.428}$ |
| G-SMC | $0.441_{\pm 0.099}$ | $0.834_{\pm 0.070}$ | $5.749_{\pm 1.368}$ | $29.15_{\pm 2.890}$ | $3.097_{\pm 0.389}$ | $-0.527_{\pm 0.262}$ | $2.057_{\pm 0.497}$ | $11.44_{\pm 3.625}$ | $58.71_{\pm 14.71}$ | $5.097_{\pm 0.575}$ |
| ECG | $0.257_{\pm 0.083}$ | $0.185_{\pm 0.003}$ | $0.622_{\pm 0.097}$ | $2.982_{\pm 0.331}$ | $0.350_{\pm 0.033}$ | $0.324_{\pm 0.145}$ | $0.415_{\pm 0.037}$ | $1.209_{\pm 0.164}$ | $5.323_{\pm 0.658}$ | $0.750_{\pm 0.069}$ |
| ECG-SMC | $0.219_{\pm 0.053}$ | $\mathbf{0.162_{\pm 0.004}}$ | $0.605_{\pm 0.139}$ | $2.667_{\pm 0.687}$ | $0.335_{\pm 0.036}$ | $0.198_{\pm 0.073}$ | $0.185_{\pm 0.009}$ | $0.779_{\pm 0.131}$ | $3.782_{\pm 0.476}$ | $0.446_{\pm 0.045}$ |
| VCG | $0.222_{\pm 0.045}$ | $\mathbf{0.166_{\pm 0.002}}$ | $\mathbf{0.590_{\pm 0.058}}$ | $\mathbf{2.661_{\pm 0.172}}$ | $\mathbf{0.335_{\pm 0.018}}$ | $0.204_{\pm 0.052}$ | $0.188_{\pm 0.001}$ | $0.672_{\pm 0.093}$ | $2.971_{\pm 0.175}$ | $0.379_{\pm 0.030}$ |
| VCG-SMC | $\mathbf{0.203_{\pm 0.061}}$ | $0.162_{\pm 0.002}$ | $0.638_{\pm 0.055}$ | $2.852_{\pm 0.058}$ | $0.346_{\pm 0.021}$ | $\mathbf{0.192_{\pm 0.063}}$ | $\mathbf{0.166_{\pm 0.005}}$ | $\mathbf{0.606_{\pm 0.094}}$ | $\mathbf{2.866_{\pm 0.543}}$ | $\mathbf{0.344_{\pm 0.027}}$ |

| Method | Annealing ($\gamma = 2.5$) | | | | | Annealing ($\gamma = 3.0$) | | | | |
|---|---|---|---|---|---|---|---|---|---|---|
| | $\Delta$NLL | MMD | SWD | Mean $L_2$ | Cov F$_{\times 10^3}$ | $\Delta$NLL | MMD | SWD | Mean $L_2$ | Cov F$_{\times 10^3}$ |
| PG | $-5.016_{\pm 1.280}$ | $0.725_{\pm 0.208}$ | $20.27_{\pm 6.796}$ | $92.04_{\pm 30.59}$ | $9.290_{\pm 2.339}$ | $-4.950_{\pm 1.342}$ | $0.758_{\pm 0.158}$ | $19.55_{\pm 4.731}$ | $93.39_{\pm 21.64}$ | $8.654_{\pm 1.850}$ |
| G-SMC | $-0.801_{\pm 0.204}$ | $0.327_{\pm 0.073}$ | $13.88_{\pm 2.770}$ | $78.11_{\pm 18.39}$ | $5.829_{\pm 1.041}$ | $-0.692_{\pm 0.414}$ | $0.493_{\pm 0.071}$ | $18.31_{\pm 3.203}$ | $102.7_{\pm 9.285}$ | $4.973_{\pm 1.255}$ |
| ECG | $-0.427_{\pm 1.185}$ | $0.248_{\pm 0.255}$ | $6.443_{\pm 6.144}$ | $30.08_{\pm 28.89}$ | $3.135_{\pm 2.090}$ | $-1.201_{\pm 1.155}$ | $0.353_{\pm 0.192}$ | $8.967_{\pm 4.501}$ | $39.24_{\pm 21.08}$ | $4.858_{\pm 1.636}$ |
| ECG-SMC | $\mathbf{0.169_{\pm 0.070}}$ | $0.021_{\pm 0.002}$ | $1.002_{\pm 0.196}$ | $4.886_{\pm 1.149}$ | $0.564_{\pm 0.088}$ | $0.184_{\pm 0.079}$ | $0.031_{\pm 0.003}$ | $1.672_{\pm 2.14}$ | $7.795_{\pm 1.164}$ | $0.850_{\pm 0.107}$ |
| VCG | $0.204_{\pm 0.058}$ | $0.023_{\pm 0.001}$ | $0.717_{\pm 0.108}$ | $3.351_{\pm 0.222}$ | $0.420_{\pm 0.037}$ | $0.209_{\pm 0.080}$ | $0.029_{\pm 0.002}$ | $0.859_{\pm 0.147}$ | $4.071_{\pm 0.507}$ | $0.505_{\pm 0.055}$ |
| VCG-SMC | $0.179_{\pm 0.065}$ | $\mathbf{0.018_{\pm 0.001}}$ | $\mathbf{0.613_{\pm 0.109}}$ | $\mathbf{2.867_{\pm 0.364}}$ | $\mathbf{0.380_{\pm 0.051}}$ | $\mathbf{0.174_{\pm 0.073}}$ | $\mathbf{0.019_{\pm 0.001}}$ | $\mathbf{0.691_{\pm 0.149}}$ | $\mathbf{3.319_{\pm 0.593}}$ | $\mathbf{0.415_{\pm 0.048}}$ |

Table 4: Performance ablation for the GMM reward-tilting task with varying reward strength $\sigma$. Results are mean$_{\pm\text{std}}$ over 5 runs. Best results per column (within each $\sigma$ block) are in bold.

| Method | Reward-Tilting ($\sigma = 100$) | | | | | Reward-Tilting ($\sigma = 200$) | | | | |
|---|---|---|---|---|---|---|---|---|---|---|
| | $\Delta$NLL | MMD | SWD | Mean $L_2$ | Cov F$_{\times 10^3}$ | $\Delta$NLL | MMD | SWD | Mean $L_2$ | Cov F$_{\times 10^3}$ |
| PG | $0.338_{\pm 2.134}$ | $0.533_{\pm 0.195}$ | $5.454_{\pm 1.710}$ | $34.57_{\pm 13.35}$ | $1.957_{\pm 0.991}$ | $-0.849_{\pm 2.772}$ | $0.614_{\pm 0.143}$ | $6.493_{\pm 2.009}$ | $32.92_{\pm 14.04}$ | $1.919_{\pm 1.384}$ |
| G-SMC | $\mathbf{0.170_{\pm 0.461}}$ | $0.168_{\pm 0.072}$ | $2.077_{\pm 0.771}$ | $14.72_{\pm 8.406}$ | $0.748_{\pm 0.400}$ | $\mathbf{-0.032_{\pm 0.242}}$ | $0.063_{\pm 0.012}$ | $1.588_{\pm 0.756}$ | $10.52_{\pm 7.139}$ | $0.688_{\pm 0.403}$ |
| ECG | $0.289_{\pm 0.104}$ | $0.035_{\pm 0.006}$ | $0.318_{\pm 0.074}$ | $1.159_{\pm 0.207}$ | $0.068_{\pm 0.020}$ | $0.273_{\pm 0.285}$ | $0.050_{\pm 0.013}$ | $0.636_{\pm 0.221}$ | $2.872_{\pm 1.023}$ | $0.186_{\pm 0.076}$ |
| ECG-SMC | $0.292_{\pm 0.042}$ | $0.016_{\pm 0.002}$ | $\mathbf{0.177_{\pm 0.028}}$ | $0.818_{\pm 0.218}$ | $0.043_{\pm 0.018}$ | $0.303_{\pm 0.048}$ | $0.022_{\pm 0.003}$ | $\mathbf{0.259_{\pm 0.058}}$ | $\mathbf{1.398_{\pm 0.301}}$ | $\mathbf{0.098_{\pm 0.037}}$ |
| VCG | $0.230_{\pm 0.115}$ | $0.030_{\pm 0.003}$ | $0.293_{\pm 0.047}$ | $1.075_{\pm 0.191}$ | $0.069_{\pm 0.023}$ | $0.253_{\pm 0.118}$ | $0.033_{\pm 0.005}$ | $0.392_{\pm 0.095}$ | $1.781_{\pm 0.938}$ | $0.119_{\pm 0.045}$ |
| VCG-SMC | $0.240_{\pm 0.051}$ | $\mathbf{0.016_{\pm 0.001}}$ | $0.177_{\pm 0.042}$ | $\mathbf{0.701_{\pm 0.267}}$ | $\mathbf{0.041_{\pm 0.025}}$ | $0.280_{\pm 0.045}$ | $\mathbf{0.018_{\pm 0.001}}$ | $0.286_{\pm 0.071}$ | $1.471_{\pm 0.449}$ | $0.105_{\pm 0.042}$ |

| Method | Reward-Tilting ($\sigma = 300$) | | | | | Reward-Tilting ($\sigma = 400$) | | | | |
|---|---|---|---|---|---|---|---|---|---|---|
| | $\Delta$NLL | MMD | SWD | Mean $L_2$ | Cov F$_{\times 10^3}$ | $\Delta$NLL | MMD | SWD | Mean $L_2$ | Cov F$_{\times 10^3}$ |
| PG | $\mathbf{-0.028_{\pm 2.553}}$ | $0.461_{\pm 0.123}$ | $5.911_{\pm 2.235}$ | $27.86_{\pm 12.14}$ | $2.248_{\pm 1.378}$ | $\mathbf{-0.010_{\pm 1.800}}$ | $0.350_{\pm 0.084}$ | $5.371_{\pm 2.223}$ | $23.93_{\pm 9.445}$ | $2.272_{\pm 1.122}$ |
| G-SMC | $0.056_{\pm 0.309}$ | $0.061_{\pm 0.019}$ | $2.651_{\pm 1.287}$ | $15.79_{\pm 7.903}$ | $1.108_{\pm 0.699}$ | $0.272_{\pm 0.102}$ | $0.049_{\pm 0.007}$ | $2.583_{\pm 1.043}$ | $17.04_{\pm 5.059}$ | $0.859_{\pm 0.166}$ |
| ECG | $0.162_{\pm 0.359}$ | $0.070_{\pm 0.026}$ | $1.078_{\pm 0.582}$ | $5.739_{\pm 3.140}$ | $0.432_{\pm 0.344}$ | $0.135_{\pm 0.381}$ | $0.114_{\pm 0.087}$ | $2.036_{\pm 2.310}$ | $10.43_{\pm 12.51}$ | $0.910_{\pm 1.122}$ |
| ECG-SMC | $0.375_{\pm 0.098}$ | $0.030_{\pm 0.008}$ | $0.545_{\pm 0.158}$ | $2.931_{\pm 0.994}$ | $0.207_{\pm 0.048}$ | $0.207_{\pm 0.196}$ | $0.036_{\pm 0.011}$ | $0.817_{\pm 0.472}$ | $4.760_{\pm 2.943}$ | $0.366_{\pm 0.173}$ |
| VCG | $0.229_{\pm 0.089}$ | $0.033_{\pm 0.006}$ | $0.454_{\pm 0.073}$ | $2.097_{\pm 1.241}$ | $0.150_{\pm 0.034}$ | $0.230_{\pm 0.091}$ | $0.032_{\pm 0.006}$ | $0.463_{\pm 0.081}$ | $2.099_{\pm 0.777}$ | $0.184_{\pm 0.052}$ |
| VCG-SMC | $0.273_{\pm 0.059}$ | $\mathbf{0.019_{\pm 0.002}}$ | $\mathbf{0.285_{\pm 0.033}}$ | $\mathbf{1.289_{\pm 0.140}}$ | $\mathbf{0.125_{\pm 0.021}}$ | $0.292_{\pm 0.043}$ | $\mathbf{0.020_{\pm 0.002}}$ | $\mathbf{0.380_{\pm 0.047}}$ | $\mathbf{1.939_{\pm 0.865}}$ | $\mathbf{0.159_{\pm 0.056}}$ |

Finally, Fig. 9 and Fig. 10 complement the figures in Sec. 4.1. Fig. 9 illustrates the evolution of ESS and variance for a milder reward-tilting task ($\sigma = 50.0$), showing that even in less challenging scenarios, our control mechanism actively stabilizes the particle weights. Fig. 10 shows the performance of all methods as a function of the number of particles for a strong reward-tilting task ($\sigma = 400.0$). It clearly indicates that our VCG and ECG methods not only outperform the baselines but also converge more efficiently, achieving better results with fewer particles.

As shown in Tab. 5, for the 30-dimensional GMM problem, the relative runtime for our VCG-SMC and ECG-SMC methods is only around 3 times that of the standard Guidance-SMC (G-SMC) baseline. This modest increase in computation time is largely due to the effective parallelization of the

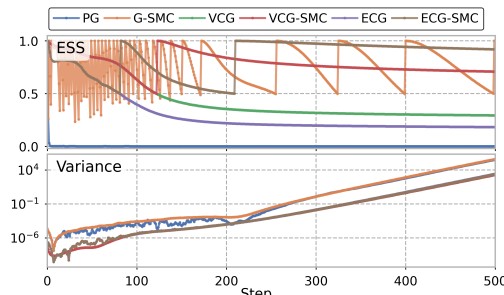
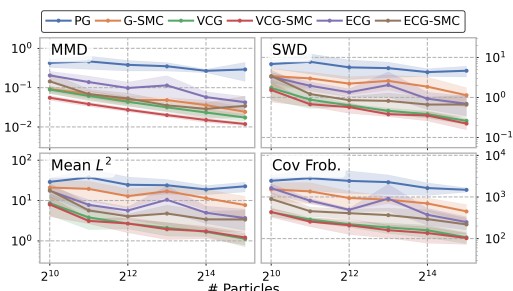

Figure 9: Evolution of ESS and potential variance for the GMM reward-tilting task ($\sigma = 50.0$).

Figure 10: Performance metrics versus number of particles for the GMM reward-tilting task ($\sigma = 400.0$).

Table 5: Elapsed time comparison for different sampling methods. Results are mean$_{\pm\text{std}}$ over 5 runs. The relative runtime, including propagated standard deviation, is computed with respect to the G-SMC method.

| Method | Annealing ($\gamma = 2.5$) | | Reward-Tilting ($\sigma = 100$) | |
| --- | --- | --- | --- | --- |
| | Runtime (s) | Relative Runtime | Runtime (s) | Relative Runtime |
| PG | $6.30_{\pm 0.17}$ | $0.90_{\pm 0.03}\times$ | $6.03_{\pm 0.19}$ | $0.91_{\pm 0.04}\times$ |
| G-SMC | $7.01_{\pm 0.12}$ | $1.00_{\pm 0.02}\times$ | $6.63_{\pm 0.15}$ | $1.00_{\pm 0.03}\times$ |
| **VCG-SMC** | $21.83_{\pm 0.05}$ | $3.11_{\pm 0.05}\times$ | $21.24_{\pm 0.06}$ | $3.20_{\pm 0.08}\times$ |
| **ECG-SMC** | $21.66_{\pm 0.32}$ | $3.09_{\pm 0.07}\times$ | $21.49_{\pm 0.08}$ | $3.24_{\pm 0.08}\times$ |

core algorithms. These results underscore the controlled additional cost required to implement the variance reduction technique, making them commonly practical. A more optimized implementation could reduce this runtime gap even further by exploiting more GPU resources.

We further conducted GMM experiments on the annealing task ($\gamma = 2.5$) and the quadratic reward-tilting task ($\sigma = 100$) under a matched wall-clock budget (Tab. 6). In these runs, the baselines (PG, G-SMC) used 8192 particles and 2000 diffusion steps, while DriftLite variants (VCG/ECG, with or without SMC) used 1024 particles and 1000 diffusion steps. Under this configuration, the wall-clock times of DriftLite methods are comparable to, and in some cases slightly lower than, those of PG and G-SMC, despite using substantially fewer particles and steps. At the same time, DriftLite variants, especially VCG-SMC and ECG-SMC, achieve markedly lower MMD, SWD, and mean-$\ell_2$ errors, while maintaining $\Delta$NLL close to zero. These results indicate that the added control drift substantially improves sample quality at essentially fixed computational cost, mitigating the variance and weight degeneracy that limit the uncontrolled SMC baseline.

**Comparison with a training-based neural drift controller (NCG).** To contextualize the trade-off between training-free inference-time control and amortized drift-learning approaches, we implemented a prototype training-based baseline, which we refer to as *Neural Controlling Guidance (NCG)*.

In NCG, the control drift is parameterized by a neural network $b_{t,\theta}(x)$ taking time and state as inputs, in contrast to DriftLite which solves for the control coefficients via small linear systems under a linear ansatz (cf. Eq. (3.3) and Ansatz 3.3). Concretely, we train $b_{t,\theta}$ by minimizing a regularized version of the same variance objective:

$$\mathcal{L}_t(\theta) = \text{Var}_{x \sim q_t}\big[g_t(x) + h_t(x; b_{t,\theta})\big] + \lambda\|b_{t,\theta}\|_2^2,$$

with $\lambda = 0.1$. We use a U-Net architecture with encoder feature dimensions $[128, 256, 512]$ and decoder feature dimensions $[256, 128]$. For training, we maintain an ensemble of $N = 8192$ particles (matching the inference-only baselines), and after each time step, we backpropagate $\nabla_\theta L_t(\theta)$ and update parameters using a learning rate $5 \times 10^{-4}$. We repeat this training–inference procedure over multiple refinement rounds, carrying the updated dynamics from the previous round as the base for the next.

Table 6: Performance with matched wall-clock budgets for the GMM tasks. Results are mean$_{\pm\text{std}}$ over 5 runs. Best results per column are in bold.

| Method | Annealing ($\gamma = 2.5$) | | | | | Reward-Tilting ($\sigma = 100$) | | | | |
|---|---|---|---|---|---|---|---|---|---|---|
| | $\Delta$NLL | MMD | SWD | Mean L2 | Runtime (s) | $\Delta$NLL | MMD | SWD | Mean L2 | Runtime (s) |
| PG | $-5.04_{\pm1.31}$ | $0.73_{\pm0.22}$ | $18.52_{\pm5.00}$ | $90.58_{\pm29.89}$ | $17.28_{\pm0.49}$ | $1.60_{\pm2.55}$ | $0.85_{\pm0.15}$ | $10.69_{\pm5.47}$ | $60.44_{\pm31.59}$ | $17.71_{\pm1.30}$ |
| G-SMC | $-0.56_{\pm0.85}$ | $0.46_{\pm0.07}$ | $18.87_{\pm3.69}$ | $102.17_{\pm20.40}$ | $18.45_{\pm0.84}$ | $\mathbf{-0.12_{\pm0.35}}$ | $0.12_{\pm0.02}$ | $1.64_{\pm0.81}$ | $9.32_{\pm4.82}$ | $17.63_{\pm1.37}$ |
| VCG | $0.11_{\pm0.20}$ | $0.06_{\pm0.00}$ | $1.85_{\pm0.36}$ | $7.76_{\pm1.55}$ | $15.12_{\pm0.14}$ | $0.29_{\pm0.19}$ | $0.09_{\pm0.01}$ | $0.81_{\pm0.15}$ | $4.29_{\pm1.30}$ | $17.24_{\pm0.48}$ |
| VCG-SMC | $\mathbf{0.04_{\pm0.23}}$ | $\mathbf{0.05_{\pm0.00}}$ | $\mathbf{1.75_{\pm0.41}}$ | $7.33_{\pm1.61}$ | $15.77_{\pm0.13}$ | $0.14_{\pm0.17}$ | $\mathbf{0.04_{\pm0.01}}$ | $0.53_{\pm0.24}$ | $3.01_{\pm1.54}$ | $15.68_{\pm0.48}$ |
| ECG | $-0.75_{\pm1.27}$ | $0.40_{\pm0.08}$ | $10.22_{\pm2.29}$ | $50.45_{\pm9.10}$ | $\mathbf{13.92_{\pm0.18}}$ | $0.19_{\pm0.40}$ | $0.12_{\pm0.03}$ | $1.15_{\pm0.32}$ | $5.66_{\pm2.05}$ | $\mathbf{15.09_{\pm0.48}}$ |
| ECG-SMC | $0.08_{\pm0.25}$ | $0.07_{\pm0.01}$ | $2.69_{\pm0.61}$ | $14.29_{\pm4.05}$ | $14.83_{\pm0.19}$ | $0.17_{\pm0.09}$ | $0.04_{\pm0.00}$ | $0.57_{\pm0.17}$ | $\mathbf{2.90_{\pm1.31}}$ | $15.30_{\pm0.58}$ |

Table 7: Comparison with training-based Neural Controlling Guidance (NCG) on the 30D GMM. NCG-SMC-$r$ denotes the $r$-th refinement/training round. *Inference Only* evaluates the learned drift from round 20 without backpropagation. Lower is better for all metrics. [*]Best-performing NCG round among those reported.

| Method | MMD | SWD | Mean $L_2$ | $\Delta$NLL | Runtime (s) |
|---|---|---|---|---|---|
| PG | 0.825 | 21.86 | 66.09 | -0.65 | 4.59 |
| G-SMC | 0.143 | 6.45 | 24.94 | 0.04 | 5.53 |
| **VCG-SMC** | 0.017 | 0.52 | 1.23 | 0.14 | 7.83 |
| **ECG-SMC** | 0.019 | 0.68 | 1.83 | 0.14 | 6.99 |
| NCG-SMC-1 | 0.437 | 12.29 | 51.44 | 0.39 | 70.79 |
| NCG-SMC-5 | 0.135 | 3.53 | 13.67 | 0.34 | 307.34 |
| NCG-SMC-9[*] | 0.091 | 2.45 | 9.11 | 0.05 | 543.77 |
| NCG-SMC-12 | 0.223 | 13.35 | 37.96 | 0.01 | 721.23 |
| NCG-SMC-16 | 0.143 | 7.61 | 21.73 | 0.16 | 957.72 |
| NCG-SMC-20 | 0.199 | 9.36 | 24.02 | 0.08 | 1194.31 |
| NCG-SMC-20 (Inference Only) | 0.190 | 8.91 | 29.04 | 0.26 | 240.88 |

Overall, NCG improves substantially during early refinement rounds but exhibits non-monotonic behavior as training continues, highlighting the optimization sensitivity of variance-based drift learning. Moreover, even without training (Inference Only), evaluating a high-capacity neural controller incurs significant inference-time overhead relative to DriftLite's lightweight linear solve.

## C.2 ADDITIONAL EXPERIMENTAL RESULTS OF PARTICLE SYSTEMS

Here, we provide additional ablation studies for the DW-4 and LJ-13 particle systems, demonstrating the robustness of our findings.

**ESS and Potential Variance Evolution.** In Fig. 12, we present a similar visualization as in the GMM example (Figs. 2 and 9), where similar trends are observed: pure guidance leads to a much rapid drop in ESS during the initial stage of the inference process, while the variance-controlling guidance postpones the drop when the resampling kicks in. We also observe that the time when the curve of the potential variance $\text{Var}_{q_t}[g_t]$ of all methods meets coincides with the drop in ESS of the variance-controlling guidance, which may correspond to the splitting of modes, and resampling may be crucial to handle.

**Elapsed Time Comparison.** The results for the DW4 system show that the performance scaling of the advanced methods is even more favorable than what was observed for the GMM task (Tab. 5). From Tab. 8, we can see that the relative runtime of the VCG-SMC methods is only about twice that of the G-SMC baseline. This remarkable efficiency stems from the nature of the problem itself; the total computational cost is dominated by repeated, computationally heavy score evaluations. Consequently, the fixed algorithmic overhead from our DriftLite variance reduction methods becomes negligible as a fraction of the total runtime, underscoring their practicality and efficiency for computationally intensive systems.

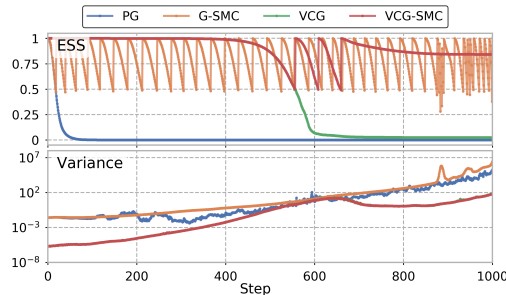
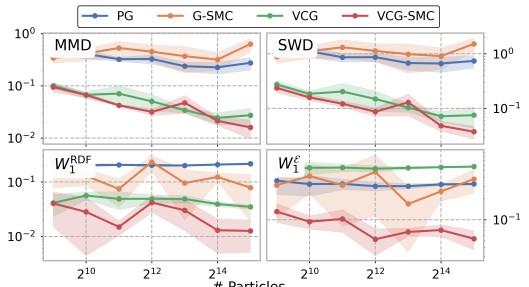

Figure 11: Evolution of ESS and potential variance for the DW-4 annealing task ($\gamma = 2.0$).

Figure 12: Performance metrics versus number of particles for the DW-4 annealing task ($\gamma = 2.0$).

Table 8: Elapsed time comparison for different sampling methods on the DW4 System. Results are mean$_{\pm\text{std}}$ over 5 runs. The relative runtime, including propagated standard deviation, is computed with respect to the G-SMC method.

| Method | Annealing ($\gamma = 2.0$) | | Reward-Tilting ($\lambda' = 0.5$) | |
|---|---|---|---|---|
| | Runtime (s) | Relative Runtime | Runtime (s) | Relative Runtime |
| G-SMC | $281.68_{\pm 9.69}$ | $1.00_{\pm 0.05}\times$ | $284.12_{\pm 8.03}$ | $1.00_{\pm 0.04}\times$ |
| **VCG-SMC** | $674.32_{\pm 6.89}$ | $2.39_{\pm 0.09}\times$ | $674.12_{\pm 1.75}$ | $2.37_{\pm 0.07}\times$ |

**Ablation on Number of Particles $N$.** Next, we present the ablation study on the number of particles on the DW-4 system as in the GMM example (Fig. 12). Despite a more complex problem nature and high-level evaluation metrics (RDF and energy distribution), our methods still present robust scaling in MMD and SWD, and show promising decay in $W_1^{\text{RDF}}$ and $W_1^{\mathcal{E}}$, highlighting the effectiveness and robustness of our methods.

**Ablation on Base Model Temperature $T$.** In Tab. 9, we investigate the effect of the base model's training temperature ($T \in \{1.0, 1.5\}$) on an annealing task for the DW-4 system in addition to the base temperature $T = 2.0$ as used in Tab. 1. The results show that our methods, particularly VCG-SMC, are effective regardless of the starting temperature. They successfully anneal the system to the target low-temperature state, consistently outperforming the baselines and confirming that the control mechanism adapts well to different initial dynamics.

Table 9: Ablation results on Annealing with $\gamma = 2.0$ from Base Temperature $T \in \{1.0, 1.5\}$. Results are mean$_{\pm\text{std}}$ over 5 runs. Best results per column (within each $T$ block) are in bold.

| Method | $T = 1.0$ | | | | | $T = 1.5$ | | | | |
|---|---|---|---|---|---|---|---|---|---|---|
| | $\Delta$NLL | MMD | SWD | $W_1^{\text{RDF}}$ | $W_1^{\mathcal{E}}$ | $\Delta$NLL | MMD | SWD | $W_1^{\text{RDF}}$ | $W_1^{\mathcal{E}}$ |
| PG | $-0.219_{\pm 0.782}$ | $0.343_{\pm 0.195}$ | $0.833_{\pm 0.405}$ | $0.194_{\pm 0.006}$ | $\mathbf{0.054_{\pm 0.007}}$ | $0.546_{\pm 0.731}$ | $0.259_{\pm 0.084}$ | $0.690_{\pm 0.226}$ | $0.217_{\pm 0.006}$ | $0.320_{\pm 0.013}$ |
| G-SMC | $2.571_{\pm 1.979}$ | $0.359_{\pm 0.163}$ | $0.973_{\pm 0.445}$ | $0.147_{\pm 0.030}$ | $0.201_{\pm 0.121}$ | $2.645_{\pm 1.065}$ | $0.639_{\pm 0.185}$ | $1.681_{\pm 0.527}$ | $0.116_{\pm 0.133}$ | $0.325_{\pm 0.182}$ |
| VCG | $-0.046_{\pm 0.093}$ | $0.026_{\pm 0.009}$ | $0.076_{\pm 0.031}$ | $\mathbf{0.045_{\pm 0.007}}$ | $0.127_{\pm 0.018}$ | $0.083_{\pm 0.121}$ | $0.032_{\pm 0.015}$ | $0.091_{\pm 0.048}$ | $0.034_{\pm 0.004}$ | $0.558_{\pm 0.031}$ |
| **VCG-SMC** | $\mathbf{-0.044_{\pm 0.029}}$ | $\mathbf{0.019_{\pm 0.005}}$ | $\mathbf{0.048_{\pm 0.013}}$ | $0.053_{\pm 0.012}$ | $\mathbf{0.053_{\pm 0.022}}$ | $\mathbf{0.077_{\pm 0.010}}$ | $\mathbf{0.015_{\pm 0.007}}$ | $\mathbf{0.035_{\pm 0.011}}$ | $\mathbf{0.011_{\pm 0.008}}$ | $\mathbf{0.078_{\pm 0.010}}$ |

**Ablation on Annealing Factor $\gamma$.** We further test the methods with varying annealing strengths ($\gamma \in \{1.5, 2.0, 2.5\}$) for both the DW-4 and LJ-13 systems. For the DW-4 system (Figs. 13 and 14 and Tab. 10), the visual and quantitative results confirm that VCG and VCG-SMC maintain high accuracy even as $\gamma$ increases. For the more complex LJ-13 system (Fig. 15 and Tab. 11), the challenge is greater. While all methods struggle with the most challenging annealing tasks, VCG-SMC consistently provides the most physically plausible results, capturing the structural features (RDF) and energy distributions far more accurately than competing methods. This highlights its superior performance in complex, high-dimensional energy landscapes.

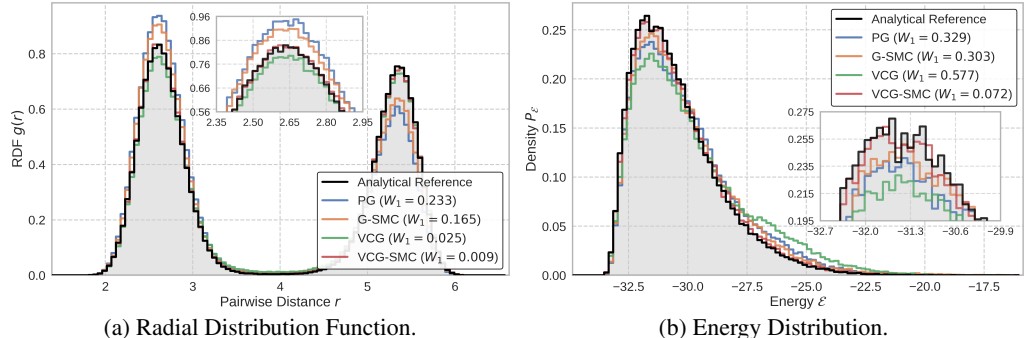

(a) Radial Distribution Function.
(b) Energy Distribution.

Figure 13: Comparison of generated distributions for the DW-4 annealing task ($\gamma = 2.0$): (a) Radial Distribution Function and (b) Energy Distribution.

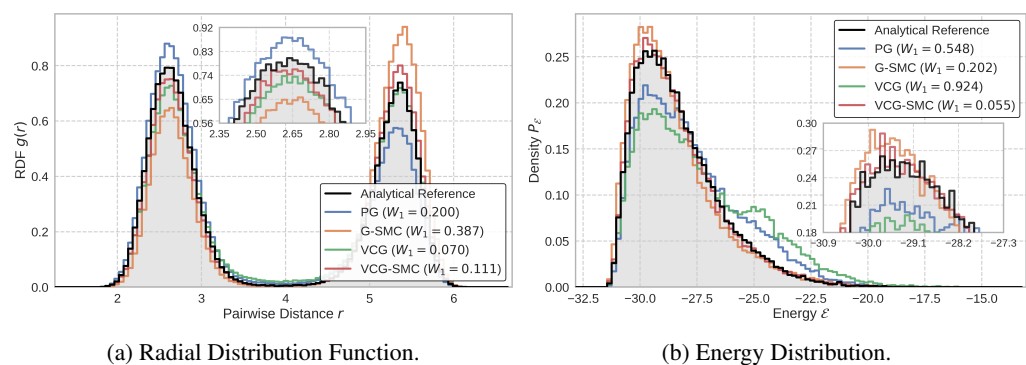

(a) Radial Distribution Function.
(b) Energy Distribution.

Figure 14: Comparison of generated distributions for the DW-4 annealing task ($\gamma = 2.5$): (a) Radial Distribution Function and (b) Energy Distribution.

Table 10: Performance ablation for the DW-4 annealing task ($T = 2.0$) with varying annealing factor $\gamma$. Results are mean$_{\pm\text{std}}$ over 5 runs. Best results per column (within each $\gamma$ block) are in bold.

| Method | Annealing ($\gamma = 1.5$) | | | | | Annealing ($\gamma = 2.5$) | | | | |
|---|---|---|---|---|---|---|---|---|---|---|
| | $\Delta$NLL | MMD | SWD | $W_1^{\text{RDF}}$ | $W_1^{\mathcal{E}}$ | $\Delta$NLL | MMD | SWD | $W_1^{\text{RDF}}$ | $W_1^{\mathcal{E}}$ |
| PG | -0.114$_{\pm0.339}$ | 0.198$_{\pm0.152}$ | 0.527$_{\pm0.396}$ | 0.132$_{\pm0.002}$ | 0.179$_{\pm0.023}$ | 0.159$_{\pm1.232}$ | 0.400$_{\pm0.168}$ | 1.088$_{\pm0.384}$ | 0.208$_{\pm0.008}$ | 0.551$_{\pm0.009}$ |
| G-SMC | -0.009$_{\pm0.041}$ | 0.053$_{\pm0.009}$ | 0.143$_{\pm0.031}$ | 0.038$_{\pm0.023}$ | 0.026$_{\pm0.011}$ | 0.038$_{\pm0.338}$ | 0.365$_{\pm0.058}$ | 1.012$_{\pm0.253}$ | 0.208$_{\pm0.146}$ | 0.190$_{\pm0.080}$ |
| **VCG** | **-0.009$_{\pm0.018}$** | 0.010$_{\pm0.001}$ | 0.024$_{\pm0.003}$ | 0.041$_{\pm0.003}$ | 0.307$_{\pm0.020}$ | -0.065$_{\pm0.041}$ | 0.024$_{\pm0.003}$ | 0.061$_{\pm0.013}$ | **0.065$_{\pm0.008}$** | 0.931$_{\pm0.024}$ |
| **VCG-SMC** | **-0.013$_{\pm0.017}$** | **0.009$_{\pm0.001}$** | **0.022$_{\pm0.002}$** | **0.037$_{\pm0.004}$** | **0.020$_{\pm0.006}$** | **-0.060$_{\pm0.008}$** | **0.023$_{\pm0.001}$** | **0.056$_{\pm0.006}$** | 0.119$_{\pm0.008}$ | **0.059$_{\pm0.008}$** |

Table 11: Performance ablation for the LJ-13 annealing task ($T = 1.0$) with varying annealing factor $\gamma$. Results are mean$_{\pm\text{std}}$ over 5 runs. Best results per column (within each $\gamma$ block) are in bold.

| Method | Annealing ($\gamma = 1.5$) | | | | | Annealing ($\gamma = 2.0$) | | | | |
|---|---|---|---|---|---|---|---|---|---|---|
| | $\Delta$NLL | MMD | SWD | $W_1^{\text{RDF}}$ | $W_1^{\mathcal{E}}$ | $\Delta$NLL | MMD | SWD | $W_1^{\text{RDF}}$ | $W_1^{\mathcal{E}}$ |
| PG | **1.224$_{\pm3.148}$** | 0.718$_{\pm0.021}$ | 0.806$_{\pm0.056}$ | 0.026$_{\pm0.001}$ | **1.855$_{\pm0.050}$** | 15.75$_{\pm10.56}$ | 0.622$_{\pm0.056}$ | 0.639$_{\pm0.150}$ | 0.053$_{\pm0.001}$ | 12.73$_{\pm0.193}$ |
| G-SMC | 3.289$_{\pm1.461}$ | 0.220$_{\pm0.112}$ | 0.190$_{\pm0.071}$ | 0.022$_{\pm0.007}$ | 3.517$_{\pm1.085}$ | 0.927$_{\pm2.384}$ | 0.416$_{\pm0.142}$ | 0.423$_{\pm0.171}$ | 0.017$_{\pm0.014}$ | 4.253$_{\pm3.053}$ |
| **VCG** | 1.642$_{\pm0.316}$ | 0.027$_{\pm0.007}$ | 0.029$_{\pm0.005}$ | 0.058$_{\pm0.001}$ | 8.848$_{\pm0.130}$ | 1.191$_{\pm1.499}$ | 0.124$_{\pm0.067}$ | 0.135$_{\pm0.076}$ | 0.088$_{\pm0.001}$ | 21.28$_{\pm0.127}$ |
| **VCG-SMC** | 2.221$_{\pm0.353}$ | **0.024$_{\pm0.004}$** | **0.024$_{\pm0.005}$** | **0.018$_{\pm0.002}$** | 3.006$_{\pm0.307}$ | **0.734$_{\pm0.490}$** | **0.091$_{\pm0.023}$** | **0.092$_{\pm0.021}$** | **0.007$_{\pm0.002}$** | **1.958$_{\pm0.610}$** |

**Ablation on Reward Strength $\lambda'$.** Finally, we perform an ablation on the reward strength ($\lambda' \in \{0.2, 0.5, 0.8\}$) for the reward-tilting task on both particle systems. The results, shown in Tab. 12 for DW-4 and Tab. 13 for LJ-13, are consistent with previous findings. As the reward strength increases, making the target distribution more distinct from the base distribution, the performance of the baseline methods deteriorates rapidly. In contrast, VCG-SMC maintains excellent performance,

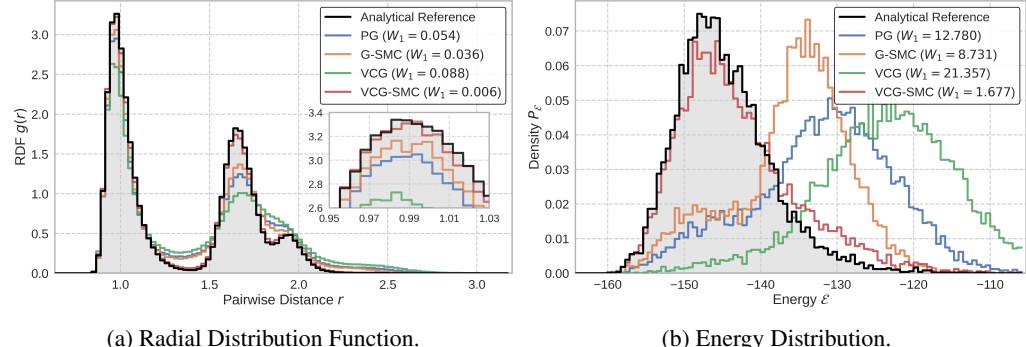

(a) Radial Distribution Function.

(b) Energy Distribution.

Figure 15: Comparison of generated distributions for the LJ-13 annealing task ($\gamma = 2.0$): (a) Radial Distribution Function and (b) Energy Distribution.

demonstrating its capability to accurately steer the particle distribution toward a sharply defined target region. This confirms the effectiveness and robustness of our drift control mechanism across a wide range of inference-time scaling challenges.

Table 12: Performance ablation for the DW-4 reward-tilting task ($T = 2.0$) with varying reward strength $\lambda'$. Results are mean$_{\pm \text{std}}$ over 5 runs. Best results per column (within each $\lambda'$ block) are in bold.

| Method | Reward-Tilting ($\lambda' = 0.2$) | | | | | Reward-Tilting ($\lambda' = 0.8$) | | | | |
|---|---|---|---|---|---|---|---|---|---|---|
| | $\Delta$NLL | MMD | SWD | $W_1^{\text{RDF}}$ | $W_1^{\mathcal{E}}$ | $\Delta$NLL | MMD | SWD | $W_1^{\text{RDF}}$ | $W_1^{\mathcal{E}}$ |
| PG | -0.182$_{\pm 0.846}$ | 0.768$_{\pm 0.078}$ | 1.808$_{\pm 0.223}$ | 0.305$_{\pm 0.006}$ | 0.368$_{\pm 0.008}$ | 2.480$_{\pm 2.35}$ | 0.775$_{\pm 0.088}$ | 1.647$_{\pm 0.239}$ | 0.852$_{\pm 0.004}$ | 3.867$_{\pm 0.010}$ |
| G-SMC | **0.094$_{\pm 0.015}$** | 0.020$_{\pm 0.003}$ | 0.046$_{\pm 0.007}$ | 0.060$_{\pm 0.004}$ | 0.094$_{\pm 0.019}$ | 0.534$_{\pm 0.088}$ | 0.079$_{\pm 0.025}$ | 0.160$_{\pm 0.043}$ | 0.150$_{\pm 0.022}$ | 0.536$_{\pm 0.088}$ |
| VCG | 0.386$_{\pm 1.000}$ | 0.403$_{\pm 0.098}$ | 1.122$_{\pm 0.359}$ | 0.095$_{\pm 0.008}$ | 0.138$_{\pm 0.012}$ | 1.050$_{\pm 2.865}$ | 0.671$_{\pm 0.087}$ | 1.768$_{\pm 0.302}$ | 0.282$_{\pm 0.121}$ | 1.162$_{\pm 0.567}$ |
| VCG-SMC | **0.096$_{\pm 0.017}$** | **0.014$_{\pm 0.001}$** | **0.032$_{\pm 0.004}$** | **0.059$_{\pm 0.004}$** | **0.093$_{\pm 0.017}$** | **0.518$_{\pm 0.020}$** | **0.028$_{\pm 0.001}$** | **0.065$_{\pm 0.004}$** | **0.149$_{\pm 0.004}$** | **0.520$_{\pm 0.020}$** |

Table 13: Performance ablation for the LJ-13 reward-tilting task ($T = 1.0$) with varying reward strength $\lambda'$. Results are mean$_{\pm \text{std}}$ over 5 runs. Best results per column (within each $\lambda'$ block) are in bold.

| Method | Reward-Tilting ($\lambda' = 0.2$) | | | | | Reward-Tilting ($\lambda' = 0.5$) | | | | |
|---|---|---|---|---|---|---|---|---|---|---|
| | $\Delta$NLL | MMD | SWD | $W_1^{\text{RDF}}$ | $W_1^{\mathcal{E}}$ | $\Delta$NLL | MMD | SWD | $W_1^{\text{RDF}}$ | $W_1^{\mathcal{E}}$ |
| PG | 1.404$_{\pm 4.812}$ | 0.718$_{\pm 0.021}$ | 0.806$_{\pm 0.056}$ | 0.026$_{\pm 0.001}$ | 1.855$_{\pm 0.050}$ | 3.253$_{\pm 5.013}$ | 0.718$_{\pm 0.020}$ | 0.800$_{\pm 0.059}$ | 0.050$_{\pm 0.001}$ | 3.705$_{\pm 0.085}$ |
| G-SMC | 0.832$_{\pm 0.072}$ | 0.031$_{\pm 0.009}$ | 0.030$_{\pm 0.013}$ | **0.010$_{\pm 0.001}$** | **0.834$_{\pm 0.072}$** | 1.277$_{\pm 0.072}$ | 0.055$_{\pm 0.006}$ | 0.049$_{\pm 0.008}$ | **0.016$_{\pm 0.001}$** | 1.278$_{\pm 0.072}$ |
| VCG | **0.106$_{\pm 3.619}$** | 0.559$_{\pm 0.127}$ | 0.588$_{\pm 0.169}$ | 0.024$_{\pm 0.015}$ | 1.869$_{\pm 1.046}$ | 7.470$_{\pm 10.167}$ | 0.669$_{\pm 0.132}$ | 0.675$_{\pm 0.167}$ | 0.024$_{\pm 0.015}$ | 1.869$_{\pm 1.046}$ |
| VCG-SMC | 0.765$_{\pm 0.117}$ | **0.012$_{\pm 0.001}$** | **0.015$_{\pm 0.002}$** | 0.016$_{\pm 0.001}$ | 1.219$_{\pm 0.079}$ | **1.218$_{\pm 0.079}$** | **0.013$_{\pm 0.001}$** | **0.015$_{\pm 0.002}$** | **0.016$_{\pm 0.001}$** | **1.219$_{\pm 0.079}$** |

### C.3 ADDITIONAL EXPERIMENTAL RESULTS OF PROTEIN-LIGAND CO-FOLDING

In Fig. 16, we show representative trajectories of the normalized Effective Sample Size (ESS) for three protein–ligand complexes (PDB IDs 7n7h, 7zu2, and 8d39). Even with a relatively small particle budget, our method (VCG-SMC) maintains substantially higher ESS than the baseline G-SMC throughout the middle and late stages of the trajectory, indicating that the controlled drift keeps a much larger fraction of particles effective. We expect this advantage to become even more pronounced as the number of particles increases.

Table 14 shows that both particle-based methods are about $3.2\times$ slower than the unsteered base sampler, reflecting the cost of propagating and reweighting a small ensemble of particles. Importantly, VCG-SMC incurs essentially no additional wall-clock overhead compared with the G-SMC baseline (31.78s vs 31.67s per complex), while delivering consistently higher ESS and improved physical validity on PoseBusters. Thus, the stability gains from DriftLite come at negligible extra cost beyond a standard SMC correction.

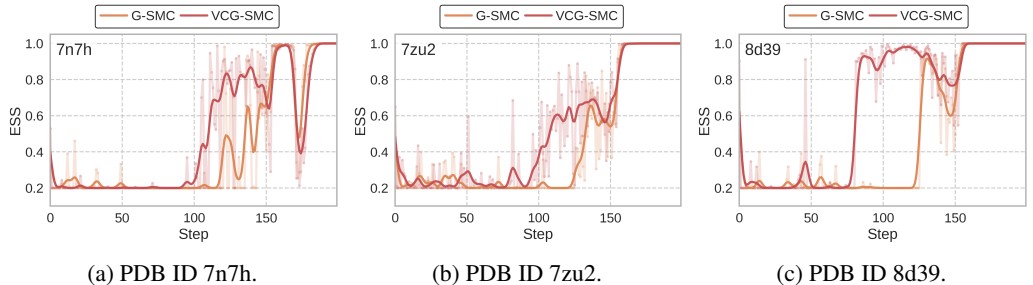

Figure 16: Evolution of ESS during inference on the protein–ligand co-folding task. The curves are smoothed using a Gaussian kernel over time, with the unsmoothed traces shown in the background for reference.

Table 14: Elapsed time per generated protein–ligand complex for different sampling methods, averaged over a random subset of 20 structures. The relative runtime is computed with respect to the G-SMC method.

| Method | Runtime (s) | Relative Runtime |
|---|---|---|
| Base | 10.05 | $0.32\times$ |
| G-SMC | 31.67 | $1.00\times$ |
| **VCG-SMC** | 31.78 | $1.00\times$ |

## C.4 ITERATIVE REFINEMENT

While a single pass of inference-time scaling is effective, its corrective power is finite; once the simulation reaches the terminal time, any residual mismatch between the particle and target distributions cannot be further addressed. The key insight behind our iterative approach is that each pass produces an improved sampling dynamic. The updated drift and uncentered potential:

$$\widetilde{\boldsymbol{v}}_t^{\text{eff}}(\boldsymbol{x}) = \widetilde{\boldsymbol{v}}_t(\boldsymbol{x}) + \boldsymbol{b}_t(\boldsymbol{x}), \quad H_t^{\text{eff}}(\boldsymbol{x}) = G_t(\boldsymbol{x}) + h_t(\boldsymbol{x}; \boldsymbol{b}_t),$$

encode richer information about the target distribution after centering $H_t^{\text{eff}}$ with the current particle weights. Our iterative refinement procedure leverages this accumulated knowledge rather than discarding it. By iterating the procedure, each round builds upon the refined dynamics of the previous one, creating a virtuous cycle that progressively sharpens the sampling path.

This iterative process can also be viewed as a practical method for approaching the optimal control drift in Prop. 3.2. Since the control $\boldsymbol{b}_t$ is computed at each step under a linear ansatz (Thms. 3.3 and 3.4), it provides an approximation of the true optimal control. This approximation becomes increasingly accurate as the underlying dynamics and particle distribution are improved in each round, allowing the linear model to operate on a better-conditioned problem.

**Algorithm.** The iterative refinement algorithm, detailed in Alg. 2, transforms the single-pass method (Alg. 1) into a multi-stage process of progressive improvement. The fundamental difference lies in the cumulative application of control. Whereas the base algorithm applies a calculated control drift just once, the iterative method repeats the entire simulation $K$ times.

The central mechanism is the permanent absorption of the learned control into the system's dynamics. After each full trajectory simulation (a "round"), the control terms are folded into an effective guidance drift $\widetilde{\boldsymbol{v}}_t^{\text{eff}}$ and an effective uncentered potential $H_t^{\text{eff}}$. While particle positions are reset to the initial noise distribution at the start of a new round, these effective dynamics are preserved and carried forward. The refined dynamics from round $j$ thus serve as the improved baseline for round $j + 1$. This process iteratively sharpens the sampling path, guiding particles more efficiently in subsequent rounds without any external training or global optimization.

**Results.** Our empirical results decisively validate the iterative refinement strategy on a Gaussian Mixture Model (GMM) target across both annealing and reward-tilting tasks.

---

**Algorithm 2:** Iterative Refinement for Inference-Time Scaling (cumulative drift/potential updates)

---

**Input:** Guidance drift path $\widetilde{v}_t$ from (2.4), uncentered potential path $G_t$ from Prop. 2.1, time steps $\{t_k\}_{k=0}^{M}$, reward $r(\boldsymbol{x})$, schedule $\beta_t$, basis functions, number of refinement rounds $K$, number of particles $N$, ESS threshold $\tau$.

1   Initialize effective drift $\widetilde{v}_t^{\mathrm{eff}}(\cdot) \leftarrow \widetilde{v}_t(\cdot)$, and effective uncentered potential $H_t^{\mathrm{eff}}(\cdot) \leftarrow G_t(\cdot)$;

2   **for** $j \leftarrow 1$ **to** $K$ **do**

3      Initialize particles $\boldsymbol{x}_{t_0}^{(i)} \sim \overleftarrow{p}_0$, log-weights $\ell_{t_0}^{(i)} \leftarrow 0$, and weights $w_{t_0}^{(i)} \leftarrow \frac{1}{N}$ for $i = 1, \ldots, N$;

4      **for** $k \leftarrow 0$ **to** $M - 1$ **do**

5          Set $\Delta t_k \leftarrow t_{k+1} - t_k$ and form estimates of $\boldsymbol{A}_{t_k}$ and $\boldsymbol{c}_{t_k}$ using $\{(\boldsymbol{x}_{t_k}^{(i)}, w_{t_k}^{(i)})\}_{i\in[N]}$;

6          Solve $\boldsymbol{A}_{t_k} \boldsymbol{\theta}_{t_k} = \boldsymbol{c}_{t_k}$ to obtain the control drift $\boldsymbol{b}_{t_k}(\cdot)$;

7          $\widetilde{v}_{t_k}^{\mathrm{eff}}(\cdot) \leftarrow \widetilde{v}_{t_k}^{\mathrm{eff}}(\cdot) + \boldsymbol{b}_{t_k}(\cdot), \quad H_{t_k}^{\mathrm{eff}}(\cdot) \leftarrow H_{t_k}^{\mathrm{eff}}(\cdot) + h_{t_k}(\cdot; \boldsymbol{b}_{t_k})$;

8          $\widehat{g}_{t_k}^{\mathrm{eff}}(\boldsymbol{x}_{t_k}^{(i)}) \leftarrow H_{t_k}^{\mathrm{eff}}(\boldsymbol{x}_{t_k}^{(i)}) - \sum_{m=1}^{N} w_{t_k}^{(m)} H_{t_k}^{\mathrm{eff}}(\boldsymbol{x}_{t_k}^{(m)})$;

9          $\ell_{t_{k+1}}^{(i)} \leftarrow \ell_{t_k}^{(i)} + \widehat{g}_{t_k}^{\mathrm{eff}}(\boldsymbol{x}_{t_k}^{(i)}) \Delta t_k, \quad \boldsymbol{w}_{t_{k+1}} \leftarrow \mathrm{softmax}(\boldsymbol{\ell}_{t_{k+1}})$;

10         $\boldsymbol{x}_{t_{k+1}}^{(i)} \leftarrow \boldsymbol{x}_{t_k}^{(i)} + \widetilde{v}_{t_k}^{\mathrm{eff}}(\boldsymbol{x}_{t_k}^{(i)}) \Delta t_k + V_{t_k} \sqrt{\Delta t_k}\, \boldsymbol{z}^{(i)}$, where $\boldsymbol{z}^{(i)} \sim \mathcal{N}(0, \boldsymbol{I})$;

11         **if** $ESS(\boldsymbol{w}_{t_{k+1}}) < \tau$ *or periodically* **then**

12            Resample $\{\boldsymbol{x}_{t_{k+1}}^{(i)}\}_{i\in[N]}$ according to $\{w_{t_{k+1}}^{(i)}\}_{i\in[N]}$;

13            Reset $\ell_{t_{k+1}}^{(i)} \leftarrow 0, w_{t_{k+1}}^{(i)} \leftarrow \frac{1}{N}$ for all $i$;

**Output:** Final samples $\{(\boldsymbol{x}_T^{(i)}, w_T^{(i)})\}_{i\in[N]}$ from the $K$-th refinement round.

---

The evolution of the sampler's internal state, shown in Figs. 17 and 18, provides clear evidence of the algorithm's success. With each successive refinement round (progressing from blue to red), the variance of the control estimates exhibits a striking, monotonic decrease. This variance reduction directly leads to a dramatically more stable ESS, which is maintained at near-optimal levels throughout the later stages of the simulation. This provides direct confirmation that the refined dynamics act as a substantially more efficient proposal distribution, a trend that holds robustly across both experimental setups.

The final sample quality metrics, reported in Tabs. 15 and 16, complete the picture. While the improvement in downstream metrics, such as MMD and SWD, is not always strictly monotonic with every iteration, the refinement process consistently uncovers solutions that are superior to those from the initial pass. The best-performing configurations (highlighted in bold) are frequently found in later iterations. This confirms that multiple refinement passes are invaluable for navigating the optimization landscape to find higher-quality final samples.

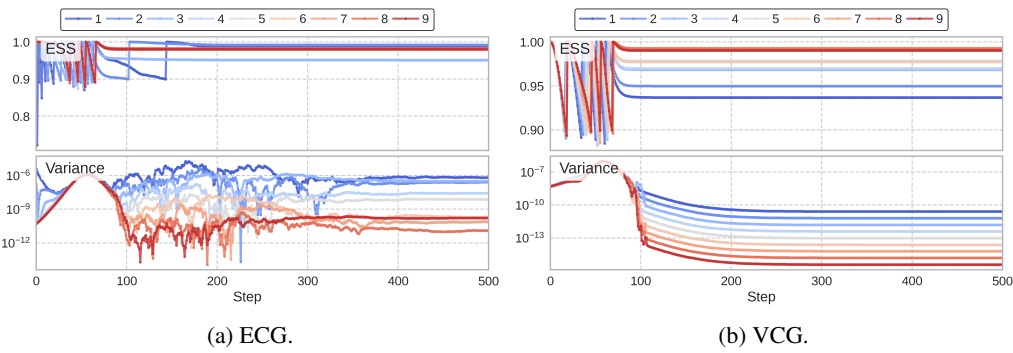

(a) ECG.        (b) VCG.

Figure 17: Evolution of ESS and potential variance across multiple refinement rounds for the GMM annealing task ($\gamma = 2.5$). Later rounds (red) show monotonically decreasing variance and more stable ESS.

Table 15: Iterative refinement performance on the GMM annealing task ($\gamma = 3.0$). Results are mean$_{\pm\text{std}}$ over 5 runs. Best result per column is in bold.

| Iter. | VCG-SMC | | | | | ECG-SMC | | | | |
|---|---|---|---|---|---|---|---|---|---|---|
| | $\Delta$NLL | MMD | SWD | Mean $L_2$ | Cov F $_{\times 10^3}$ | $\Delta$NLL | MMD | SWD | Mean $L_2$ | Cov F $_{\times 10^3}$ |
| 1 | $0.174_{\pm 0.073}$ | $0.019_{\pm 0.001}$ | $0.691_{\pm 0.149}$ | $3.319_{\pm 0.593}$ | $0.415_{\pm 0.048}$ | $0.184_{\pm 0.079}$ | $0.031_{\pm 0.003}$ | $1.672_{\pm 0.214}$ | $7.795_{\pm 1.164}$ | $0.850_{\pm 0.107}$ |
| 2 | $0.159_{\pm 0.068}$ | $0.019_{\pm 0.001}$ | $0.797_{\pm 0.151}$ | $3.906_{\pm 0.806}$ | $0.450_{\pm 0.055}$ | $0.178_{\pm 0.058}$ | $0.028_{\pm 0.002}$ | $1.105_{\pm 0.129}$ | $5.748_{\pm 0.805}$ | $0.787_{\pm 0.091}$ |
| 3 | $0.176_{\pm 0.022}$ | $0.019_{\pm 0.002}$ | $\mathbf{0.599_{\pm 0.142}}$ | $\mathbf{3.083_{\pm 0.794}}$ | $0.406_{\pm 0.078}$ | $0.166_{\pm 0.015}$ | $0.027_{\pm 0.002}$ | $1.201_{\pm 0.267}$ | $5.990_{\pm 0.866}$ | $0.762_{\pm 0.090}$ |
| 4 | $\mathbf{0.152_{\pm 0.078}}$ | $0.019_{\pm 0.001}$ | $0.720_{\pm 0.061}$ | $3.346_{\pm 0.545}$ | $0.414_{\pm 0.066}$ | $0.168_{\pm 0.059}$ | $0.029_{\pm 0.003}$ | $1.696_{\pm 0.455}$ | $6.653_{\pm 1.728}$ | $0.816_{\pm 0.129}$ |
| 5 | $0.165_{\pm 0.052}$ | $\mathbf{0.018_{\pm 0.001}}$ | $0.694_{\pm 0.167}$ | $3.124_{\pm 0.765}$ | $\mathbf{0.386_{\pm 0.061}}$ | $\mathbf{0.158_{\pm 0.058}}$ | $\mathbf{0.025_{\pm 0.002}}$ | $\mathbf{1.096_{\pm 0.386}}$ | $\mathbf{4.854_{\pm 0.929}}$ | $\mathbf{0.693_{\pm 0.082}}$ |
| 6 | $0.163_{\pm 0.073}$ | $0.019_{\pm 0.001}$ | $0.744_{\pm 0.077}$ | $3.378_{\pm 0.409}$ | $0.412_{\pm 0.022}$ | $0.164_{\pm 0.074}$ | $0.031_{\pm 0.007}$ | $1.494_{\pm 0.572}$ | $7.135_{\pm 2.767}$ | $0.908_{\pm 0.302}$ |
| 7 | $0.161_{\pm 0.050}$ | $0.019_{\pm 0.001}$ | $0.736_{\pm 0.095}$ | $3.343_{\pm 0.285}$ | $0.408_{\pm 0.027}$ | $0.160_{\pm 0.055}$ | $0.031_{\pm 0.005}$ | $1.477_{\pm 0.354}$ | $7.010_{\pm 1.843}$ | $0.874_{\pm 0.151}$ |
| 8 | $0.180_{\pm 0.055}$ | $0.019_{\pm 0.001}$ | $0.766_{\pm 0.120}$ | $3.407_{\pm 0.269}$ | $0.420_{\pm 0.043}$ | $0.181_{\pm 0.052}$ | $0.029_{\pm 0.003}$ | $1.406_{\pm 0.184}$ | $6.748_{\pm 1.192}$ | $0.806_{\pm 0.119}$ |
| 9 | $0.184_{\pm 0.050}$ | $0.019_{\pm 0.001}$ | $0.670_{\pm 0.117}$ | $3.238_{\pm 0.371}$ | $0.395_{\pm 0.046}$ | $0.187_{\pm 0.046}$ | $0.027_{\pm 0.004}$ | $1.137_{\pm 0.247}$ | $5.669_{\pm 1.519}$ | $0.739_{\pm 0.201}$ |

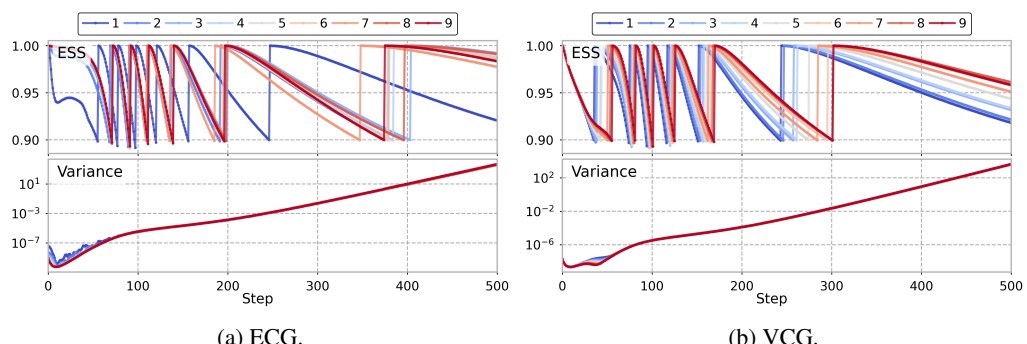

(a) ECG.      (b) VCG.

Figure 18: Evolution of ESS and potential variance across multiple refinement rounds for the GMM reward-tilting task ($\sigma = 100.0$). Later rounds (red) show monotonically decreasing variance.

Table 16: Iterative refinement performance on the GMM reward-tilting task ($\sigma = 100.0$). Results are mean$_{\pm\text{std}}$ over 5 runs. Best result per column is in bold.

| Iter. | VCG-SMC | | | | | ECG-SMC | | | | |
|---|---|---|---|---|---|---|---|---|---|---|
| | $\Delta$NLL | MMD | SWD | Mean $L_2$ | Cov F $_{\times 10^3}$ | $\Delta$NLL | MMD | SWD | Mean $L_2$ | Cov F $_{\times 10^3}$ |
| 1 | $0.240_{\pm 0.051}$ | $0.016_{\pm 0.001}$ | $0.177_{\pm 0.042}$ | $0.701_{\pm 0.267}$ | $0.041_{\pm 0.025}$ | $0.292_{\pm 0.042}$ | $\mathbf{0.016_{\pm 0.002}}$ | $0.177_{\pm 0.028}$ | $0.818_{\pm 0.218}$ | $0.043_{\pm 0.018}$ |
| 2 | $0.262_{\pm 0.020}$ | $\mathbf{0.016_{\pm 0.001}}$ | $\mathbf{0.140_{\pm 0.021}}$ | $\mathbf{0.634_{\pm 0.182}}$ | $\mathbf{0.031_{\pm 0.015}}$ | $\mathbf{0.218_{\pm 0.070}}$ | $0.018_{\pm 0.002}$ | $0.196_{\pm 0.058}$ | $1.103_{\pm 0.768}$ | $0.046_{\pm 0.019}$ |
| 3 | $0.277_{\pm 0.068}$ | $0.017_{\pm 0.002}$ | $0.168_{\pm 0.048}$ | $0.985_{\pm 0.644}$ | $0.039_{\pm 0.015}$ | $0.224_{\pm 0.061}$ | $0.017_{\pm 0.001}$ | $0.169_{\pm 0.040}$ | $0.823_{\pm 0.292}$ | $0.045_{\pm 0.019}$ |
| 4 | $0.244_{\pm 0.058}$ | $0.016_{\pm 0.002}$ | $0.198_{\pm 0.085}$ | $0.909_{\pm 0.494}$ | $0.061_{\pm 0.038}$ | $0.305_{\pm 0.066}$ | $0.018_{\pm 0.002}$ | $0.194_{\pm 0.046}$ | $0.880_{\pm 0.205}$ | $0.053_{\pm 0.024}$ |
| 5 | $\mathbf{0.226_{\pm 0.045}}$ | $0.017_{\pm 0.002}$ | $0.166_{\pm 0.055}$ | $0.779_{\pm 0.267}$ | $0.045_{\pm 0.025}$ | $0.257_{\pm 0.054}$ | $0.017_{\pm 0.002}$ | $0.181_{\pm 0.049}$ | $0.870_{\pm 0.290}$ | $0.046_{\pm 0.024}$ |
| 6 | $0.244_{\pm 0.053}$ | $0.017_{\pm 0.001}$ | $0.184_{\pm 0.058}$ | $1.031_{\pm 0.646}$ | $0.054_{\pm 0.033}$ | $0.257_{\pm 0.055}$ | $0.017_{\pm 0.002}$ | $\mathbf{0.152_{\pm 0.038}}$ | $0.750_{\pm 0.221}$ | $\mathbf{0.039_{\pm 0.016}}$ |
| 7 | $0.312_{\pm 0.060}$ | $0.018_{\pm 0.002}$ | $0.207_{\pm 0.067}$ | $1.133_{\pm 0.645}$ | $0.060_{\pm 0.023}$ | $0.290_{\pm 0.023}$ | $0.017_{\pm 0.002}$ | $0.162_{\pm 0.051}$ | $0.876_{\pm 0.321}$ | $0.049_{\pm 0.023}$ |
| 8 | $0.251_{\pm 0.072}$ | $0.018_{\pm 0.003}$ | $0.224_{\pm 0.106}$ | $1.363_{\pm 1.113}$ | $0.059_{\pm 0.038}$ | $0.304_{\pm 0.052}$ | $0.017_{\pm 0.002}$ | $0.171_{\pm 0.031}$ | $\mathbf{0.712_{\pm 0.109}}$ | $0.047_{\pm 0.024}$ |
| 9 | $0.244_{\pm 0.046}$ | $0.017_{\pm 0.002}$ | $0.192_{\pm 0.045}$ | $0.885_{\pm 0.240}$ | $0.068_{\pm 0.032}$ | $0.297_{\pm 0.100}$ | $0.018_{\pm 0.003}$ | $0.250_{\pm 0.077}$ | $1.286_{\pm 0.405}$ | $0.095_{\pm 0.042}$ |

