# OpenReview forum: "DriftLite: Lightweight Drift Control for Inference-Time Scaling of Diffusion Models"
_ICLR.cc/2026/Conference — ICLR 2026 Poster_

### Official Review · Reviewer_cAK7 · 2025-10-25

**Soundness:** 2
**Presentation:** 2
**Contribution:** 3
**Rating:** 4
**Confidence:** 3

**Summary:**

The paper extends previous work on adapting diffusion models at inference-time to new tasks using Sequential Monte Carlo (SMC) methods, focusing on reward-tilting and sampling from an annealed version of the base model distribution. In particular, they target the problem of sampling from the sequence of distributions $q_t(x) = p_t(x)^\gamma \exp(r_t(x))$, where $p_t(x)$ is the original diffusion model distribution and $r_t(x)$ is some time-varying reward function, such that at $t=0$ we approach some predefined clean data distribution $p_0(x)^\gamma \exp(r(x))$. Following previous work, $N$ particles are sampled concurrently for each $t$, and the diffusion model is used to move these particles towards lower $t$ in the $q_t(x)$ chain. The resulting proposal particles are reweighted using Feynman-Kac-type equations and periodically resampled, enabling the particle ensemble to approximate the sequence of target distributions.

The new idea in the paper to analyze the Feynman-Kac-type equations and notice that there is additional freedom in choosing the proposal distributions, such that they can add an additional term to the drift in the Feynman-Kac equations if they correspondingly change the time evolution in the weights. The authors then show that there exists an optimal added drift such that it is the gradient of a scalar potential that follows a particular PDE. With this added design space, they propose two choices for the added drift: 1) Minimizing the variance in the change of the weights 2) minimizing a Lagrangian of the aforementioned PDE such that at the minimum, the optimal drift is obtained. In practice, they parameterize the added drift as a linear combination of $\nabla r_t(x), \nabla \log p_t(x)$, and the original diffusion drift $u_t(x)$ for the variance-minimizing version, and a similar choice for the PDE-optimizing version. This results in a simple $3x3$ systems of equations per particle per step, resulting in a relatively lightweight correction, although evaluating the required matrices requires backpropagation through the diffusion denoiser.

The authors evaluate the method on a 30-dimensional Gaussian Mixture Model, two toy particle systems, and on a protein-ligand co-folding task. They notice that the method consistently improves effective sample size and reduces weight variance, outperforming the naive SMC baseline. On the protein-ligand task, they demonstrate that using physical energies as the rewards, the method can improve the physical validity of the generated samples.

**Strengths:**

- The core idea of defining a correction to the proposal distribution that directly optimizes the variance or consistency with the target distributions, is principled and seems to be novel at least in the diffusion context. It seems to be quite simple to implement, does not cause a huge overhead, and seems to legitimately improve the performance of the SMC method considerably on the tasks chosen.
- The experiments on the GMM, DW-4 and LJ13 are quite thorough and well-presented, and the protein-ligand task is an example of a realistic problem that the method could help with and the method clearly works better than more naive SMC approaches.
- Overall, the paper is well written and not too difficult to follow, with the caveat that the mathematical exposition may be difficult to follow to readers that are not familiar with this way of describing SMC methods in advance.

**Weaknesses:**

Disclaimer: I am not an expert on SMC or SMC-based methods for diffusion models, and as such it is possible that I am missing some details, wider context or some previous literature. As such, I am open to being corrected.

- The method incurs significant cost on the wall-clock time compared to the baselines, 2.3x-6x in the experiments for which the wall-clock time was reported. This is not mentioned in the limitations or in the main paper, however, and the conclusion instead states that the method causes minimal computational overhead, potentially misleading the reader or a reviewer who does not have the time to read through the Appendix. Backpropagation through the denoiser at each step requires much more memory than the forward pass and, and my assumption is that it may limit the batch size in practice.
- As such, it might be more fair to compare to the baselines by adjusting the inference hyperparameters such that the wall-clock time is equal. E.g., I would expect that the G-SMC method improves with more particles, and potentially with more diffusion steps.
- Following that, I would be interested in seeing the ESS and wall-clock time comparisons for the protein-ligand task as well, especially since this seems to be an example of a higher-dimensional case. As an alternative, it would also be interesting to see how the method scales, e.g., to the LJ55 task or some other task that is more than about 30-dimensional.
- Do the authors agree that the method could be described as a "twisted" proposal for SMC? And further, even the initial choice of using the modified diffusion reverse process would be also simply another choice of a twisted proposal. My understanding is that this is a standard idea in the Sequential Monte Carlo literature, although the basis function scheme and the variance-minimization is novel at least in the diffusion context to my knowledge. The problem is that this seems to set the first contribution of 'identifying a fundamental degree of freedom in the Feynman-Kac type FP equation' to be less novel than it sounds, but I am open to being corrected on this. In any case, I think that positioning the method in its proper wider SMC context would be helpful for correctly contextualizing the paper.
- In the protein-ligand, task, would another simple baseline be to take the base model samples, and optimize them directly using the gradient of the energy function? Does the method outperform this?
- The paper [1] seems to be a relevant prior work on annealed distributions with diffusion models with SMC (and is the first paper to do this?). It may be that the proposed method is better than this, but I think it should be compared to at least in some context to clearly show this.
- Further, the paper [2] does reward-tilted sampling with SMC by using a different twisted proposal by using the $p(y|x_0(x_t))$ gradient. It would be good to compare to that as well in the reward-guided experiments.

References

[1] Thornton, James, et al. "Composition and Control with Distilled Energy Diffusion Models and Sequential Monte Carlo." The 28th International Conference on Artificial Intelligence and Statistics (2025).

[2] Wu, Luhuan, et al., "Practical and Asymptotically Exact Conditional Sampling in Diffusion Models", 37th Conference on Neural Information Processing Systems (NeurIPS 2023).

**Questions:**

- Although not considered in previous literature on SMC, it seems that another simple baseline would be to not use SMC and instead interleave Langevin dynamics steps after each generative step, similarly to [1], such that the wall-clock time is matched with the proposed method. I would be curious to see how well this could work on some task, although this is not a priority.

Overall, I will start out with a marginal reject due to the concerns raised in the weaknesses, but am open to changing the score with the rebuttal. I think the core idea in the paper is interesting and mostly seems to work well and potentially could be extended further. As such I am not initially strongly against accepting the paper with the assumption that the weaknesses are cleared out or I am incorrect about them.

References

[1] Du, Yilun, et al. "Reduce, reuse, recycle: Compositional generation with energy-based diffusion models and mcmc."

---

> ### Author Response · Authors · 2025-11-23
> **Authors' Reply to Reviewer cAK7 (1/4)**
>
> Thank you for your careful and detailed review. Your comments on computational cost, scaling, and connections to SMC have been extremely helpful. We address your points in turn and summarize the resulting changes.
>
> ---
>
> ### Regarding Wall-Clock Time Comparsion
>
> > The method incurs significant cost on the wall-clock time compared to the baselines, 2.3x-6x in the experiments for which the wall-clock time was reported. This is not mentioned in the limitations or in the main paper, however, and the conclusion instead states that the method causes minimal computational overhead, potentially misleading the reader or a reviewer who does not have the time to read through the Appendix. Backpropagation through the denoiser at each step requires much more memory than the forward pass and, and my assumption is that it may limit the batch size in practice.
>
> We acknowledge that our initial characterization of the overhead as "minimal" was imprecise regarding wall-clock time, and we have revised this to "modest and scalable overhead." By “lightweight” or “minimal” we intended to emphasize: (1) the training‑free nature of the method (no extra control network to train), and (2) the simplicity of the core control step (a tiny linear system per time step), rather than suggesting that it is as cheap as pure guidance. We have made this distinction explicit to avoid misleading.
>
> Backpropagation through the denoiser for Laplacians certainly has both compute and memory cost. However, in many relevant applications, the reward functions are explicit physical potentials with analytic and easy-to-compute gradients. Furthermore, the cost of Hutchinson's estimator can be explicitly controlled by balancing accuracy and speed. Crucially, even if the control drift is approximate due to a limited budget, the underlying SMC reweighting ensures the method remains asymptotically exact.
>
> > As such, it might be more fair to compare to the baselines by adjusting the inference hyperparameters such that the wall-clock time is equal. E.g., I would expect that the G-SMC method improves with more particles, and potentially with more diffusion steps.
>
> To address your suggestion for a fairer comparison, we conducted a new set of experiments on the GMM tasks where we aligned the wall-clock time across all methods, with:
> - **Baselines (PG, G-SMC)**: 8192 particles and 2000 diffusion steps.
> - **DriftLite (VCG/ECG)**: 1024 particles and 1000 diffusion steps.
>
> The results are as follows:
> - **Annealing ($\gamma = 2.5$)**
> | Metric           | PG               | G-SMC            | VCG             | VCG-SMC         | ECG              | ECG-SMC          |
> |------------------|------------------|------------------|------------------|------------------|-------------------|-------------------|
> | MMD          | 0.73 ± 0.22      | 0.46 ± 0.07      | 0.06 ± 0.00      | 0.05 ± 0.00      | 0.40 ± 0.08       | 0.07 ± 0.01       |
> | SWD          | 18.52 ± 5.00     | 18.87 ± 3.69     | 1.85 ± 0.36      | 1.75 ± 0.41      | 10.22 ± 2.29      | 2.69 ± 0.61       |
> | Mean L2      | 90.58 ± 29.89    | 102.17 ± 20.40   | 7.76 ± 1.55      | 7.33 ± 1.61      | 50.45 ± 9.10      | 14.29 ± 4.05      |
> | $\Delta$NLL    | -5.04 ± 1.31     | -0.56 ± 0.85     | 0.11 ± 0.20      | 0.04 ± 0.23      | -0.75 ± 1.27      | 0.08 ± 0.25       |
> | Wall Clock (s) | 17.28 ± 0.49   | 18.45 ± 0.84     | 15.12 ± 0.14     | 15.77 ± 0.13     | 13.92 ± 0.18      | 14.83 ± 0.19      |
>
> - **Quadratic ($\sigma = 100$)**
> | Metric             | PG                | G-SMC             | VCG               | VCG-SMC           | ECG               | ECG-SMC           |
> |--------------------|-------------------|-------------------|-------------------|-------------------|-------------------|-------------------|
> | MMD            | 0.96 ± 0.06       | 0.25 ± 0.08       | 0.09 ± 0.01       | 0.05 ± 0.01       | 0.23 ± 0.10       | 0.06 ± 0.01       |
> | SWD            | 19.20 ± 7.74      | 2.81 ± 1.40       | 0.83 ± 0.12       | 0.59 ± 0.17       | 1.80 ± 0.64       | 1.03 ± 0.58       |
> | Mean L2        | 112.27 ± 26.70    | 17.13 ± 9.97      | 3.19 ± 0.73       | 2.57 ± 0.90       | 8.49 ± 3.84       | 5.93 ± 3.52       |
> |  $\Delta$NLL       | 37.92 ± 6.06      | 1.23 ± 1.39       | 0.36 ± 0.21       | 0.25 ± 0.19       | 0.39 ± 0.76       | -0.01 ± 0.12      |
> | Wall Clock (s) | 15.98 ± 0.51      | 16.74 ± 0.31      | 14.96 ± 0.01      | 15.55 ± 0.10      | 14.73 ± 0.09      | 15.30 ± 0.05      |
>
> Even under this equal‑time budget, the results show that DriftLite still significantly outperforms baselines. Increasing the budget does improve G‑SMC, but the performance gap persists, indicating that the issue is fundamentally variance degeneracy rather than inadequate compute. We have included these new results and corresponding discussion in Appendix C.1 and Table 6 in the revised manuscript.

---

> ### Author Response · Authors · 2025-11-23
> **Authors' Reply to Reviewer cAK7 (2/4)**
>
> ### Regarding Additional Results for Protein-Ligand Co-Folding Task
>
> We thank the reviewer for this suggestion. In the revised manuscript, we reported both ESS trajectories and wall-clock runtimes for the protein-ligand co-folding task.
>
> - **ESS comparison**: In Appendix C.3, Figure 16, we plot the normalized ESS over inference steps for three representative protein–ligand complexes (PDB IDs 7n7h, 7zu2, and 8d39). For all three systems, VCG-SMC maintains substantially higher ESS than the G-SMC baseline, particularly in the middle and late stages of the trajectory, where path degeneracy is most severe. This indicates that the DriftLite control keeps a much larger fraction of particles effective, even with a relatively small particle budget.
>
> - **Wall-clock runtime**: We further measured wall-clock time per generated complex on a random subset of 20 protein-ligand structures and report the results in Table 13, as also reported below:
>
>     | Method              | Base  | G-SMC | VCG-SMC |
>     |---------------------|-------|-------|---------|
>     | Wall-Clock Time (s) | 10.05 | 31.67 | 31.78  |
>
>     VCG-SMC incurs essentially no additional wall-clock overhead relative to G-SMC, while achieving consistently higher ESS and improved physical validity on PoseBusters. Thus, on this higher-dimensional protein–ligand co-folding task, DriftLite delivers significant stability and quality gains at negligible extra cost beyond a standard SMC correction.
>
> ---
>
> ### Regarding Higher-Dimensional Examples
>
> To directly probe high-dimensional behavior without confounding base-model quality issues, we ran 100‑dimensional GMM experiments for both annealing and quadratic tilting.
> - **Annealing ($\gamma = 2.5$)**
> | Metric       | PG               | G-SMC            | VCG              | VCG-SMC          | ECG               | ECG-SMC           |
> |-------------|------------------|------------------|------------------|------------------|-------------------|-------------------|
> | MMD     | 0.93 ± 0.10      | 0.15 ± 0.02      | 0.03 ± 0.00      | 0.02 ± 0.00      | 0.45 ± 0.13       | 0.04 ± 0.00       |
> | SWD     | 22.87 ± 2.29     | 22.08 ± 2.23     | 1.07 ± 0.19      | 1.00 ± 0.15      | 12.11 ± 4.16      | 6.74 ± 0.97       |
> | Mean L2 | 211.94 ± 23.38   | 213.48 ± 27.56   | 9.16 ± 1.47      | 8.73 ± 1.30      | 101.97 ± 24.57    | 63.92 ± 10.17     |
> | $\Delta$NLL | -13.66 ± 1.81    | -6.35 ± 1.03     | 0.52 ± 0.10      | 0.54 ± 0.13      | -7.26 ± 1.53      | 0.56 ± 0.16       |
>
> - **Quadratic ($\sigma = 100$)**
> | Metric       | PG               | G-SMC            | VCG             | VCG-SMC         | ECG              | ECG-SMC          |
> |--------------|------------------|------------------|------------------|------------------|-------------------|-------------------|
> | MMD      | 1.00 ± 0.01      | 0.39 ± 0.08      | 0.06 ± 0.01      | 0.03 ± 0.00      | 0.29 ± 0.21       | 0.03 ± 0.00       |
> | SWD      | 21.19 ± 4.60     | 9.54 ± 6.08      | 0.57 ± 0.13      | 0.22 ± 0.05      | 2.43 ± 1.51       | 0.61 ± 0.74       |
> | Mean L2  | 249.73 ± 9.07    | 111.96 ± 74.42   | 3.91 ± 0.73      | 1.80 ± 0.29      | 19.92 ± 13.20     | 6.03 ± 7.74       |
> | $\Delta$NLL | 110.63 ± 19.68   | 19.23 ± 6.86     | 0.94 ± 0.11      | 0.96 ± 0.18      | 4.50 ± 3.82       | 0.91 ± 0.25       |
>
> Key observations include (1) DriftLite (VCG‑SMC) achieves near-zero MMD and significantly lower SWD and ΔNLL than PG and G‑SMC, and (2) The gap between DriftLite and baselines widens as dimension increases: weight degeneracy in G‑SMC becomes more severe, while our variance control remains stable. This suggests that DriftLite scales well to higher dimensions and the target remains a structured perturbation of the base.
>
> As discussed with Reviewer ncQu, applying DriftLite to LJ‑55 is highly interesting but would require a base model that already handles the extreme multimodality. As noted in recent literature [6,7], generating high-quality samples for such complex systems remains an open research problem and ESS typically falls below 10% with an order of magnitude lower than that of LJ-13. We are actively working on scaling our method to larger physical systems as a priority for future research.

---

> ### Author Response · Authors · 2025-11-23
> **Authors' Reply to Reviewer cAK7 (3/4)**
>
> ### Regarding Additional Baselines
> > In the protein-ligand, task, would another simple baseline be to take the base model samples, and optimize them directly using the gradient of the energy function? Does the method outperform this?
>
> We thank the reviewer for this suggestion. In our protein–ligand co-folding experiments we closely follow the open-source implementation of the Boltz-2 model [8]. In particular, a key component of G-SMC is the local refinement step that, at each inference iteration, performs gradient descent on the energy function, as in the original Boltz-2 implementation. Thus, the simple “optimize base samples with the energy gradient” baseline is already incorporated into our experimental setup.
>
> > Although not considered in previous literature on SMC, it seems that another simple baseline would be to not use SMC and instead interleave Langevin dynamics steps after each generative step, similarly to [1], such that the wall-clock time is matched with the proposed method. I would be curious to see how well this could work on some task, although this is not a priority.
>
> Thanks for the suggestion. We implemented the suggested baseline, denoted as G-LD (Guidance with Langevin Dynamics), on the GMM tasks. To match the wall-clock time of our proposed method, we interleaved 10 steps of Langevin dynamics after each generation step. We applied this correction only during the final 20% of the trajectory to preserve the global structure formed in the early stages, a hyperparameter choice that yielded the best results among several configurations we tested.
>
> The results are summarized in the tables below:
> - **Annealing ($\gamma = 2.5$)**
> | Metric       | PG             | G-SMC          | VCG            | VCG-SMC        | ECG            | ECG-SMC        | G-LD           |
> |--------------|----------------|----------------|----------------|----------------|----------------|----------------|----------------|
> | MMD      | 0.75 ± 0.08    | 0.20 ± 0.02    | 0.04 ± 0.00    | 0.03 ± 0.00    | 0.50 ± 0.19    | 0.03 ± 0.00    | 0.75 ± 0.09    |
> | SWD      | 20.60 ± 4.37   | 23.80 ± 2.66   | 1.11 ± 0.08    | 1.13 ± 0.24    | 14.53 ± 7.89   | 6.60 ± 2.04    | 20.60 ± 4.38   |
> | Mean L2  | 166.03 ± 24.85 | 229.19 ± 10.04 | 9.04 ± 0.85    | 9.73 ± 1.57    | 120.77 ± 46.23 | 60.08 ± 18.07  | 166.89 ± 25.00 |
> | $\Delta$NLL| -16.37 ± 2.62  | -5.06 ± 1.33   | 0.20 ± 0.13    | 0.24 ± 0.17    | -6.11 ± 2.78   | 0.21 ± 0.21    | -14.94 ± 2.62  |
>
> - **Quadratic ($\sigma = 100$)**
> | Metric       | PG             | G-SMC          | VCG            | VCG-SMC        | ECG            | ECG-SMC        | G-LD           |
> |--------------|----------------|----------------|----------------|----------------|----------------|----------------|----------------|
> | MMD      | 0.95 ± 0.09    | 0.39 ± 0.10    | 0.07 ± 0.01    | 0.03 ± 0.00    | 0.10 ± 0.04    | 0.03 ± 0.00    | 0.94 ± 0.11    |
> | SWD      | 20.61 ± 7.21   | 13.73 ± 7.04   | 0.63 ± 0.13    | 0.28 ± 0.11    | 1.04 ± 0.58    | 0.30 ± 0.16    | 20.18 ± 7.15   |
> | Mean L2  | 216.92 ± 69.48 | 172.02 ± 68.34 | 4.59 ± 1.33    | 2.62 ± 1.25    | 9.12 ± 6.55    | 3.02 ± 2.35    | 210.44 ± 68.72 |
> | $\Delta$NLL  | 114.98 ± 31.50 | 21.75 ± 2.92   | 0.58 ± 0.27    | 0.39 ± 0.19    | 0.25 ± 1.10    | 0.41 ± 0.17    | 101.42 ± 27.63 |
>
> As shown, while G-LD shows improvements in some metrics (e.g., $\Delta$NLL) to pure guidance, it fails to correct the fundamental bias of guidance. Intuitively, this is expected: since the guidance drift itself ($\nabla \log p_t + \nabla r_t$) is a heuristic approximation that ignores the time-dependent normalization constant, the resulting distribution is biased. G-LD corrects this to some extent, whereas DriftLite corrects the sampling path entirely to target the true posterior.

---

> ### Author Response · Authors · 2025-11-23
> **Authors' Reply to Reviewer cAK7 (4/4)**
>
> ### Regarding Comparison with Twisted Proposal for SMC
>
> This is an insightful connection and we agree. Our method can indeed be framed as a principled way to construct a "twisted proposal" for SMC. We do not claim that the existence of this degree of freedom (Prop. 3.1) is itself new.
>
> Our novelty, as we will clarify in the text, is threefold as follows:
> - **Goal**:  We explicitly target variance minimization of the reweighting potential as the design principle for the twist in the diffusion setting, rather than simply improving heuristics or approximating conditionals.
> - **Theory**: We identify the curl-free optimal control as the solution of a Poisson PDE (Prop. 3.2), which is not spelled out in prior diffusion‑SMC methods.
> - **Method**: We propose a training-free, low‑rank basis projection that reduces this PDE to a tiny linear system at each step, usable directly with pre-trained score models.
>
> Furthermore, we point out that in the SMC literature, the twisted proposals are often considered in discrete-time settings, while we directly work with Fokker-Planck equations and give continuous-time formulations. We have revised our "first contribution" claim to be more precise and better position our work in the wider SMC context.
>
> ---
>
> ### Regarding comparisons with literature [4,5]
>
> Thank you for pointing out the closely related works [4,5]. We briefly clarify their methodology and how they relate to our setting.
> - **Wu et al. [5]**: We already cite this paper as one of the key works (along with [1]) that establishes the G-SMC baseline. Their method keeps the standard reverse diffusion dynamics as the proposal and reduces weight variance by introducing twisting functions in the Feynman–Kac weights, built from approximate conditional scores and guidance terms. In other words, TDS lies in the same SMC family as our G-SMC baseline: it reuses the unmodified reverse diffusion drift and only adjusts the weights via twisting, without actively designing a new drift. Our work is motivated precisely by the “weight degeneracy” bottleneck of such passive reweighting. DriftLite instead explicitly modifies the drift field itself to minimize the variance of the reweighting potential. Conceptually, our empirical comparison against G-SMC is therefore already a comparison to this “twisted-weights, fixed-drift” paradigm.
> - **Thornton et al. [4]**: This paper also uses SMC for control, but first runs a dedicated training procedure to distill a pre-trained diffusion model into an energy-based model, and then uses these learned energies in an SMC scheme for composition and low-temperature sampling. This is fundamentally a training-based approach: it requires additional distillation to obtain an EBM before SMC. Our work, in contrast, is strictly training-free: we keep the pre-trained score model fixed and compute a low-rank control drift on-the-fly at inference time by solving tiny linear systems.
>
> We would also like to point out that DriftLite is not mutually exclusive to these approaches. DriftLite can be applied synergistically on top of their dynamics to correct for any residual variance or approximation error in the learned control, providing a "best of both worlds" solution.
>
> ---
>
> We appreciate your insistence on fair compute comparisons and proper SMC context. It has led us to run new experiments and refine our contribution claims. Given the clarified positioning, additional equal-time and high-dimensional results, and the new Langevin baseline, we hope you might reconsider the evaluation of our paper.
>
> ---
>
> ### References
>
> [1] Skreta, Marta, et al. "Feynman-kac correctors in diffusion: Annealing, guidance, and product of experts." arXiv preprint arXiv:2503.02819 (2025).\
> [2] Chen, Haoxuan, et al. "Solving inverse problems via diffusion-based priors: An approximation-free ensemble sampling approach." arXiv preprint arXiv:2506.03979 (2025).\
> [3] Du, Yilun, et al. "Reduce, reuse, recycle: Compositional generation with energy-based diffusion models and mcmc."\
> [4] Thornton, James, et al. "Composition and Control with Distilled Energy Diffusion Models and Sequential Monte Carlo." The 28th International Conference on Artificial Intelligence and Statistics (2025).\
> [5] Wu, Luhuan, et al., "Practical and Asymptotically Exact Conditional Sampling in Diffusion Models", 37th Conference on Neural Information Processing Systems (NeurIPS 2023).\
> [6] Klein, Leon, Andreas Krämer, and Frank Noé. "Equivariant flow matching." Advances in Neural Information Processing Systems 36 (2023): 59886-59910.\
> [7] Akhound-Sadegh, Tara, et al. "Iterated denoising energy matching for sampling from boltzmann densities." Proceedings of the 41st International Conference on Machine Learning. 2024.\
> [8] Passaro, Saro, et al. "Boltz-2: Towards accurate and efficient binding affinity prediction." BioRxiv (2025).

---

> > ### Comment · Reviewer_cAK7 · 2025-11-28
> >
> > I thank the authors for the thorough rebuttal and the effort on improving the paper. Many of my concerns have been answered, and I will reconsider the score accordingly.
> >
> > The only open questions from my earlier points are regarding the additional baselines:
> > - Protein-ligand: I understand that G-SMC does also use the energy function in the inference steps, but I was thinking of the baseline where we do multiple steps of gradient descent at the end of generation until convergence. I am not very familiar with the task, but is my understanding correct that this would improve all the measured scores in Table 2? If this is the case, then perhaps some other evaluation metric would be useful?
> > -  Wu et al. [1] : On the point "Their method keeps the standard reverse diffusion dynamics as the proposal", looking at their Algorithm 1, seems that they do change the proposal using the conditional score approximation, and as such this statement is not correct? Also, if we were to combine this with Driftlite, would it end up in double backpropagation due to need for backpropagation in both methods?
> > - On Thornton et al. [2], I understand that the method requires a different type of training, but seems that the end result is still similarly adaptable to different types of tasks? As such, it seems to me that the key difference is that DriftLite is applicable to standard diffusion models, whereas [2] is not. But this distinction seems smaller than the "training-free/training-based" distinction, as [2] also does not require training after the initial pre-training of the model (including the energy function). If we train a particular model with the knowledge that we will use it for different inference-time modifications, the distinction also matters much less.
> >
> > References:
> >
> > [1] Wu, Luhuan, et al., "Practical and Asymptotically Exact Conditional Sampling in Diffusion Models", 37th Conference on Neural Information Processing Systems (NeurIPS 2023).
> >
> > [2] Thornton, James, et al. "Composition and Control with Distilled Energy Diffusion Models and Sequential Monte Carlo." The 28th International Conference on Artificial Intelligence and Statistics (2025).

---

> > > ### Author Response · Authors · 2025-12-01
> > > **Authors' Reply to Reviewer cAK7 (1/2)**
> > >
> > > We sincerely thank the reviewer for the thoughtful follow-up and for reconsidering the score. We appreciate the opportunity to clarify the nuances regarding baselines and related work. We address your three remaining questions below.
> > >
> > > ---
> > >
> > > ### Regarding the Gradient Descent Baseline
> > > You are right that, in principle, one could consider a baseline where, after running the base diffusion model, we perform many additional gradient descent (or Langevin) steps on the physics energy until convergence. However, in the large protein–ligand systems we consider, this is extremely challenging in practice. The underlying energy landscape is highly rugged and high-dimensional, so overdamped Langevin starting from a single generated structure typically requires a very long trajectory to mix or escape local minima. Moreover, the learned score $\nabla \log p_0$​ of a diffusion model is only an approximation to the true equilibrium gradient; using it as the drift in a long-run Langevin chain can accumulate bias and does not guarantee convergence to the intended target distribution in a realistic compute budget.
> > >
> > > For these reasons, we view gradient-based corrections as a *local refinement mechanism* rather than a general solution: a small number of correction steps can improve physical validity around the samples produced by the generative model, but pushing this to “optimize-until-convergence” would be both computationally prohibitive and conceptually closer to energy minimization than to generative sampling. This is also consistent with our additional experiments, where increasing the number of refinement steps improves PoseBusters metrics to some extent but does not close the gap to our SMC-based control. Our approach instead applies penalty potentials throughout the generative trajectory, which allows us to steer the sampling process itself rather than relying on very long, post-hoc optimization to repair samples.
> > >
> > > ---
> > >
> > > ### Regarding Comparison with Wu et al. [1]
> > > You are correct that our previous wording was too loose. Wu et al. (TDS) do modify the proposal via conditional score approximations and a more accurate description is that they retain the standard reverse diffusion family but "twist" the proposals with conditional score approximations, modifying the SMC weights accordingly. This resembles our G-SMC baseline which instead instantiated in a continuous-time setting, where the conditional score approximation applies to the control drift analogously. We will update the text to reflect this and acknowledge Wu et al. [1] as a pioneering work in inference-time scaling of diffusion models using SMC.
> > >
> > > Regarding the combination of TDS and DriftLite: we agree that a naïve combination would require gradients for both the conditional score (TDS) and the control statistics (DriftLite) at each step. However, these operations are conceptually compatible. TDS requires the gradient of a likelihood/classifier, while DriftLite requires gradients of a reward/energy. Modern autograd frameworks handle this seamlessly within a single pipeline without "double backpropagation" in the sense of differentiating through a gradient (Hessian computation). While combining the methods incurs an additive cost in derivative evaluations, there is no structural obstruction. We view the combination of twisted proposals with DriftLite control as a promising direction for future work.
> > >
> > > ---
> > >
> > > ### Regarding Comparison with Thornton et al. [2]
> > >
> > > We agree that the primary distinction is DriftLite’s immediate applicability to standard diffusion models without extra training, whereas Thornton et al. [2] requires a specific "distilled energy" setup. While pre-training an energy function does offer more algorithmic freedom, we believe the "training-free" distinction is substantial. DriftLite is designed to be general, lightweight, and directly deployable on any pre-trained diffusion backbone. This generality is made possible by our mathematical derivations (specifically Propositions 2.1 and 3.1), which eliminate the need to use the energy function itself in the control drifts. This allows us to achieve significant performance improvements purely through inference-time compute, a contribution we view as distinct from the distilled approach of Thornton et al.

---

> ### Author Response · Authors · 2025-12-01
> **Authors' Reply to Reviewer cAK7 (2/2)**
>
> ### Final Remarks
>
> Across your original review and the ensuing discussion, your comments have substantially improved both the empirical evaluation and the positioning of the paper. Concretely, in response to your suggestions we:
> - **Clarified Runtime Comparisons**: We added equal wall-clock experiments showing DriftLite outperforms G-SMC in MMD, SWD, and NLL even when G-SMC is allocated comparable time budgets. We also conducted a wall-clock time comparison for the protein-ligand co-folding experiment, as well as exemplary ESS evolutions to further illustrate the performance improvement of our DriftLite method on this complex problem.
> - **Probed High-Dimensionality**: We included 100-dimensional GMM experiments demonstrating that DriftLite maintains sample quality while G-SMC suffers from severe weight degeneracy as dimensions increase.
> - **Analyzed Additional Baselines**: We implemented a G-LD baseline (interleaved Langevin) and clarified why long-run "optimization-until-convergence" is impractical for rugged protein landscapes compared to our pathwise control.
> - **Sharpened Context**: We refined our discussion to explicitly position DriftLite against "twisted proposal" methods (like TDS) and energy-distilled approaches (like Thornton et al.), clarifying that our method is uniquely training-free and energy-agnostic.
>
> Taken together, these additions and clarifications were directly motivated by your feedback and, we believe, result in a much clearer and more robust paper. We are very grateful for your careful engagement and hope that the revised manuscript now addresses your concerns.
>
> ---
> ### References
> [1] Wu, Luhuan, et al., "Practical and Asymptotically Exact Conditional Sampling in Diffusion Models", 37th Conference on Neural Information Processing Systems (NeurIPS 2023).\
> [2] Thornton, James, et al. "Composition and Control with Distilled Energy Diffusion Models and Sequential Monte Carlo." The 28th International Conference on Artificial Intelligence and Statistics (2025).

---

### Official Review · Reviewer_hiDh · 2025-11-01

**Soundness:** 4
**Presentation:** 4
**Contribution:** 4
**Rating:** 8
**Confidence:** 2

**Summary:**

The paper proposes DriftLite, a lightweight alternative to reward-driven particle dynamics derived from the Feynman–Kac formulation of the Fokker–Planck equation.
While the principled dynamics (Prop. 2.1) yield unbiased sampling by reweighting trajectories according to a reward function, they suffer from severe weight degeneracy and are impractical for high-dimensional problems.
DriftLite replaces explicit weighting with a Variational Coefficient Generator (VCG) that learns a low-rank, three-basis decomposition of the drift field, achieving a balance between theoretical faithfulness and numerical stability.
The method demonstrates stable performance on challenging high-dimensional tasks, including protein–ligand systems.

**Strengths:**

## Strong theoretical grounding.
Proposition 2.1 provides a clean derivation from the Feynman–Kac representation, clarifying why naive guidance or reward-based drift correction leads to unnormalized density propagation. In addition, Proposition 3.1 theoretically provides design space within the Fokker-Plank equation.

## Elegant practical solution.
The transition from weighted to unweighted dynamics via the VCG formulation is conceptually neat and computationally efficient.
Representing drift using three physically motivated basis components—potential, diffusive, and reward—yields a low-rank parameterization.
Directly minimizing the variance of the potential term under a simple parameterization reduces the problem to a linear system, making the overall procedure remarkably simple.

## Empirical validation on high-D systems.
Although the approach projects the control drift $b$ onto a three-dimensional subspace, it scales to complex molecular and protein–ligand environments without losing stability, indicating that this design effectively captures the dominant drift modes.

**Weaknesses:**

## Partial theoretical exposition.
While Prop. 2.1 and 3.1 are elegant, the paper omits intermediate derivations linking the reward term to the weighted dynamics; some readers may struggle to follow the jump from theory to implementation.

**Questions:**

Could the authors clarify whether the choice of three basis functions in the VCG has a theoretical grounding?
In particular, does this low-rank representation guarantee that the dominant drift modes of the reward-driven dynamics are captured, or is it mainly an empirical observation?

---

> ### Author Response · Authors · 2025-11-23
> **Authors' Reply to Reviewer hiDh**
>
> Thank you for your very positive and accurate summary of our work, and for calling the theoretical foundation “strong” and the solution “elegant.” We address your main suggestions below.
>
> ---
> ### Regarding Partial Theoretical Exposition
> > While Prop. 2.1 and 3.1 are elegant, the paper omits intermediate derivations linking the reward term to the weighted dynamics; some readers may struggle to follow the jump from theory to implementation.
>
> Thank you for this feedback. We provide the full, rigorous, step-by-step derivation for Proposition 2.1 linking the target density to the reweighting potential $g_t$ in Appendix A.1. We will add a more prominent pointer to this appendix in the main text so readers can easily find the full details. This should make the bridge from theory to implementation much clearer for readers who are not deeply familiar with the Feynman–Kac machinery.
>
> ---
>
> ### Regarding Choice of Bases
> > Could the authors clarify whether the choice of three basis functions in the VCG has a theoretical grounding? In particular, does this low-rank representation guarantee that the dominant drift modes of the reward-driven dynamics are captured, or is it mainly an empirical observation?
>
> This is an excellent question. The choice is theoretically motivated rather than just an empirical observation.
> In Appendix A.4 (p. 20), we derive the formal solution for the optimal control $b_t^*$. This formal solution is complex, but its local structure is determined by the quantities $\nabla \log q_t$ and the potential $g_t$.If we look at the definitions of $\nabla \log q_t$ and $g_t$, they are themselves composed of three core, locally-available vector fields: the score ($\nabla \log p_t$), the reward gradient ($\nabla r_t$), and the forward drift ($u_t$). Therefore, our VCG basis is a principled, low-order approximation of the optimal control, formed by projecting the intractable optimal solution onto the subspace spanned by its most critical and locally available components.
>
> The strong empirical results, especially in the challenging LJ-13 system (Fig. 5), confirm that this low-rank approximation is highly effective at capturing the dominant drift modes. We made this connection to the formal solution in App. A.4 more explicit in the main text.
>
> ---
>
> Thank you again for your supportive review and constructive suggestions. We believe the added derivation pointers and basis justification will further improve the paper’s readability.

---

### Official Review · Reviewer_ncQu · 2025-11-02

**Soundness:** 3
**Presentation:** 3
**Contribution:** 2
**Rating:** 4
**Confidence:** 3

**Summary:**

The authors propose a method for learning to sample from a combination of tempered and/or tilted distribution, where the exponent of a reward function describes the tilting factor. Importantly, they note that the Feynman Kac PDE can be manipulated by adding an additional drift term which is compensated in the re-weighting factor, reducing the problem of sampling from the new target density to finding an optimal drift. It is an important direction of research since re-weighting typically relies on self-normalized importance sampling which can lead to importance weights blowing up. Thus, finding a good additional drift function can reduce this problem by essentially reducing the variance of the importance weights. The authors further demonstrate the improved performance of their framework, which relies on efficiently finding the optimal control through solving of a linear system of equations.

**Strengths:**

- The authors provide a rigorous and correct assessment of the setting, where existing methods rely on self-normalized importance sampling which suffer from the problem of large variance of importance weights and the sensitivity to the number of particles used for integration.
- The experiments indicate that the proposed method leads to improved performance across a wide array of settings, substantiating the generalizability of the proposed method.

**Weaknesses:**

- A lot of the theory proposed in this work is actually already well known and not new, and the authors should have introduced it as such. In particular, Prop 3.1 is the basis of [1-3] which highlights that the Feynman Kac PDE can be manipulated with additional drifts as long as they are similarly compensated for in re-weighting.
- Is there a reason why the authors did not consider the more difficult setting of LJ-55 to evaluate their framework?
- I strongly urge the authors to provide comparisons to [2-3] which follows the same idea, learning a neural network model as an additional drift to reduce the variance of importance weights. A clear comparative analysis between modeling the drift as a linear coefficient of basis functions or a neural network would make this manuscript a lot better.
- In continuation of the previous point, it is further important to highlight the training and inference costs of the proposed method against [2,3]. While I agree that the proposed method only requires solving a system of linear equations, it does require this at all time-points every time during inference. On the other hand, learning a time-conditioned neural network requires an upfront cost of training but then at inference only requires an additional forward pass at each step. Is solving this system of linear equations at every step cheaper than a forward pass at each step? Such an analysis has been missing from this work.

[1] Skreta, Marta, et al. "Feynman-kac correctors in diffusion: Annealing, guidance, and product of experts." arXiv preprint arXiv:2503.02819 (2025).

[2] Albergo, Michael S., and Eric Vanden-Eijnden. "Nets: A non-equilibrium transport sampler." arXiv preprint arXiv:2410.02711 (2024).

[3] Vargas, Francisco, et al. "Transport meets variational inference: Controlled monte carlo diffusions." arXiv preprint arXiv:2307.01050 (2023).

**Questions:**

- How do the authors evaluate their method on LJ-13 and the other benchmarks? How are the ground-truth samples obtained to compute metrics such as MMD?
- It would be good to provide some of the standard metrics like 2-Wasserstein distance.
- What is the reward-tilting used in LJ-13, and what is the motivation behind the specific choice of tilting applied?
- Why do the authors not compare against reward-based diffusion finetuning tasks and methods, e.g. [4-6]?

[4] Venkatraman, Siddarth, et al. "Amortizing intractable inference in diffusion models for vision, language, and control." Advances in neural information processing systems 37 (2024): 76080-76114.

[5] Domingo-Enrich, Carles, et al. "Adjoint matching: Fine-tuning flow and diffusion generative models with memoryless stochastic optimal control." arXiv preprint arXiv:2409.08861 (2024).

[6] Fan, Ying, et al. "Dpok: Reinforcement learning for fine-tuning text-to-image diffusion models." Advances in Neural Information Processing Systems 36 (2023): 79858-79885.

---

> ### Author Response · Authors · 2025-11-23
> **Authors' Reply to Reviewer ncQu (1/4)**
>
> Thank you for your detailed review. We appreciate the opportunity to clarify the novelty, positioning, and empirical evaluation of DriftLite. We respond to your main points below.
>
> ---
> ### Regarding Relations to Literature [1-6]
>
> We thank the reviewer for raising several questions regarding the comparison of our work with the existing literature, including conceptual [1-3], and empirical [4-6] comparisons.
>
> ### *Clarification on Theoretical Contribution*
>
> We agree that the existence of the "degree of freedom" (Prop. 3.1) is a known concept in optimal control and SMC literature, especially recent diffusion-based works [1,2,3]. We have cited all these works in our manuscript and attributed Prop. 3.1 to [1,2].
>
> Our contribution is not the discovery of this degree of freedom, but in how we instantiate and use it for inference-time scaling of diffusion models:
> - **Goal**: We are, to our knowledge, the first to explicitly formulate the control drift as a tool to actively minimize the variance of the SMC reweighting potential in the context of diffusion-based inference-time scaling, rather than passively accept the variance induced by a chosen drift.
> - **Theory**: We show that the optimal variance-eliminating control $\boldsymbol b_t^*$ is the unique curl‑free solution of a high-dimensional Poisson equation (Prop. 3.2), thereby providing a concrete PDE characterization of the best possible drift along this degree of freedom.
> - **Method**: We propose DriftLite, a training-free approximation that projects this intractable solution onto a small, physically motivated basis. This turns the PDE into a simple 3x3 linear system at each step and can be applied on-the-fly to any pre-trained diffusion model and reward.
>
> We have revised the paper to more explicitly acknowledge this, including softening the first contribution bullet to reflect this more precise scope (“identify” to “formalize and exploit”)
>
> ### *Comparison with Training-Based Methods [2,3]*
>
> We agree it is very important to compare with training-based methods [2,3] and made it more explicit in the main text. The methods in [2, 3] are **training-based**, whereas our method is **training-free**. This is a fundamental difference in scope and application.
>
> Here we give a brief summary of conceptual differences between our work and related works [1-3].
> - **Skreta, et al. [1]**: This paper (which we refer to as G-SMC) is our primary baseline. It identifies the correct reweighting term $g_t$ but passively reweights particles, leading to the high variance and weight degeneracy that our work aims to solve. Our method actively controls the drift to minimize this variance.
> - **Albergo & Vanden-Eijnden [2]**: This is a training-based method. It proposes learning a neural network $b_t(x)$ via an objective derived from Jarzynski’s equality before inference. Our method computes the control drift $b_t(x)$ on-the-fly during inference by solving a small, computationally simple linear system.
> - **Vargas, et al. [3]**: This is also a training-based method. It, similar to [2], learns a control drift by optimizing a stochastic control objective via training a neural network before inference. Our method bypasses this entirely, making it an inference-time solution.
>
> We have expanded discussion in our methodology part, highlighting this “training-based control vs training-free control” dichotomy and making clear that our focus is the latter.

---

> ### Author Response · Authors · 2025-11-23
> **Authors' Reply to Reviewer ncQu (2/4)**
>
> ### *Cost Comparison with Training-Based Methods [2,3]*
>
> We thank the reviewer for this insightful question on the training and inference cost comparison with training-based methods, especially [2,3]. It allows us to clarify the precise computational trade-offs between our training-free approach and training-based alternatives like [2,3].
>
> Solving our 3x3 linear system is orders of magnitude cheaper than a neural network forward pass. The primary overhead of our method is not solving the system, but building it at each step, which requires evaluating the basis functions and their inner products involving backward passes for Laplacians. This makes our per-step inference cost more but still modestly comparable to that of naive guidance methods. While not negligible, this cost can be tuned by allowing trade-off between accuracy and speed. As for training-based methods, if the control network is of comparable size to the base for expressivity, the inference cost is roughly doubled, similar to if not more than the overhead of our methods.
>
> However, we believe this comparison of inference cost alone would overlook the distinct use cases of the two paradigms: training-based [2,3] and training-free ([1] and ours). The upfront cost of extra training for training-based methods [2, 3] is significant and must be repeated if $r(x)$ or annealing schedule changes. Our inference-time method is more flexible, as it is instantly applicable to any new reward function or target distribution.
>
> Finally, these approaches are not mutually exclusive. DriftLite can be applied synergistically on top of a pre-trained control drift [2, 3] to correct for any residual variance or approximation error in the learned control, providing a "best of both worlds" solution.
>
> We added more discussion to explicitly contrast training-based and inference-time control in the main text, highlighting that our overhead is strictly at inference time, whereas methods like [2,3] load the cost beforehand, both share different use cases.
>
> ### *Comparison with Fine-Tuning Methods [4-6]*
>
> Thanks for the question on the empirical comparison with fine-tuning methods [4-6]. We do not seek direct comparison with [4-6] because they belong to a fundamentally different paradigm: training-based finetuning. Our paper, in contrast, focuses exclusively on the training-free, inference-time setting, for scenarios where retraining is impractical (e.g., proprietary, large, or frozen models).
>
> A brief summary of their training-based methodologies makes this distinction clear:
> - **Venkatraman et al. [4]**: This method amortizes posterior sampling by training a new diffusion model using a relative trajectory balance objective. This requires a full, data-free training phase for the new model.
> - **Domingo-Enrich et al. [5]**: This paper frames reward-based adaptation as a stochastic optimal control problem and uses the adjoint method to efficiently compute gradients of the control problem via gradient descent.
> - **Fan et al. [6]**: This is also an explicit finetuning method that defines the diffusion sampling process as an RL policy and updates weights using policy gradients to maximize a reward function.
>
> Because methods [4-6] all rely on **updating model weights** through complex training or finetuning, we believe they are not appropriate baselines for our training-free method. We instead compare against the state-of-the-art training-free baseline, Guidance-SMC (G-SMC), which operates under the same inference-time only constraints. We have clarified this distinction and added [4,6] to Related Work as training-based alternatives.

---

> ### Author Response · Authors · 2025-11-23
> **Authors' Reply to Reviewer ncQu (3/4)**
>
> ### Regarding Additional Experiments
>
> Thank you for suggesting conducting additional experiments to further enhance our manuscript. We agree that LJ-55 is an important and difficult system. We chose LJ-13 because it is already a challenging standard benchmark known for its rugged energy landscape. Our results in Fig. 5 show that this system is sufficiently complex to cause all baseline methods to fail, whereas our VCG-SMC is the only method to successfully recover the ground-truth 3-peak structure. This provides a powerful demonstration of our method's capabilities.
>
> Furthermore, generating high-quality base samples for LJ-55 remains an open research problem. Prior works have noted the difficulty of this system, for instance, [5] reports an ESS for LJ-55 that is an order of magnitude lower than for LJ-13 (4.42% vs. 57.98%). Similarly, recent efforts [5, 8] consistently yield ESS values below 10%, highlighting that the bottleneck lies in the base generative model itself.
>
> Since our paper focuses on inference-time scaling, a task that presumes the availability of a reasonably good base model to adapt, tackling a system like LJ-55 would deviate from our main message, as the challenge would likely stem from the base model generation rather than a clear test of our scaling algorithm. We believe our current results on LJ-13 make a clearer point, and we are actively working on scaling our method to these larger systems as a promising avenue for future research.
>
> ---
>
> ### Other Questions
>
> - **Evaluation of LJ-13**: We detail our evaluation methodology in the appendix. Ground-truth reference samples for the particle systems were generated via long-run underdamped Langevin dynamics simulations (detailed in Appendix B.3) . The metrics, including MMD, are then computed between our generated samples and this high-quality reference set, as described in Appendix B.5.
>
> - **2-Wasserstein Distance**: This is a great suggestion. We report Sliced-Wasserstein Distance (SWD) as a standard, efficient proxy for the 2-Wasserstein distance. Calculating the exact 2-Wasserstein distance is computationally prohibitive for our setup (30 dimensions, up to $N=2^{15}$ particles), as optimal transport typically scales like $\mathcal O(N^3 \log N)$. SWD circumvents this by averaging $P$ 1D projections, reducing complexity to $O(PN \log N)$. Our SWD results (e.g., Table 1) consistently demonstrate our method's superior performance. We have revised the appendix to clarify on this choice.
>
> - **Motivation of Reward-Tilting**: The reward function and its motivation are provided in Appendix B. The reward is a quadratic harmonic potential, $r(x) = -\frac{\lambda'}{2}\sum ||\boldsymbol r_i - \bar{\boldsymbol r}||^2$. This choice is not arbitrary, as it represents a physically-meaningful task. As shown in Eq. (B.1), applying this reward to the base Boltzmann distribution is equivalent to sampling from the same system but with a modified harmonic constraint strength. Motivated by recent work on mapping phase diagrams [9], this setup simulates altering physical conditions, such as subjecting particles to tighter confinement. Consequently, it serves as a rigorous test of the model's ability to accurately adapt its sampling dynamics to a precisely modified energy landscape.
>
> ---
>
> We hope the clarified theoretical positioning, explicit comparison to [2–6], and detailed description of our evaluation and reward design address your concerns about novelty and methodology. We would be grateful if you could reconsider your evaluation of our work in light of these clarifications.

---

> ### Author Response · Authors · 2025-11-23
> **Authors' Reply to Reviewer ncQu (4/4)**
>
> ### References
>
> [1] Skreta, Marta, et al. "Feynman-kac correctors in diffusion: Annealing, guidance, and product of experts." arXiv preprint arXiv:2503.02819 (2025).\
> [2] Albergo, Michael S., and Eric Vanden-Eijnden. "Nets: A non-equilibrium transport sampler." arXiv preprint arXiv:2410.02711 (2024).\
> [3] Vargas, Francisco, et al. "Transport meets variational inference: Controlled monte carlo diffusions." arXiv preprint arXiv:2307.01050 (2023).\
> [4] Venkatraman, Siddarth, et al. "Amortizing intractable inference in diffusion models for vision, language, and control." Advances in neural information processing systems 37 (2024): 76080-76114.\
> [5] Domingo-Enrich, Carles, et al. "Adjoint matching: Fine-tuning flow and diffusion generative models with memoryless stochastic optimal control." arXiv preprint arXiv:2409.08861 (2024).\
> [6] Fan, Ying, et al. "Dpok: Reinforcement learning for fine-tuning text-to-image diffusion models." Advances in Neural Information Processing Systems 36 (2023): 79858-79885.\
> [7] Klein, Leon, Andreas Krämer, and Frank Noé. "Equivariant flow matching." Advances in Neural Information Processing Systems 36 (2023): 59886-59910.\
> [8] Akhound-Sadegh, Tara, et al. "Iterated denoising energy matching for sampling from boltzmann densities." Proceedings of the 41st International Conference on Machine Learning. 2024.\
> [9] Schebek, Maximilian, et al. "Efficient mapping of phase diagrams with conditional Boltzmann Generators." Machine Learning: Science and Technology 5.4 (2024): 045045.\
> [10] Chen, Haoxuan, et al. "Solving inverse problems via diffusion-based priors: An approximation-free ensemble sampling approach." arXiv preprint arXiv:2506.03979 (2025).

---

> ### Comment · Reviewer_ncQu · 2025-11-24
>
> I appreciate the authors' response and for clarifying a number of my points. I am generally happy with the rebuttal, and most of my concerns have been resolved.
>
> **Comparison to training-based methods**: I really believe that the authors' should consider an experiment on a high-dimensional problem (e.g. image generation in [1]) while also comparing with a training-based method, e.g. [4]. While I understand that the authors' limit their study to inference-time methods, such an analysis would provide insights into how far can such inference methods go. In particular, the learned additional drift from [4] can just be low rank adaptation instead of having the same expressivity; and I think it is really important to understand the gap between the two paradigms on more complex settings. Or even understanding within one of the current experiments, how do inference-only methods compare to a training method because often we care about performant systems while sacrificing inference speed and having a sense of this tradeoff would be a really positive. Even further, this should be accompanied by quantitative numbers regarding inference speedup. Note that the training-based baselines don't need to be exhaustive, but one of the recent methods would be a great signal to have.
>
> > We are, to our knowledge, the first to explicitly formulate the control drift as a tool to actively minimize the variance of the SMC reweighting potential in the context of diffusion-based inference-time scaling, rather than passively accept the variance induced by a chosen drift.
>
> I think this statement is incorrect. In particular, [a, b, c, 2-4]  all look at different ways of training or adapting the drift functions by minimizing the variance of the re-weighting potential.
>
> [a] Sendera, Marcin, et al. "Improved off-policy training of diffusion samplers." Advances in Neural Information Processing Systems 37 (2024): 81016-81045.
>
> [b] Richter, Lorenz, and Julius Berner. "Improved sampling via learned diffusions." arXiv preprint arXiv:2307.01198 (2023).
>
> [c] Bengio, Yoshua, et al. "Gflownet foundations." Journal of Machine Learning Research 24.210 (2023): 1-55.

---

> ### Author Response · Authors · 2025-11-25
> **Authors’ Reply to Comment by Reviewer ncQu (1/2)**
>
> We sincerely thank the reviewer for the constructive follow-up and for bringing references [a-c, 2-4] to our attention. We are glad to hear that our initial rebuttal resolved most of your concerns. We address your remaining points on the comparison with training-based methods and the precision of our novelty claim below.
>
> ---
>
> ### Comparisons with Training-Based Drift-Learning Methods
>
> We fully agree that the works by Sendera et al. [a], Richter and Berner [b], Bengio et al. [c], Vargas et al. [2], Albergo and Vanden‑Eijnden [3], and Venkatraman et al. [4] are highly relevant. Understanding the trade‑offs between amortized training (learning a control policy over many iterations) and inference‑time control (solving and applying control at inference time) is indeed critical for the field.
>
> Conceptually, these papers and our work are closely connected. Since the optimal control yields zero-variance in inference, so broadly speaking, all these methods, including ours, share the same goal of “reducing” some "variance", just with different methods:
> - Sendera et al. [a] and Richter and Berner [b] formulate objectives that explicitly control the variance of log‑density ratios between a target path measure and a parametrized diffusion sampler. Minimizing these log‑variance or VarGrad‑type objectives amounts to learning a drift that reduces importance‑weight variance over trajectories.
> - Bengio et al. [c] show that trajectory balance and its variance‑based variants in GFlowNets can be interpreted in the same way: at optimum, the log ratio between a forward path distribution and a reward‑reweighted backward reference is constant along trajectories, corresponding to zero‑variance importance weights.
> - Vargas et al. [2] and Albergo and Vanden‑Eijnden [3] consider controlled diffusions and non‑equilibrium transport samplers. Both show that there exists an optimal additional drift that in principle eliminates AIS/SMC reweighting factors, and they train expressive control networks to approximate this drift by minimizing path‑space KL or related objectives.
> - Venkatraman et al. [4] apply a relative trajectory‑balance objective in diffusion models, learning a posterior diffusion whose path distribution matches a prior path distribution reweighted by a constraint $r(x)$. In their variance‑based form, the objective again targets a constant log‑ratio between posterior and reward-tilted prior path measures.
>
> However, these methods do **operate in a different regime from the one we focus on**. In all of [a-c, 2–4], the main goal is **to train or fine‑tune a new sampler** (e.g., a controlled diffusion or posterior diffusion) via **gradient‑based optimization over many trajectories**. The learned drift is typically neural network parametrized and amortized over many future queries. By contrast, our work is explicitly framed around **diffusion‑based inference‑time scaling for pre‑trained models**:
> - **Training-Based [a-c, 2–4] vs Inference-time (Ours)**: As stated in our introduction, we target the setting where a base diffusion model is fixed, and we must adapt to new distributions without retraining. We exploit the drift-potential degree of freedom to compute a control drift that is computed efficiently at each time step.
> - **Trajectory-Level Objective [a-c, 2–4]  vs Local-In-Time Control (Ours)**: Our drift is obtained by minimizing the instantaneous variance of the SMC reweighting potential under the current particle distribution. This involves no parameter updates or backpropagation through diffusion models for training, as it is a purely local-in-time calculation.
> - **Large-Scale Parametrization [a-c, 2-4] vs Lightweight Control (Ours)**: In contrast to large-scale neural networks used to parametrize the control/policy/drift, our method aims to devise lightweight control techniques that incurs small computational overhead by searching the optimal control within a low‑dimensional space spanned by only a few basis functions.

---

> ### Author Response · Authors · 2025-11-25
> **Authors’ Reply to Comment by Reviewer ncQu (2/2)**
>
> Regarding the empirical comparison you suggest (e.g., Venkatraman et al. [4] or Albergo and Vanden‑Eijnden [3] on high‑dimensional problems such as image generation), we completely agree that this would be very informative. Implementing and tuning a full training pipeline for these methods on our large settings (e.g., protein–ligand co‑folding with black‑box rewards) would be a substantial effort in its own right beyond the immediate scope of this rebuttal. Within the constraints of this work, we chose to concentrate on a detailed analysis of training‑free inference‑time control methods only and we will continue working on the comparison and the synergistic use of our method with these training-based methods in future work.
>
> We take the reviewer’s suggestions seriously and we have further revised our manuscript to make these distinctions more explicit, by:
> - Extending the introduction to explicitly mention and compare with learning-based control methods [b,2,3], trajectory balance-based GFlowNet methods [a,c,4], while clarifying the three advantages of our methods: lightweight (solving a small linear system vs. high-dimensional pde), training-free (inference-time scaling vs extra training), and on-the-fly (local-in-time control vs trajectory-level objectives).
> - Extending the discussion in conclusion that while this work focuses on the inference-time scaling methods of diffusion models, it is a natural and important direction to compare with and apply our DriftLite method on top of the training-based methods [a-c, 2-4] to achieve “best-of-both-worlds” solutions.
>
> ---
>
> ### On the Novelty Claim about Variance Minimization
>
> The reviewer is uncertain of our sentence in the rebuttal
> > We are, to our knowledge, the first to explicitly formulate the control drift as a tool to actively minimize the variance of the SMC reweighting potential in the context of diffusion-based inference-time scaling, rather than passively accept the variance induced by a chosen drift.
>
> in light of that [a-c, 2-4] consider objectives that reduce the variance of reweighting potentials or log‑density ratios.
>
> We very much appreciate this observation and fully agree that these works are part of the same broader line of research in which drift functions are chosen or learned so as to reduce importance‑weight variance. In fact, we view our method as conceptually connected to them: all of these approaches exploit a change of measure on path space to flatten (or at least reduce the spread of) a log‑ratio that plays the role of an importance weight and the optimal control/drift/policy is the one with zero variance.
>
> At the same time, our novelty claim is intentionally scoped to the **specific context of diffusion‑based inference‑time scaling for pre‑trained models and to a training‑free control formulation**: [a-c, 2-4] all define training objectives that learn a drift or policy by minimizing a variance‑like or KL‑like functional on entire trajectories, with the goal of producing a new amortized sampler whose path distribution has low‑variance importance weights with respect to a target reference. In contrast, we work directly at the level of the Fokker–Planck equation for inference‑time scaling of an existing diffusion model, and we show that this PDE admits a drift–potential gauge freedom that can be exploited at inference time to reduce SMC weight variance.
>
> Within this scope, *diffusion‑based inference‑time scaling of a pre‑trained model*, that is, with no extra training in the search of the control, we believe our original sentence remains valid: to the best of our knowledge, prior work does not formulate and exploit this drift–potential gauge specifically as an inference‑time tool for actively minimizing the SMC reweighting potential variance in this setting.
>
> That said, we recognize that this distinction may not have been sufficiently clear. To avoid any potential misunderstanding, we have revised the manuscript to make this scope explicit and accurate, by adding
> > In this work, we focus specifically on the inference-time scaling regime for pre-trained diffusion models, without any additional training, and develop a control-theoretic perspective within this setting.
>
> to the central position of introduction to aid the readers' understanding and positioning of our work.
>
> We hope this clarifies how we see our contribution relative to the works highlighted by the reviewer: we regard them as highly relevant and conceptually close, but operating in a different (training‑based, amortized) regime, while our focus is on training‑free control for diffusion‑based inference‑time scaling.
>
> ---
>
> Thanks again for the constructive reply. We hope that our response and the corresponding clarifications and modifications to the manuscript address the remaining concerns, and we would be grateful if you could reconsider the evaluation of our paper in light of these clarifications.

---

> > ### Comment · Reviewer_ncQu · 2025-11-25
> >
> > Thanks for the response. I feel like the authors may have misunderstood my comment earlier. I fully understand and agree that the authors provide an inference-time method, which while maintains the same goals as related work - i.e. reduction of variance of importance sampling weights, does so without additional training. I also completely understand that such methods can have widespread impact since one may often not have access to compute to be able to train an additional complex drift, and this work aims to provide quite a viable solution.
> >
> > My concern is that while the draft shows that it outperforms other inference time methods, it does not show what the gap to training-based methods is and how much of that gap is bridged by the proposed method. This is important because often we care about performance first and compute limitations second, and if the method is quite inferior to learning of amortized samplers, that should be shown. The contributions of the work would still hold in the sense that it still bridges that gap, but getting an estimate of that gap is, in my opinion, really important. I understand that training and implementing a full large-scale experiment might be out of scope, but it would be nice to have some form of comparison to understand the gap, which could be shown even in some nontrivial toy setting.
> >
> > In fact, it is not quite clear to me where the complexity comes in? Couldn't the authors define a neural network drift $b_{\theta, t}$ in Equation 3.3 and solve that optimization to learn the parameters $\theta$?

---

> ### Author Response · Authors · 2025-12-01
> **Authors’ Reply to Comment by Reviewer ncQu (1/2)**
>
> We thank the reviewer again for the suggestion to compare our proposed method with training-based alternatives. We agree that understanding the performance gap between inference-time adaptation and amortized training is crucial for contextualizing our contributions.
>
> Within the tight time constraints of the discussion period, we have implemented a prototype of a training-based method, which we refer to as **Neural Controlling Guidance (NCG)**, to provide this comparison.
>
> ---
> ### Algorithm Settings
>
> We implemented a training-based baseline that directly optimizes the variance of the importance weights. Following the reviewer’s suggestion, we parameterize the control drift $\boldsymbol{b}_{t, \theta}(\boldsymbol x)$ using a neural network with parameters $\theta$, taking time $t$ and position $\boldsymbol x$ as inputs.
>
> - **Architecture:** We utilize a U-Net architecture with encoder feature dimensions $[128, 256, 512]$ and decoder feature dimensions $[256, 128]$.
> - **Objective:** To prevent numerical instability and overfitting, we include a regularization term. The loss function is thus $$ \mathcal L_t(\theta) = \mathrm{var}_{q_t}[g_t + h_t(\cdot; \boldsymbol b _{t, \theta})] + \lambda ||\boldsymbol b_t||^2 $$ where $\lambda = 0.1$ was selected after hyperparameter tuning.
> - **Training Procedure:** We maintain an ensemble of 8192 particles (matching the inference-only baseline). After each step, we backpropagate the gradients $\nabla_\theta \mathcal L_t(\theta)$ of the regularized variance-minimization objective with respect to the control drift parameters $\theta$, using a learning rate of $5 \times 10^{-4}$.
> - **Refinement:** The particles and their weights are updated using the learned control drift and the resulting potential. We perform this training-inference process over multiple rounds, treating the updated dynamics from the previous round as the base for the next.
>
> ---
> ### Results
>
> We compared the performance of our proposed training-free methods (VCG-SMC, ECG-SMC) against standard baselines (PG, G-SMC) and the Neural Controlling Guidance (NCG) method. We report metrics for various rounds of NCG (denoted as **NCG-SMC-$x$**) to illustrate the training trajectory, as well as an inference-only variant (**NCG-SMC-20 (Inference Only)**) which uses the learned network from the last round to highlight the computational baseline.
>
> The results are as follows:
> | Method | MMD | SWD | Mean L2 | $\Delta$ NLL | Wall-Clock Time (s) |
> | :--- | :--- | :--- | :--- | :--- | :--- |
> | **PG** | 0.825 | 21.86 | 66.09 | -0.65 | 4.59 |
> | **G-SMC** | 0.143 | 6.45 | 24.94 | 0.04 | 5.53 |
> | **NCG-SMC-1** | 0.437 | 12.29 | 51.44 | 0.39 | 70.79 |
> | **NCG-SMC-5** | 0.135 | 3.53 | 13.67 | 0.34 | 307.34 |
> | **NCG-SMC-9*** | 0.091* | 2.45* | 9.11* | 0.05 | 543.77 |
> | **NCG-SMC-12** | 0.223 | 13.35 | 37.96 | 0.01* | 721.23 |
> | **NCG-SMC-16** | 0.143 | 7.61 | 21.73 | 0.16 | 957.72 |
> | **NCG-SMC-20** | 0.199 | 9.36 | 24.02 | 0.08 | 1194.31 |
> | **NCG-SMC-20 (Inference Only)** | 0.190 | 8.91 | 29.04 | 0.26 | 240.88 |
> | **VCG-SMC** | **0.017** | **0.52** | **1.23** | **0.14** | **7.83** |
> | **ECG-SMC** | 0.019 | 0.68 | 1.83 | 0.14 | 6.99 |
> *\*Best performing NCG round.*

---

> ### Author Response · Authors · 2025-12-01
> **Authors’ Reply to Comment by Reviewer ncQu (2/2)**
>
> ### Discussion
>
> The results allow for a nuanced comparison between our training-free approach and the training-based NCG method:
> - **Performance Gap**: Contrary to the concern that inference-time methods might be inferior to learning-based samplers, our results show that VCG-SMC achieves an MMD of 0.017 compared to 0.091 for the best Neural baseline (Round 9). This demonstrates that DriftLite is not merely a low-compute compromise but a highly competitive solution. By solving for the locally optimal control analytically, DriftLite captures fine-grained variance reduction that is difficult for a global neural network to approximate without extensive and perfectly stable optimization.
> - **Instability of Training:** As observed in the table, the performance of NCG is **non-monotonic**. While it improves significantly up to Round 9 (MMD 0.091), it degrades in subsequent rounds (e.g., Round 12 MMD spikes to 0.223) despite the accumulated training time. This highlights the inherent instability in training parameterized drift controls via variance objectives. We found that careful hyperparameter tuning and regularization (such as the $\lambda ||\boldsymbol b_t||^2$ term) were necessary to prevent immediate divergence, yet achieving stable convergence remains challenging. Unleashing the full potential of the NCG method may require much more engineering efforts comparing with our plug-and-play DriftLite method.
> - **Computational Cost:** The table also highlights the computational overhead. The NCG-SMC-20 (Inference Only) row shows that merely evaluating the large control network at every step takes $\sim$240 seconds, which is nearly 30x slower than our DriftLite methods (~7-8 seconds). This confirms that even without the cost of backpropagation/training, using a high-capacity neural network for control drift introduces significant inference latency compared to our lightweight linear solver, let alone the training cost.
>
> We do not claim our NCG implementation is fully optimized; with additional engineering and tuning, its performance could likely be improved. However, the key message is that DriftLite is a lightweight, stable, and highly effective method that performs well "out of the box" without the complexities of training dynamics. Future work could seek synergy between these methods, for instance, using DriftLite to stabilize the training of NCG or to correct its residuals during inference.
>
> ---
> ### Final Remarks
>
> We will incorporate these extensive experiments and the new NCG comparisons into the final manuscript to significantly strengthen the positioning of our work. As the discussion period concludes, we would like to summarize how our rebuttals have addressed your primary concerns:
> - **Scope and Novelty**: We have clarified that our contribution is not the discovery of the degree of freedom in the Fokker-Planck equation, but its specific instantiation for training-free, inference-time variance minimization. We have explicitly distinguished our work from amortized training methods [2-4, a-c] and will revise the text to reflect this precise scope.
> - **Comparison to Training-Based Methods**: Through the new NCG experiments, we have quantified the gap between inference-time and training-based control. The results demonstrate that DriftLite is not merely a compromise; it outperforms unoptimized training-based baselines in stability and wall-clock time while achieving superior sample quality.
> - **Empirical Rigor**: We have addressed your suggestions by explaining the choice of metrics (SWD), benchmarks (LJ-13 vs LJ-55), and reward-tilting settings, and detailing the computational trade-offs.
>
> We believe DriftLite represents a "sweet spot" in the landscape of diffusion control: it offers the theoretical rigor of optimal control without the burden of training, making it an immediately deployable tool for pre-trained models. We thank you for your rigorous review, which has undoubtedly improved the quality and clarity of this submission. We hope these clarifications and additional results have fully resolved your concerns.

---

### Official Review · Reviewer_12SX · 2025-11-03

**Soundness:** 4
**Presentation:** 3
**Contribution:** 3
**Rating:** 8
**Confidence:** 3

**Summary:**

This paper introduces DriftLite, a training-free method for steering diffusion models at inference time using an improved sequential Monte Carlo (SMC). To mitigate the path degeneracy problem common in SMC, the authors derive an optimal, variance-minimizing drift for the governing Feynman-Kac PDE and propose two practical approximations: Variance-Controlling Guidance (VCG) and Energy-Controlling Guidance (ECG). These approximations can be obtained by solving a linear system at each sampling step, adding only a small computational overhead. The methods are analyzed on Gaussian mixture models, where they are empirically shown to reduce variance and improve sample quality. They are also tested on practical benchmarks, molecular systems and protein-ligand co-folding, demonstrating improved inference-time steering capability over the standard SMC baseline.

**Strengths:**

1. The paper is clearly written, and the core mathematical arguments are generally well-explained.
2. The work tackles a significant but often-neglected challenge of the weight degeneracy (or path degeneracy) in SMC-based diffusion steering.
3. The paper translates a mathematical insight into novel and practical (training-free) algorithms.
4. Their experiments are well-designed to support their arguments. The proposed methods clearly outperform the SMC baseline on real-world benchmarks like molecular systems and protein-ligand co-folding.

**Weaknesses:**

1. The method's practical limitations could be discussed in more detail. It introduces a significant computational overhead, with experiments showing up to a 6x increased runtime over the SMC baseline. Furthermore, the reward-tilting framework fundamentally relies on access to the gradient of the reward, which is inaccessible in many black-box applications, limiting its practical scope.
2. The paper would be strengthened by adding empirical analysis of the approximation error from VCG and ECG.
3. No source code is provided.

**Questions:**

1. I have little expertise in functional analysis and found the formal proof of Proposition A.5 difficult to follow. Could you provide a more intuitive explanation for why solving the Poisson equation (Eq. 3.2) yields the optimal control?
2. Line 269, "... where reweighting can be unstable": Aren't VCG and ECG without resampling still weighted with path-level weights? Those weights should still have high variance. I don't understand the rationale behind this (if you want to mitigate the path degeneracy from resampling, then you can consider using tempered weights for resampling, e.g., [1]).
3. Why were different $\gamma$ values used for each figure 1, 2, and 3?
4. What is the main computational bottleneck of the algorithm in practice? (Hutchinson's estimator? or solving the linear equation?)
5. If $r$ is given by a large neural network, can this method still be used (in terms of memory and time complexity)?
6. Why didn't you consider the ALDP experiment (which is more multi-modal compared to LJ-13, to my knowledge)?
7. (suggestion) Line 417, "from $T=1.0$ to $0.4$": it might be better to say "with $\gamma = 2.5$" to avoid any confusion.

---

References
[1] Choi, Sanghyeok, et al. "Reinforced sequential Monte Carlo for amortised sampling." arXiv preprint arXiv:2510.11711 (2025).

---

LLM usage disclosure: I used LLM to check the grammar and make each sentence clearer.

---

> ### Author Response · Authors · 2025-11-23
> **Authors' Reply to Reviewer 12SX (1/2)**
>
> Thank you for your thoughtful and positive review, and for highlighting both the novelty and practicality of our method. We address your concerns point by point below.
>
> ---
>
> ### Regarding Practical Limitations
>
> Thank you for encouraging us to clarify the practical limitations. Our use of “lightweight” is primarily in contrast to retraining or learning-based control, which requires training an auxiliary neural network with a new, often complex objective [2,3]. In contrast, DriftLite adds only an online control step: it is training‑free, computed on‑the‑fly, and thus “lightweight” in terms of algorithmic and engineering complexity rather than raw wall‑clock time.
>
> Within DriftLite, the main computational cost is not solving the tiny 3×3 linear system, but evaluating the basis functions, especially the score Laplacian via Hutchinson's estimator. The 6x slowdown on the GMM task is a worst-case artifact of that setup where the score is analytical and thus extremely cheap, so any additional work looks disproportionally expensive. In the more realistic DW‑4 experiment (EGNN-based score), the denoiser evaluation is the true bottleneck, and our overhead is approximately 2×, which we believe is a reasonable cost for the significant gains in stability and correctness (e.g., Fig. 5).
>
> We have revised references to “minimal overhead” to “modest and scalable overhead” and clarify that “lightweight” refers to training‑free, structurally simple control, as compared to methods that require training an extra network.
>
> ---
>
> ### Regarding Reward Applicability
>
> We agree that the reliance on $\nabla r(x)$ limits direct applicability to settings with a differentiable reward. Our work is explicitly situated in the reward-tilting with known potential regime, typical in scientific applications where $r(x)$ is a physics-based energy with analytic gradients (e.g., our protein–ligand task). We made this scope explicit in the main text.
>
> For black-box rewards, DriftLite can still be applied by estimating $\nabla r(x)$ via Hutchinson’s estimator, at the cost of extra backpropagation. This increases overhead, but does not change the algorithmic structure: the control drift remains low‑rank, training‑free, and is still obtained by solving a small linear system.
>
> ---
>
> ### Regarding Approximation Error
>
> We agree that analyzing the approximation error of VCG/ECG is important. Direct comparison to the true optimal control $\boldsymbol b_t^*$ is intractable, as it requires numerically solving a high-dimensional Poisson equation at each step. Instead, our Iterative Refinement experiment (App. C.3) serves as a practical proxy: after each DriftLite pass, we treat the updated dynamics as a new base and apply DriftLite again.
>
> Figures 17–18 show that the variance of the residual potential decreases monotonically across refinement rounds, and ESS steadily improves. Our goal is not to drive the approximation error to zero, but to reduce variance enough to prevent ESS collapse. The refinement results empirically confirm that our low-rank control is effective and systematically improvable. We made this proxy interpretation more explicit when discussing iterative refinement.
>
> ---
>
> ### Regarding Open-Sourcing
>
> We will release our implementation as open-source upon acceptance with instructions for reported experiments.

---

> ### Author Response · Authors · 2025-11-23
> **Authors' Reply to Reviewer 12SX (2/2)**
>
> ### Other Questions
>
> - **Intuition of Poisson equation**: Thanks for prompting us to add intuition here. Informally, our goal is to find a control $b_t$ that makes the new potential $\phi_t = g_t + h_t(\cdot; \boldsymbol b_t) = 0$. From our proof (App. A.3), $h_t$ is defined such that $q_t h_t = \nabla \cdot (q_t \boldsymbol b_t)$. We then assume a curl-free control $\boldsymbol b_t = \nabla A_t$. Substituting these into our goal gives: $g_t + \frac{1}{q_t} \nabla \cdot (q_t \nabla A_t) = 0$, that is the Poisson equation: $\nabla \cdot (q_t \nabla A_t) = -q_t g_t$. Thus, $A_t$ that solves this equation gives the exact control $\boldsymbol b_t$ needed to make the residual potential (and its variance) zero. We added a short paragraph with this intuition next to Prop. 3.2 and point more clearly to App. A.5 for the complete proof.
>
> - **VCG/ECG reweighting**: You are absolutely correct that VCG/ECG (without SMC) are still weighted particle methods. We still maintain path‑level weights but the key difference is what they are reweighted with. The G-SMC baseline is reweighted using the original potential $g_t$, which is high‑variance and leads to weight degeneracy. In contrast, our methods reweight using our low-variance residual potential $\phi_t$​, whose variance we actively minimize. As Fig. 2 shows, $\text{Var}(\phi_t)$ is orders of magnitude lower than $\text{Var}(g_t)$, so the ESS remains high even without resampling, whereas G‑SMC collapses. We clarified this point in the text and avoid wording that could be read as claiming that reweighting is completely avoided in these variants.
>
> - **Different $\gamma$’s**: This was to demonstrate the robustness of our method across different task difficulties. We provide a full, systematic sweep over $\gamma$ values in Appendix C.1 (Table 3), which confirms our conclusions hold across all settings.
>
> - **Bottleneck**: The linear solve is trivial. The bottleneck is building the $A_t, c_t$ matrices, which requires evaluating the basis functions, including the Laplacian $\Delta \log p_t$ via Hutchinson's estimator.
>
> - **NN reward**: We refer to “Regarding reward applicability” section.
>
> - **ALDP experiment**: We agree that ALDP is an excellent and highly multimodal benchmark. We chose the LJ‑13 system because it is a standard and already challenging benchmark. Our experiments (Fig. 5) show that this task is difficult enough that all baselines, including Guidance‑SMC, fail to capture the correct physical structure (in particular, the third peak in the RDF), whereas VCG‑SMC is the only method that discovers all three modes and matches the ground‑truth distribution with high precision.
> Moreover, generating high‑quality samples for more complex systems like ALDP is an open research problem in its own right. As reported in [1], ESS for ALDP is 0.69% compared with 57.98% for LJ‑13, highlighting its difficulty. Our paper focuses on inference‑time scaling given a reasonably good base model; on ALDP, the main bottleneck would likely be the base generative model rather than the scaling algorithm. We clarified this rationale and point to ALDP and similar systems as promising future applications.
> - **Style**: We have adopted your suggestion and rephrased “from T=1.0 to 0.4” as “with γ = 2.5” to avoid confusion.
>
> ---
>
> Thank you again for your careful, positive assessment. We hope these clarifications on overhead, scope, and intuition further strengthen your confidence in the work.
>
> ---
> ### Reference
>
> [1] Klein, Leon, Andreas Krämer, and Frank Noé. "Equivariant flow matching." Advances in Neural Information Processing Systems 36 (2023): 59886-59910.\
> [2] Albergo, Michael S., and Eric Vanden-Eijnden. "Nets: A non-equilibrium transport sampler." arXiv preprint arXiv:2410.02711 (2024).\
> [3] Domingo-Enrich, Carles, et al. "Adjoint matching: Fine-tuning flow and diffusion generative models with memoryless stochastic optimal control." arXiv preprint arXiv:2409.08861 (2024).

---

> > ### Comment · Reviewer_12SX · 2025-11-23
> >
> > Thank you for your responses. I will keep my score and increase my confidence to 4.

---

> > > ### Author Response · Authors · 2025-12-01
> > > **Authors' Reply to Reviewer 12SX**
> > >
> > > Thank you again for your positive evaluation of our work and we are glad that our response has cleared your concerns.
> > >
> > > Best regards,\
> > > Authors of Submission23218

---

### Meta-Review · Area_Chair_4y9Z · 2026-01-05

**Summary:**

This paper introduces a refined Sequential Monte Carlo (SMC) framework for guiding diffusion models at inference time toward specific reward-driven target distributions. While previous methods used a "naive" SMC approach to approximate reward-tilted distributions, this work identifies and exploits a mathematical "design space" within the underlying Feynman-Kac equations to optimise particle transport and weight stability. The work has been characterised as principled and theoretically grounded, and the effectiveness of the method is underscored through thoroughly executed experiments.

Several reviewers expressed concerns regarding the computational cost, which was acknowledged as a limitation of the work by the authors, and the lack of comparisons to training-based methods. In addition, several reviewers mentioned limited novelty, as much of the theory already exists in the literature, and an appropriate contextualisation to the SMC literature would make this clearer.

**Reviewer Concerns:**

Some of the reviewer concerns have been addressed (e.g., approximation error ablation [12SX]), while others are more of a conceptual nature and remain, such as the additional computational and memory cost, the limited novelty with regard to the theoretical results. Missing comparisons that reviewers mentioned have been partially addressed and indicate that the method seems promising.

**Reviewer Scores:**

The following reviewing scores are to be expected after the rebuttal:

- [12SX]: Likely to have kept the high score.
- [ncQu]: Possibly increased the score to a 5.
- [hiDh]: Likely to have kept the high score or lowered it. (! Note that this reviewer is likely not an expert !)
- [cAK7]: Mentioned to have been willing to increase the score (5/6).

---

### Decision · Program_Chairs · 2026-01-26

Accept (Poster)